# Yet Another ICU Benchmark: A Flexible Multi-Center Framework for Clinical ML

**Robin van de Water**[1] ⊙*     **Hendrik Schmidt**[1] ⊙     **Paul Elbers**[2] ⊙
**Patrick Thoral**[2] ⊙     **Bert Arnrich**[1] ⊙     **Patrick Rockenschaub**[3] ⊙
[1]Hasso Plattner Institute, University of Potsdam, Germany
[2]Amsterdam UMC, Vrije Universiteit, Amsterdam, The Netherlands
[3]Lab for AI in Medicine, Charité - Universitätsmedizin Berlin, Germany

## Abstract

Medical applications of machine learning (ML) have experienced a surge in popularity in recent years. The intensive care unit (ICU) is a natural habitat for ML given the abundance of available data from electronic health records. Models have been proposed to address numerous ICU prediction tasks like the early detection of complications. While authors frequently report state-of-the-art performance, it is challenging to verify claims of superiority. Datasets and code are often not published, and cohort definitions, preprocessing pipelines, and training setups are difficult to reproduce. This work introduces *Yet Another ICU Benchmark* (*YAIB*), a modular framework that allows researchers to define reproducible and comparable clinical ML experiments; we offer an end-to-end solution from cohort definition to model evaluation. The framework natively supports most open-access ICU datasets (MIMIC III/IV, eICU, HiRID, AUMCdb) and is easily adaptable to future and custom ICU datasets. Combined with a transparent preprocessing pipeline and extensible training code for multiple ML and deep learning models, *YAIB* enables unified model development, transfer, and evaluation. Our benchmark comes with five predefined established prediction tasks (mortality, acute kidney injury, sepsis, kidney function, and length of stay) developed in collaboration with clinicians. Adding further tasks is straightforward by design. Using *YAIB*, we demonstrate that the choice of dataset, cohort definition, and preprocessing have a major impact on the prediction performance, often more so than model class, indicating an urgent need for *YAIB* as a holistic benchmarking tool. We provide our work to the clinical ML community to accelerate method development and enable real-world implementations.
**Software Repository:** `https://github.com/rvandewater/YAIB`

## 1 Introduction

The intensive care unit (ICU) has long been a focus for research into data-driven decision support, owing to the impact of medical decisions as well as the breadth and depth of data collected in this setting (Johnson et al., 2017). The COVID-19 pandemic confirmed the need for reliable machine learning (ML)-based clinical decision support that can alert healthcare professionals to worsening patient states, help them make a clinical diagnosis, or recommend treatment (Medic et al., 2019).

Despite a steep increase in the number of published ICU prediction models (Shillan et al., 2019), hardly any have made their way into clinical practice (Eini-Porat et al., 2022; Fleuren et al., 2020b). A major obstacle to translation is an ongoing lack of comparability and reproducibility (Johnson et al., 2017). By using custom datasets and definitions, preprocessing pipelines, and evaluation schemes, the benefits of novel models are conflated with differences between patient case mix, task definitions, and cohort selection (Sarwar et al., 2023; Kelly et al., 2019). Reviewing models for early prediction of sepsis, for example, Moor et al. (2021b) found that the definition of sepsis, the time of prediction, and the available features differed substantially between the 22 included studies; similar results were found in an earlier review (Fleuren et al., 2020a). Even among studies from the same research

---

*Corresponding author email: robin.vandewater@hpi.de

group (Hyland, 2020; Yèche et al., 2022), cohort definitions may vary substantially, precluding a meaningful comparison. Inconsistencies in imputation and feature extraction further complicate an objective evaluation of research progress.

The increasing availability of open-access ICU datasets is a first, important step towards urgently needed model comparability (Sauer et al., 2022a). However, models derived from the same dataset may still vary considerably in their analytical setup. Earlier work has therefore created benchmarks that establish a single pipeline for preprocessing and modeling (Yèche et al., 2022; Harutyunyan et al., 2019). These benchmarks are hard-coded for a given dataset, following proprietary formats and supporting a limited, fixed set of tasks. Extending an existing benchmark to include new datasets or tasks requires changes to the benchmark's — often lightly documented — source code. Despite the existence of multiple benchmarks, new models are therefore rarely evaluated on more than one dataset or do not use *any* benchmark (Shillan et al., 2019).

We address this gap by providing *Yet Another ICU Benchmark* (*YAIB*) as a modular multi-dataset framework specifically designed for extensibility. Building on recent work to harmonize ICU data (Bennett et al., 2023) (i.e., match time-scale, clinical definitions, and units across datasets), we standardize the entire modeling workflow from the definition of clinical concepts (a medical abstraction to facilitate patient care) and data extraction to model fitting and evaluation across several established open-source ICU datasets (Sauer et al., 2022a). We provide a predefined set of common prediction tasks, developed in collaboration with clinical intensivists, that can be easily extended to fit user needs. Our benchmark, by default, provides endpoint prediction for ICU mortality, sepsis (Singer et al., 2016), acute kidney injury (AKI) (KDIGO, 2012), kidney function (KF), and length of stay (LoS). With this work, we aim to **(1)** dramatically reduce the overhead of developing new ICU prediction methods, **(2)** provide a transparent, open-source, and reproducible definition of experiments, and **(3)** unify ML workflows for ICU prediction modeling.

## 2 RELATED WORK

Our work builds upon several previous efforts to harmonize the definition, development, and evaluation of ICU prediction models. *YAIB* combines these existing works in a novel, end-to-end fashion to enable quick, reproducible, and comparable model development.

**Publicly available ICU datasets**    Our benchmark currently supports four established ICU datasets (Sauer et al., 2022b): the Medical Information Mart for Intensive Care (MIMIC) version III (Johnson et al., 2016) and IV (Johnson et al., 2023), the eICU Collaborative Research Database (eICU) (Pollard et al., 2018), the High Time Resolution ICU Dataset (HiRID) (Hyland, 2020), and the Amsterdam-mUMCdb (AUMCdb) (Thoral et al., 2021). These datasets contain similar data items but differ in size and scope (Table 13). Together, they cover 334,812 ICU stays. We plan to integrate two recently released ICU datasets in the future (Rodemund et al., 2023; Jin et al., 2023).

**Benchmarks**    To improve comparability between models trained on these ICU datasets, several benchmarks or benchmark-like applications have been developed (Table 1). These solutions mainly differ in the tasks and models they support. Notably, existing benchmarks heavily focus on benchmarking results, often hardcoding key steps like data extraction, task definition, preprocessing, feature generation, and sometimes model training. While they may reduce implementation overhead when evaluating new ML approaches, present benchmarks are difficult to adapt to user requirements. Core code base changes are often necessary if the users' problems do not fit into the provided task definitions. Even advanced modeling frameworks such as Jarrett et al. (2021) and Saveliev & van der Schaar (2023) share this weakness, as they do not support reproducible data extraction or task definitions; thus, they do not provide an end-to-end solution like *YAIB*.

**Multi-dataset support**    Due to considerable heterogeneity in data structure, existing benchmarks tend to focus on a single dataset, most frequently MIMIC-III. As MIMIC-III also has a large existing user base (Syed et al., 2021), it thus often becomes the default choice (Shillan et al., 2019). This has potentially resulted in a self-enforcing bias towards the MIMIC-III datasets, which represent a single-center US population. Even frameworks that work with its successor MIMIC-IV lack backward compatibility (Mandyam et al., 2021; Gupta et al., 2022). Among the few multi-dataset solutions, (Tang et al., 2020) operates on both eICU and MIMIC-III, but lacks many of the model architectures found in others works and does no longer appear to be in active development. Oliver et al. (2023) provides a hardcoded pipeline to combine several datasets without providing cohort definitions,

TABLE 1: *Comparison of existing benchmarks and YAIB on ICU data, ordered by publication date.*

| | | Johnson et al. | Purushotham et al. | Harutyunyan et al. | Barbieri et al. | Wang et al. | Jarrett et al. | Sheikhalishahi et al. | Tang et al. | Yèche et al. | Mandyam et al. | Gupta et al. | Yang et al. | Saveliev et al. | Oliver et al. | **YAIB (ours)** |
|---|---|---|---|---|---|---|---|---|---|---|---|---|---|---|---|---|
| **Datasets** | MIMIC-III | ✓ | ✓ | ✓ | ✓ | ✓ | ✓ | ✗ | ✓ | ✗ | ✗ | ✗ | ✓ | ✗ | ✓ | ✓ |
| | MIMIC-IV | ✗ | ✗ | ✗ | ✗ | ✗ | ✗ | ✗ | ✗ | ✗ | ✓ | ✓ | ✓ | ✗ | ✓ | ✓ |
| | eICU | ✗ | ✗ | ✗ | ✗ | ✗ | ✗ | ✓ | ✓ | ✗ | ✗ | ✗ | ✓ | ✗ | ✓ | ✓ |
| | HiRID | ✗ | ✗ | ✗ | ✗ | ✗ | ✗ | ✗ | ✗ | ✓ | ✗ | ✗ | ✗ | ✗ | ✓ | ✓ |
| | AUMCdb | ✗ | ✗ | ✗ | ✗ | ✗ | ✗ | ✗ | ✗ | ✗ | ✗ | ✗ | ✗ | ✗ | ✓ | ✓ |
| **Prediction tasks** | Mortality risk | ✓ | ✓ | ✓ | ✗ | ✓ | ✗ | ✓ | ✓ | ✓ | ✓ | ✗ | ✓ | ✗ | ✗ | ✓ |
| | Circulatory failure | ✗ | ✗ | ✗ | ✗ | ✗ | ✗ | ✗ | ✗ | ✓ | ✗ | ✓ | ✗ | ✗ | ✗ | * |
| | Kidney function (KF) | ✗ | ✗ | ✗ | ✗ | ✗ | ✗ | ✗ | ✗ | ✓ | ✗ | ✓ | ✗ | ✗ | ✗ | ✓ |
| | Respiratory failure | ✗ | ✗ | ✗ | ✗ | ✗ | ✓ | ✗ | ✓ | ✓ | ✗ | ✓ | ✗ | ✗ | ✗ | * |
| | Sepsis | ✗ | ✗ | ✗ | ✗ | ✗ | ✗ | ✗ | ✗ | ✗ | ✗ | ✗ | ✗ | ✗ | ✗ | ✓ |
| | Acute kidney injury | ✗ | ✗ | ✗ | ✗ | ✗ | ✗ | ✗ | ✗ | ✗ | ✗ | ✗ | ✗ | ✗ | ✗ | ✓ |
| | Phenotyping§ | ✗ | ✓ | ✓ | ✗ | ✗ | ✗ | ✓ | ✓ | ✗ | ✗ | ✗ | ✗ | ✗ | ✗ | * |
| | Interventions | ✗ | ✗ | ✗ | ✗ | ✓ | ✓ | ✗ | ✗ | ✗ | ✗ | ✗ | ✗ | ✗ | ✗ | * |
| | Length of stay (LoS) | ✗ | ✓ | ✓ | ✓ | ✗ | ✓ | ✗ | ✓ | ✗ | ✗ | ✗ | ✗ | ✓ | ✗ | ✓ |
| | Readmission§ | ✗ | ✗ | ✗ | ✓ | ✗ | ✗ | ✗ | ✗ | ✗ | ✗ | ✗ | ✓ | ✗ | ✗ | * |
| **Preproc.** | Feature engineering | ✗ | ✓ | ✓ | ✓ | ✓ | ✓ | ✗ | ✓ | ✓ | ✓ | ✓ | ✗ | ✓ | ✗ | ✓ |
| | Temporal imputation | ✗ | ✗ | ✗ | ✓ | ✓ | ✗ | ✗ | ✓ | ✓ | ✓ | ✓ | ✗ | ✓ | ✗ | ✓ |
| | Temporal resampling | ✗ | ✗ | ✗ | ✗ | ✗ | ✗ | ✗ | ✓ | ✓ | ✗ | ✓ | ✗ | ✗ | ✗ | ✓ |
| | Modular pipeline | ✗ | ✗ | ✗ | ✗ | ✓ | ✗ | ✗ | ✓ | ✗ | ✗ | ✓ | ✗ | ✓ | ✗ | ✓ |
| **Model architectures — ML** | LR | ✓ | ✓ | ✓ | ✓ | ✓ | ✗ | ✓ | ✓ | ✓ | ✗ | ✓ | ✗ | ✗ | ✗ | ✓ |
| | Random forest | ✗ | ✓ | ✗ | ✗ | ✓ | ✗ | ✗ | ✓ | ✗ | ✗ | ✓ | ✗ | ✗ | ✗ | ✓ |
| | Gradient boost | ✓ | ✓ | ✗ | ✗ | ✗ | ✗ | ✗ | ✗ | ✓ | ✓ | ✗ | ✗ | ✗ | ✗ | ✓ |
| **Model architectures — DL** | RNN | ✗ | ✓ | ✗ | ✓ | ✗ | ✓ | ✗ | ✗ | ✗ | ✗ | ✗ | ✓ | ✓ | ✗ | ✓ |
| | LSTM | ✗ | ✗ | ✓ | ✗ | ✓ | ✓ | ✓ | ✓ | ✓ | ✗ | ✗ | ✓ | ✓ | ✗ | ✓ |
| | GRU | ✗ | ✗ | ✗ | ✓ | ✓ | ✗ | ✗ | ✓ | ✗ | ✗ | ✗ | ✓ | ✓ | ✗ | ✓ |
| | Temporal CNN | ✗ | ✗ | ✗ | ✗ | ✓ | ✗ | ✓ | ✓ | ✓ | ✗ | ✗ | ✓ | ✓ | ✗ | ✓ |
| | Transformer | ✗ | ✗ | ✗ | ✓ | ✓ | ✗ | ✗ | ✓ | ✗ | ✗ | ✗ | ✓ | ✓ | ✗ | ✓ |
| Code available | | ✗ | ✓ | ✓ | ✓ | ✓ | ✓ | ✓ | ✓ | ✓ | ✓ | ✓ | ✓ | ✓ | ✓ | ✓ |
| Extensible† | | ✗ | ✗ | ✗ | ✗ | ✗ | ✗ | ✓ | ✗ | ✗ | ✗ | ✗ | ✗ | ✓ | ✗ | ✓ |
| Dataset interoperability‡ | | ✗ | ✗ | ✗ | ✗ | ✗ | ✗ | ✗ | ✗ | ✗ | ✗ | ✗ | ✗ | ✗ | ✗ | ✓ |

*: These tasks are not included by default but may be easily added through our cohort definition pipeline.
§: Due to lack of recorded database information, these tasks can only be defined for MIMIC III and IV.
†: Interface and extensive instructions to add interoperable modules following a provided abstraction (datasets, prediction tasks, models) and adjust existing modules without extensive rewriting or refactoring.
‡: Provides an uncoupled interoperable dataset definition, allowing a.o. transfer learning and domain adaption.

benchmarking, or an end-to-end pipeline. Finally, Yang et al. (2023) recently proposed PyHealth as a comprehensive deep learning toolkit for both ML researchers and healthcare practitioners; it is perhaps most closely related to our work. Unfortunately, PyHealth only supports subsets of the full datasets, and tasks must be defined anew for each dataset. It also does not currently include time series or ways to deal with missing data, limiting its use for novel clinical or ML developments.

## 3 BENCHMARK DESIGN

*YAIB* addresses the issues identified above and provides a unified interface to develop clinical prediction models for the ICU. An experiment in *YAIB* consists of four steps: **1)** define clinical concepts from the raw data; **2)** extract the patient cohort and specify the prediction task; **3)** preprocess the data and generate features; and **4)** train and evaluate the ML models (Figure 1).

### 3.1 DESIGN PHILOSOPHY

We strongly believe that medical research is inherently complex and that — rather than providing a rigid benchmark — there lies most value in providing a modular setup where the user can exchange

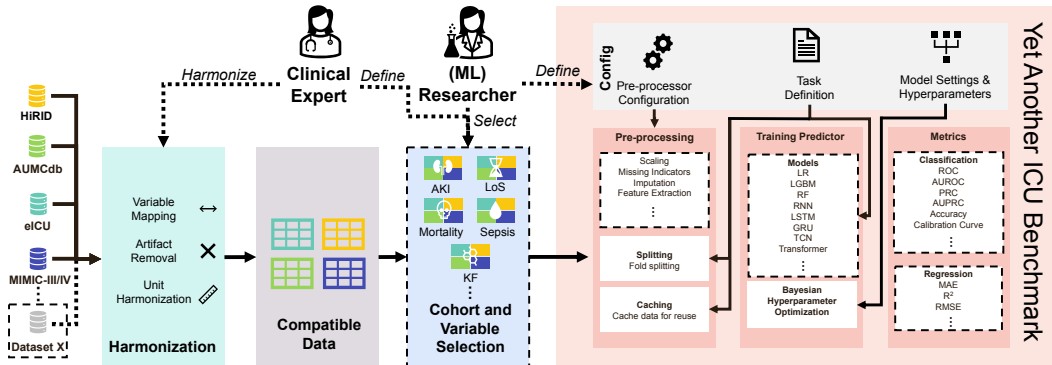

FIGURE 1: *Schematic overview of benchmark pipeline.* On the left side, the creation of harmonized ICU cohorts is shown. Note that the domain expertise of clinicians is often necessary for defining clinically useful tasks. The schematic overview of the benchmark stages can be found on the right. Note that the dotted line indicates that this component can be easily extended, as it follows an abstracted interface.

any part with something that better suits their needs and, importantly, do so reproducibly. For example, users frequently want to highlight a particular aspect of their model, prompting them to adapt the default tasks. Changes, however minor, can render results incomparable. We, therefore, prioritized extensibility across the entire experiment lifecycle. This high level of extensibility may increase the complexity of our benchmark. We mitigate this by providing a range of default experiments for users with limited access to medical expertise or who are content with a fixed set of medical tasks. The experiments were designed to be directly comparable and provide a common benchmark. This allows for a standardized evaluation of models similar to existing benchmarks but still benefits from out-of-the-box support for multiple datasets and easy adaptability if need be. While we did our best to ensure extensibility, *YAIB* cannot currently support all possible use cases. Specialized use cases like federated learning or reinforcement learning currently require custom code. However, we keep adding functionality to *YAIB*, and users may nevertheless benefit from using parts of our framework. We provide detailed documentation on how to implement any extensions (Appendix F). We strongly request users of YAIB to provide their code and a detailed list of the changes they have made to the repository to accurately and transparently provide results for their experiments.

## 3.2 CLINICAL CONCEPTS

We ensured that our benchmark supports existing and future ICU datasets. Working with multiple datasets requires careful data harmonization, as datasets are collected in different locations, with different clinical recording, and may have completely different data structures. We use the `ricu` (Bennett et al., 2023) R package to bring datasets into a common, semantically interoperable format. This harmonization relies on two things: **1)** a common temporal reference point and **2)** a dataset-independent definition of clinical concepts. `ricu` by default distinguishes measurements recorded for a patient, a hospital admission, or an ICU admission, and supports conversion between these levels of measurement. Through definition of reference points, it facilitates temporal comparability between datasets. `ricu` also allows defining clinical concepts such as heart rate or SOFA score independently of any particular dataset, specifying their meaning, plausible min/max ranges, and units of measurement. A concept can be enabled for a dataset by specifying how it should be extracted from the data, for example, by selecting an entire column or subsetting a table based on an item identifier. `ricu` thus acts as an interface to the raw data (stored in a fast, compressed column format), on command returning the data for a concept in a table of ID-time-value pairs. This is still no panacea to make ICU datasets immediately interoperable, but it provides a helpful framework for harmonization. For users unfamiliar with R, we provide an interface to access `ricu` concepts directly from Python. PYICU, a native Python implementation of `ricu`, is in development.

## 3.3 PATIENT COHORT AND TASK DEFINITION

Once in a common format, the same task definition can be applied across datasets. This facilitates code reuse and eliminates opportunities for error. Even so, care must be taken to combine clinical concepts, define meaningful prediction targets, and apply appropriate exclusion criteria. We provide default workflows and helper functions to support this process, including a transparent pipeline for applying exclusion criteria and reporting patient attrition. We supplied this functionality in a

TABLE 2: *Prediction task overview.* Note that the related work is non-exhaustive.

| No | Task | Frequency | Type | Related work |
|---|---|---|---|---|
| 1 | Mortality | Once per stay* | C | Baker et al. (2020); Lu et al. (2022); Medic et al. (2019); Sharma et al. (2017); Syed et al. (2021) |
| 2 | AKI | Hourly | C | Huang et al. (2021); Nikkinen et al. (2022); Pan et al. (2019); Rank et al. (2020); Shamout et al. (2021); Wang et al. (2020a); Koyner et al. (2018) |
| 3 | Sepsis | Hourly | C | Kok et al. (2020); Lauritsen et al. (2020); Merath et al. (2020); Fleuren et al. (2020b); Moor et al. (2021a; 2019); Muralitharan et al. (2021); Reyna et al. (2019); Shamout et al. (2021); Wang et al. (2022) |
| 4 | KF | Once per stay* | R | Tomašev et al. (2019); Futoma et al. (2016); Perotte et al. (2015); Cheng et al. (2018) |
| 5 | LoS | Hourly | R | Shillan et al. (2019); Guo et al. (2020) |

C: Classification, R: Regression, * Using data from 0-24 hours.

standalone repositoryto facilitate its use with other modeling frameworks such as Clairvoyance (Jarrett et al., 2021). The specification of our adaptive and re-definable pipeline is found in Appendix D.

### 3.4 PREPROCESSING AND FEATURE EXTRACTION

Further preprocessing is often required at runtime, including data normalization, generation of missingness indicators, and imputation. We provide a transparent, flexible way for users to define their preprocessing pipeline (also available as a standalone package), including default implementations of historical aggregation (e.g., mean or variance), resampling of the time resolution, imputation methods, and a wrapper for any Scikit-learn (Pedregosa et al., 2011) preprocessing step. Custom steps can be added by subtyping an abstracted step interface or providing a callable object to a generic step.

### 3.5 TRAINING AND EVALUATION

A single *YAIB* experiment creates and optimizes a model for a given task and preprocessing pipeline. Experiments are defined using the `gin-config` library (Dan Holtmann-Rice et al., 2018) in simple Python-like text files. The model configuration defines the model architecture and contains information on hyperparameters and optimizers. Every aspect of a model is fully configurable. The task configuration defines the target, the data source, the features, and the preprocessing. Additionally, one can define the cross-validation splits and the number of iterations. By defining the model and task separately, they can be mixed and matched, training the same architecture for multiple tasks or training multiple models for a single task. We provide details for adding new datasets, preprocessing, models, and an example of sepsis prediction in Appendix E. Training is supervised by PyTorch Lightning (Falcon & team, 2023), which uses standardized training and logging, GPU parallelism, and advanced debugging. Users can configure hyperparameter ranges and sampling methods for model optimization. A Gaussian Process is fit to the hyperparameters using `scikit-optimize` (Head et al., 2021) as a robust alternative to random search (Snoek et al., 2012).

**Result tracking** Results are automatically aggregated and written to a JSON file, in addition to optional Tensorboard (Abadi et al., 2016), PyTorch Lighting (Falcon & team, 2023), and WandB (Biewald, 2020) logging for easy experiment tracking. Performance evaluation records widely-used metrics out of the box (AUROC, AUPRC, calibration curve, accuracy, loss) and supports multiple evaluation libraries: TorchMetrics (Nicki Skafte Detlefsen et al., 2022), Pytorch-Ignite (Fomin et al., 2020), and Scikit-Learn (Pedregosa et al., 2011) metrics. New metrics, either developed by the user or from existing libraries, can be easily added (see Appendix F.6).

## 4 EXPERIMENTS

We ran experiments for five common prediction tasks: ICU mortality, onset of acute kidney injury (AKI), onset of sepsis, kidney function (KF) on day 2, and remaining length of stay (LoS) (Table 2). Mortality and KF used data from 0-24 hours. All other task used all available data until the event or discharge. We ensured adequate data quality by excluding: **1)** patients younger than 18 years; **2)** stays with missing discharge times; **3)** stays with less than six hours in the ICU; **4)** stays with measurements in less than four time bins; and **5)** stays with no measurement for more than 12 consecutive hours in the ICU. We also applied task-specific exclusion criteria. For example, we excluded stays of less than 30 hours for the ICU mortality task, as this could introduce causal leakage from patients already dead or about to die at the time of prediction. For each task, we included 52 features, of which 4 were static and 48 were time series. Various additional features, including prescriptions and diagnoses, can be directly used in *YAIB* by adjusting the cohort generation module (YAIB-cohorts); if features are not available, their implementation is straightforward (Appendix F). Information on the datasets, features, and individual cohort definitions can be found in Appendix C and D. The code to define these cohorts

is publicly available. In addition to the baseline performance for each task, dataset, and model, we used *YAIB* to investigate the effects of small variations in task definitions on predictive performance — a common obstacle to model comparability (Moor et al., 2021b; Fleuren et al., 2020b). Specifically, we **i)** only excluded stays of less than 24 hours to assess the effects of causal leakage by aligning our mortality task with Yèche et al. (2022), **ii)** omitted static and dynamic historical features (i.e., `min`, `max`, `count`, `mean`) to simulate access to fewer input data, and **iii)** compared alternative definitions for sepsis . We, additionally, evaluated transfer learning with the harmonized datasets (**iv**).

**Preprocessing 1. Scaling:** The data was scaled to zero mean and unit variance. **2. Imputation:** After adding missing indicators, we forward-filled all columns for the dynamic data, replacing missing values with the last known values of the same stay. Missing values without a prior measurement were filled with the sample mean. To prevent data leakage, we used the mean of the train split as the sample mean for all splits. **3. Feature generation:** We generated the `min`, `max`, `mean`, and `count` of measurements for each feature in the dynamic data. We only applied this step for the conventional ML models, e.g., Light Gradient Boosting Machine (LGBM), as they cannot capture sequential information natively.

## 4.1 MODELS AND EXPERIMENTAL SETUP

We considered a range of algorithms used in previous benchmarks (Table 1)and applied work (Hyland, 2020; Pirracchio et al., 2015; Silva et al., 2012; Syed et al., 2021), including regularized logistic regression (LR) and elastic net (EN) (used for classification and regression, respectively (Pedregosa et al., 2011)), LGBM (Ke et al., 2017), and four variations of neural networks: Gated Recurrent Unit (GRU) (Cho et al., 2014), Long Short-Term Memory (LSTM) (Hochreiter & Schmidhuber, 1997), Temporal Convolutional Network (TCN) (Bai et al., 2018) and transformer (TF) (Vaswani et al., 2017). LR, EN, and LGBM were used with the feature generation described above, as they are unable to utilize time series. The implementation of neural networks was adapted from Yèche et al. (2022).

For our experiments, unless stated otherwise, we used 5 iterations of 5-fold cross-validation. Hyperparameters were tuned on the training set using 30/50 (DL/ML, respectively) iterations of Bayesian hyperparameter optimization (Snoek et al., 2012). For computational reasons, hyperparameter tuning used only the first 2/3 folds, respectively (see Appendix H for a definition of all searched and selected hyperparameters). The final validation of the best hyperparameters used all 5 folds. Each model was optimized for a maximum of 1000 epochs. Training was stopped early if performance on the validation set did not improve for 10 epochs. The epoch with the best performance on the validation set was retained and evaluated on the test set. This process was repeated for 5 iterations, after which the results were averaged, and the standard deviation was calculated.

## 4.2 BENCHMARKING BASELINE MODELS ON MAJOR ICU DATASETS

Baseline results for all tasks can be found in Table 3 and 4. Note that we have also benchmarked our tasks for two openly available demo datasets from MIMIC-III and eICU; these can be directly accessed without completing a credentialing procedure (see Table 11 and 12).

**ICU mortality** The performance of traditional ML and DL models was highly comparable among each other and across datasets when predicting mortality based on data from the first 24 hours. Notably, AUPRC was higher in AUMCdb due to a higher outcome prevalence (Table 13).

**Acute kidney injury (AKI)** Maximum achievable performance was also similar across datasets when predicting the hourly onset of AKI, with the notable exception of HiRID, which had both lower AUROC and AUPRC for all models. GRU models consistently achieved the best performance.

**Sepsis** The performance of baseline models was worst for the hourly onset of sepsis, both for AUROC and especially AUPRC. This may be explained by the particularly low prevalence of $\sim 1\%$ hourly bins classified as septic and the relative difficulty of predicting sepsis in general (Moor et al., 2021b).

**Kidney function (KF)** Classical ML models achieved relatively good performance for this task, which may reflect the dependence of KF on a limited number of features (Grinsztajn et al., 2022).

**Remaining length of stay (LoS)** The performance of ML and DL models was also comparable across datasets. Nevertheless, predicting the length of stay seems difficult, given that the average MAE is almost two days. Transformers consistently outperformed most other model types.

TABLE 3: *Baseline performance on the classification tasks.* We **embolden** the best mean AUROC × 100 (↑, i.e., higher is better) and AUPRC × 100 (↑) per dataset and those within a standard deviation (±).

| | AUMCdb | | HiRID | | eICU | | MIMIC-IV | |
|---|---|---|---|---|---|---|---|---|
| **Algorithm** | AUROC | AUPRC | AUROC | AUPRC | AUROC | AUPRC | AUROC | AUPRC |
| **Mortality** | | | | | | | | |
| LR | 83.7±0.6 | 52.9±1.2 | 84.0±0.3 | 36.9±1.1 | 84.8±0.2 | 33.0±0.7 | 86.1±0.1 | 39.7±0.6 |
| LGBM | **84.5±0.5** | **53.7±1.2** | 84.4±0.3 | **40.6±0.8** | **85.7±0.2** | **36.0±0.6** | **87.7±0.2** | **44.2±0.7** |
| GRU | 83.9±0.3 | **53.8±0.7** | **84.8±0.2** | 39.4±0.4 | **86.0±0.1** | 35.6±0.1 | **87.6±0.1** | 42.8±0.3 |
| LSTM | 83.7±0.7 | **53.6±1.4** | 84.0±0.7 | 37.8±1.0 | 85.5±0.2 | **35.7±0.8** | 86.7±0.4 | 41.0±0.7 |
| TCN | **84.0±0.6** | **54.2±1.4** | **84.6±0.7** | 39.2±1.3 | 85.4±0.2 | 34.3±0.6 | 87.1±0.3 | 41.4±0.8 |
| TF | 84.1±0.2 | **54.4±1.1** | **84.9±0.7** | 39.3±1.5 | **85.9±0.2** | 34.7±0.8 | 86.9±0.3 | 42.2±0.3 |
| **AKI** | | | | | | | | |
| LR | 85.5±0.3 | 45.1±0.4 | 79.6±0.1 | 31.8±0.8 | 72.8±0.1 | 32.2±0.2 | 77.1±0.2 | 37.7±0.3 |
| LGBM | 85.8±0.3 | 48.4±0.6 | 80.2±0.2 | 32.8±0.4 | 84.6±0.1 | 50.8±0.2 | 83.8±0.1 | 53.3±0.2 |
| GRU | **90.6±0.3** | **52.8±0.7** | **82.2±0.2** | 33.9±0.4 | **90.9±0.0** | **72.2±0.1** | **90.7±0.1** | **69.6±0.2** |
| LSTM | 86.5±0.4 | 40.6±0.6 | 81.0±0.4 | 31.8±0.4 | 90.2±0.1 | 69.9±0.2 | 89.7±0.1 | 66.5±0.2 |
| TCN | 89.6±0.2 | 50.0±0.9 | 81.2±0.2 | 32.3±0.4 | 90.4±0.0 | 70.4±0.2 | 89.8±0.1 | 66.8±0.2 |
| TF | 88.2±0.2 | 48.2±0.7 | 81.5±0.2 | 33.4±0.5 | 89.9±0.1 | 68.0±0.3 | 89.6±0.1 | 65.6±0.2 |
| **Sepsis** | | | | | | | | |
| LR | 74.7±1.0 | 4.0±0.4 | 76.5±0.6 | 8.4±0.3 | 71.8±0.3 | 2.9±0.1 | 77.1±0.4 | 4.6±0.1 |
| LGBM | 74.0±0.8 | 5.2±0.7 | 76.1±0.4 | 10.4±0.5 | 69.1±0.3 | 3.3±0.1 | 77.5±0.3 | 5.9±0.2 |
| GRU | 79.7±0.9 | 7.7±0.7 | **80.6±0.5** | **12.6±0.5** | **77.4±0.2** | **5.1±0.1** | **83.6±0.3** | **9.1±0.3** |
| LSTM | 77.1±0.8 | 6.4±0.5 | 78.8±0.4 | 11.1±0.5 | 74.0±0.2 | 4.0±0.1 | 82.0±0.3 | 8.0±0.2 |
| TCN | 78.7±0.7 | 7.1±0.6 | **80.8±0.5** | **13.0±0.4** | 76.7±0.1 | 4.9±0.1 | 82.7±0.3 | **8.8±0.2** |
| TF | **80.7±0.9** | **8.6±0.8** | **80.8±0.3** | **12.6±0.6** | 76.2±0.1 | 4.6±0.1 | 80.0±0.8 | 6.6±0.2 |

TABLE 4: *Baseline performance on the regression tasks.* Results are reported in Mean Absolute Error (↓)

| | Kidney function *in mg/dL* | | | | Length of Stay *in hours* | | | |
|---|---|---|---|---|---|---|---|---|
| **Algo.** | **AUMCdb** | **HiRID** | **eICU** | **MIMIC-IV** | **AUMCdb** | **HiRID** | **eICU** | **MIMIC-IV** |
| EN | **0.24±0.00** | 0.28±0.00 | 0.31±0.00 | 0.25±0.00 | 54.9±0.0 | 47.2±0.1 | 43.6±0.0 | 46.5±0.0 |
| LGBM | 0.32±0.00 | 0.34±0.00 | **0.29±0.00** | **0.24±0.00** | 44.7±0.0 | **39.2±0.1** | 39.3±0.0 | 40.1±0.0 |
| GRU | 0.29±0.00 | 0.32±0.01 | 0.34±0.01 | 0.30±0.01 | 42.9±0.1 | 39.6±0.1 | 38.9±0.1 | 39.9±0.1 |
| LSTM | 0.29±0.00 | 0.33±0.00 | **0.28±0.01** | 0.28±0.01 | 44.8±0.1 | 39.8±0.1 | 39.2±0.1 | 40.6±0.1 |
| TCN | 0.28±0.01 | **0.23±0.01** | 0.31±0.00 | 0.28±0.01 | 43.7±0.1 | 39.9±0.1 | 38.9±0.0 | 40.4±0.1 |
| TF | 0.26±0.00 | 0.31±0.01 | 0.33±0.01 | 0.32±0.01 | **41.8±0.1** | **39.1±0.1** | **38.2±0.1** | **39.0±0.1** |

We provide the average and Interquartile range for Kidney Function and Length of Stay in Table 14.

## 4.3 USING YAIB AS AN EXPERIMENTAL ML FRAMEWORK

**Changing exclusion criteria for mortality cohorts** As hypothesized, the choice of exclusion criteria could majorly impact achievable prediction performance (Table 5). Compared to the peak performance achieved with the HiRID-benchmark (Yèche et al., 2022), our baseline performance for the mortality task was noticeably lower. Aligning the exclusion criteria accounted for half of the performance difference. The remaining difference was likely due to the inclusion of additional predictors — most notably drug usage — in the HiRID-benchmark. This highlights the difficulties of comparing works that ostensibly address the same task, even using the same dataset and model implementation.

**Restricting input features** We observed that dynamic feature generation consistently outperformed task definitions that did not include them (Table 7 and 8). LR on MIMIC-IV showed a considerable performance gap, whereas AUMCdb remained stable. We noted a performance decrease that ranges between 4.0% and 19.1% for LR and between 5.2% and 13.1% for LGBM. Omitting static features led to minor drops in performance (Table 9 and 10); averaged across datasets, we observe a performance differences ranging between 0.5% and 0.2% for the transformer model.

**Comparing sepsis definitions** Label definitions also had a considerable impact on AUROC and/or AUPRC (Table 6), which was not always apparent from the definition alone. Sepsis has been defined in several ways (Fleuren et al., 2020b), mainly because a clinical gold standard that can be transferred

TABLE 5: *ICU mortality prediction on HiRID with (>24h) and without (>30h) possibility of causal leakage.*

| | Cohort definition | | | | | |
| | w/o leakage | | w/ leakage | | Yèche et al. (2022) | |
| Algorithm | AUROC | AUPRC | AUROC | AUPRC | AUROC | AUPRC |
|---|---|---|---|---|---|---|
| LR | 84.0±0.3 | 36.9±1.1 | 87.2±0.4 | 43.1±1.3 | 89.0±0.0 | 58.1±0.0 |
| LGBM | **84.5±0.3** | 40.6±0.9 | **87.9±0.5** | **47.7±1.2** | 88.8±0.2 | 54.6±0.8 |
| GRU | **84.8±0.2** | 39.4±0.4 | **88.2±0.3** | 46.1±1.2 | 90.0±0.4 | **60.3±1.6** |
| TCN | 84.6±0.7 | 39.2±1.3 | 87.8±0.2 | 45.2±1.0 | 89.7±0.4 | **60.2±1.1** |
| TF | **84.9±0.7** | 39.4±1.5 | **88.2±0.3** | **47.1±1.2** | **90.8±0.2** | 61.0±0.8 |

TABLE 6: *Sepsis prediction on MIMIC-IV for different definitions of sepsis.*

| | Sepsis definition | | | | | |
| | Seymour et al. (2016)* | | Moor et al. (2021a) | | Calvert et al. (2016) | |
| Algorithm | AUROC | AUPRC | AUROC | AUPRC | AUROC | AUPRC |
|---|---|---|---|---|---|---|
| LGBM | 75.9±0.2 | 4.3±0.0 | 72.4±0.0 | 10.5±0.0 | 62.2±0.2 | 1.8±0.0 |
| GRU | **79.2±0.1** | **6.1±0.0** | **80.9±0.0** | **17.7±0.0** | **89.2±0.0** | **9.3±0.2** |

* Our definition; adapted to be more clinically actionable, see Appendix D.

to ML models is currently lacking. Our sepsis definition (adapted from Seymour et al. (2016), see Appendix D) can be considered closely related to that used by Moor et al. (2021a), who implement a variant of Sepsis-3 (Singer et al., 2016). However, we required that antibiotics were administered continuously for ≥ 3 days (Reyna et al., 2019). We judged that this would increase the clinical usability of the task but found that it also severely reduced the achievable AUPRC — likely due to a much lower prevalence (Table 17). The definition used by Calvert et al. (2016) on the other hand adapted Sepsis-2 (Levy et al., 2003), which differs fundamentally from Sepsis-3 and resulted in a notably higher AUROC (Engoren et al., 2020). This highlights the importance of precise cohort definitions, as some definitions may, by design, be more difficult to predict.

## 4.4 TRANSFER LEARNING

**External validation** *YAIB*'s common dataset format allowed us to evaluate a model trained on an equal sample of one dataset on data from all other datasets. We additionally trained a model on pooled (d-1) data from three datasets and evaluated on the fourth, held-out dataset. For the ICU mortality task (Figure 2), models, as expected, performed best on independent test data from their training dataset (diagonal). Performance could drop considerably when models were evaluated in another database (off-diagonal). Notably, AUPRC performance could increase in the evaluation dataset (rows) but always remained lower than the highest achievable performance for that dataset (columns). We found that MIMIC-IV and eICU transferred well among each other. The pooled model usually performed as well as the best single-dataset model. Notably, AUMCdb AUPRC results demonstrate decidedly

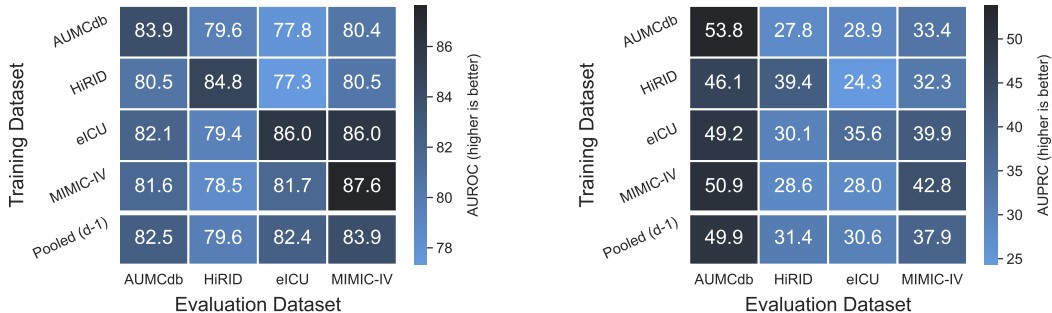

FIGURE 2: *Performance of prediction models when trained on one dataset (row) and evaluated on all others (columns).* **Left**: Performance in AUROC of the GRU model on *ICU mortality*. **Right**: Performance in AUPRC for the same models. Pooled (d-1) refers to training a model on every dataset except the evaluation dataset.

higher performance than evaluation on other datasets, which could be the result of a patient case mix and outcome prevalence (see Table 14).

**Fine-tuning** In Figure 2, we saw that eICU resulted in the most generalizable model for ICU mortality, which may serve as a strong pre-training for transfer learning. Since it worked worst for HiRID, we further fine-tuned the eICU GRU model (source) for HiRID (target) by retraining it using an increasing number of samples from the HiRID dataset. We compared the results to a model trained from scratch on the same amount of HiRID samples (Figure 3). Fine-tuning was profitable for any number of additional samples and especially for <4,000 samples.

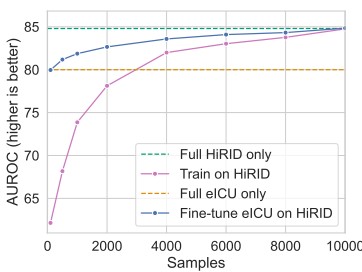

FIGURE 3: *Fine-tuning an eICU model for ICU mortality prediction on HiRID.*

## 5 DISCUSSION

We provide extensive ML and DL baselines for five clinical prediction tasks trained across four major open-source ICU datasets. While we frequently obtained comparable results across model architectures, seemingly small differences in cohort definition could substantially impact the achieved accuracy. Our findings highlight not only the need for standardized training pipelines but also for harmonized cohort definitions to allow for a meaningful comparison of clinical prediction models. Our work provides the first international, multi-center ICU benchmark, including the first-ever benchmark for the AmsterdamUMCdb dataset. It naturally facilitates sorely needed external validation of model performances and allows fine-tuning of pre-trained models for new datasets. This makes *YAIB* relevant to a wide range of research areas beyond classical supervised learning, including domain adaption and generalization. We hope this broad reach encourages ICU data providers to ensure compatibility with *YAIB*, as they can expect a larger overall research impact. This simplifies the use of novel datasets by the clinical and ML community.

*YAIB* aids researchers in training baseline models by providing them with ready-to-use implementations of state-of-the-art model architectures; new model implementations can therefore be easily compared. While most existing benchmarking studies are hard-coded, we utilize flexible, *dataset-independent* cohort definitions and configurable preprocessing facilities linked via a common, shareable syntax. This setup acknowledges that task definitions inevitably involve arbitrary decisions, without one "size" that fits all. In our work, we embrace this idea and aim to equip researchers — both applied and theoretical — with the tools to quickly adapt a task to their individual needs (including the use of custom proprietary data) while maintaining reproducibility and reusability across studies. Models can thus be compared across multiple, slightly different task definitions and datasets, still ensuring an apples-to-apples comparison. We hope this lowers the bar for researchers to test their approaches across a range of configurations and datasets.

*YAIB* is currently limited to ICU settings, where several datasets are publicly available. A similar setup could be beneficial for data from other medical settings, such as inpatient wards. Although created for critical care, *YAIB* is not specific to the ICU and can be readily extended to other settings, provided a suitable configuration is defined. Features included in *YAIB*, at the time of writing, mainly relate to vital signs, lab tests, and data relevant to outcome definitions. Further clinician-assisted harmonization efforts will be necessary to increase the breadth of features, most notably medications and comorbidities. If *YAIB* is adapted to general EHR, including clinical notes and medical imaging is a logical next step. We also note that we compared these models on the basis of commonly used ML metrics; we leave the comparison with respect to clinical fairness and bias as an easy future extension to our framework (see Appendix F). Finally, we advise users of our benchmark to carefully consider the compromises made to allow for cohort harmonization; we strongly recommend clinical validation before making practical decisions based on the developed models.

## 6 CONCLUSION

Routine medical data is highly complex. Without clear ground truth, researchers are inevitably forced to make arbitrary design choices when defining outcomes and populations of interest. To promote comparable and reproducible models in this setting, we believe that further tools are needed that allow researchers to define clinical prediction tasks transparently, share experimental setups easily, and validate results against various data sources. As a flexible and extensible framework for clinical modeling on ICU data, *YAIB* is meant to be a step towards that goal.

## 7 ACKNOWLEDGEMENTS

Robin van de Water is funded by the "Gemeinsamer Bundesausschuss (G-BA) Innovationsausschuss" in the framework of "CASSANDRA - Clinical ASSist AND aleRt Algorithms" (project number 01VSF20015). We would like to acknowledge the work of Alisher Turubayev, Anna Shopova, Fabian Lange, Mahmut Kamalak, Paul Mattes, and Victoria Ayvasky for adding Pytorch Lightning, Weights and Biases compatibility, and several optional imputation methods to a later version of the benchmark repository.

## 8 ETHICS STATEMENT

We do not manage access and do not provide access to any of the full medical datasets included in this work, and we adhere to the usage licenses for each dataset. Users can follow the credentialing procedures outlined in Appendix C. However, we provide two preprocessed demo datasets out of the box for reproducibility and experimentation. The demo task cohorts for MIMIC-III and eICU mentioned in that section are derived from the official demo datasets published on PhysioNet by the original authors of the respective databases. Each demo dataset represents a small, curated subset of data that is freely accessible without any need for human subject training. Both demo datasets are published under an Open Data Commons Open Database License v1.0, which explicitly permits the adoption and sharing of the data. The original demo data, as well as further information, can be found at the MIMIC-III demo and eICU demo Physionet pages.

## 9 REPRODUCIBILITY STATEMENT

We include the source code of YAIB[1] (main benchmark), YAIB-cohorts[2] (adaptable cohort extraction) and ReciPys[3] (extensible preprocessing package) in our submission. Models for each task and architecture are publicly available[4]. In the included source code, a file called PAPER.md[5] describes the reproducibility steps of the experiments in this paper. Specifically, one requires the standalone codebase of YAIB-cohorts to first create the cohorts from the acquired data, once you have completed the required credentialing (see Appendix C for details). As mentioned, we include demo cohort data for each task (results for these cohorts are shown in Appendix B). Appendix D describes the data processing and task creation. The usage of *YAIB* is detailed in Appendix E. Appendix F shows how *YAIB* can be extended with new datasets, clinical concepts, tasks, models, and evaluation metrics. Additionally, we refer to the README.md[6] and the wiki[7] for the usage of *YAIB*. Appendix G and H detail the experiment design and chosen hyperparameters, respectively. Finally, Appendix I contains the machine learning reproducibility checklist for our work.

---

[1] https://github.com/rvandewater/YAIB
[2] https://github.com/rvandewater/YAIB-cohorts
[3] https://github.com/rvandewater/ReciPys
[4] https://github.com/rvandewater/YAIB-models
[5] https://github.com/rvandewater/YAIB/blob/master/PAPER.md
[6] https://github.com/rvandewater/YAIB/blob/master/README.md
[7] https://github.com/rvandewater/YAIB/wiki/YAIB-wiki-home

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

APPENDICES: TABLE OF CONTENTS

## A APPENDIX: YAIB'S CONTRIBUTION IN CONTEXT

This Appendix provides an extensive description for the positioning of *YAIB* in the contemporary clinical ML research landscape. We particularly recommend it to those that are looking into creating their own solutions for clinical ML.

### A.1 EXTENSIBILITY AND REPRODUCIBILITY

We designed *YAIB* to be as extensible as possible while retaining full reproducibility. This means easy support of new databases, clinical concepts, tasks, experiment configurations, preprocessing pipelines, imputation methods, models, and evaluation metrics. If changes are necessary, they need to be reproducible and easily shareable across research teams. If the user only requires a few default ICU tasks from a single i.i.d. dataset to test their new method, any existing ICU benchmarks could be sufficient. Users do not need to apply for access to multiple datasets and do not have to deal with the intricacies of the clinical task definition. As long as the integration of a new model is seamless, such simple frameworks are fit-for-purpose and abstract much of the complexity, allowing the user to only worry about one thing: their model. If multiple papers used the exact same benchmark, results are also directly comparable between papers (an "apples-to-apples" comparison).

However, we found that this setup tends to be too restrictive and thus unrealistic. Users often want to highlight a particular aspect of their model, prompting them to adapt to the default task. At other times, they want to show clinical impact and need to adapt the default task to make it more realistic. Given the lack of successful translation of prediction models into clinical practice, reviewers are also increasingly requesting external validation – sometimes with multiple endpoints – which is difficult to shoehorn into most existing solutions. *YAIB* embraces the need to tweak experimental setups. Results will no longer be directly comparable between papers, but we argue that true apples-to-apples comparisons were inherently rare. Instead of forcing users into a rigid framework, it allows for adaptations but requires them to be done in a transparent manner. Absolute performance should be compared only within the same paper or among papers with the same task setup (see our examples in Tables 5 and 6).

To facilitate the transparency of adaptations, we rely on a sophisticated framework to define clinical concepts across multiple datasets (ricu). We have adapted and extended ricu to provide a standard workflow for *YAIB* to integrate new databases and define new clinical concepts. To date, it has been successfully used to bring 4/5 +1 ICU datasets into a common format (including our addition of the Salzburg Intensive Care Database, which is currently in quality control). This approach is flexible enough that we have not yet encountered significant restrictions in mapping admissions, demographics, vital signs, laboratory values, medication (including rates and durations), clinical scores, and outcomes at different time scales across datasets. The main restriction of ricu is that it is currently implemented in the R language only, but we provide guidance on how to access it via rpy2, and we are in the process of porting it to Python; this will make our pipeline even more accessible, especially to clinical researchers. Our cohort definition functionality provides helper functions to apply inclusion/exclusion criteria on top of ricu and report step-by-step attrition numbers. The cohorts can be used in a modular fashion with custom preprocessing steps, imputations, prediction models, and evaluation metrics, all using the exact same code across multiple datasets.

Even so, there will likely be situations where the user may be better off with a custom solution. We expect this to occur once their use case diverges significantly from standard supervised learning. For example, federated learning or reinforcement learning setups may require significantly different training and evaluation loops. These are not currently supported, but we consider this as future work. In any case, the user can still use our data processing, cohort generation, and possibly other parts of *YAIB* (e.g., by exchanging the default training module with a custom module). Authors using *YAIB* should, therefore, provide their code and a detailed list of the changes they have made to the repository; modern version control allows us to verify this against the original *YAIB* repository easily.

The *YAIB* pipeline has helped us to produce reproducible results quickly and provides the required extensibility for our purposes. We are in touch with some researchers who have used *YAIB* to date and provided feedback, although mainly in an informal way. We refer to van de Water et al. (2023) as an example of the usability of *YAIB*. This work used YAIB as a bedrock for implementing imputation methods and are in the process of extending this to more methods and downstream tasks. For concrete examples and guidance for how to extend *YAIB* , we refer to Appendix D and the wiki documentation.

## A.2 THE CHOICE OF FEATURES

We chose the 52 most common clinical features shared by all datasets. They were readily available in all benchmarked datasets, demonstrating *YAIB*'s adaptability. This is done because our work focuses on the interoperability of datasets and the opportunity for experiments with a.o. transfer learning and domain adaption. We believe there is the most value in providing a modular setup where the user can add or remove features to suit their needs better and, most importantly, do so reproducibly.

Nevertheless, several medications for eICU and MIMIC-IV are readily available; the ricu package maintains a full list of the currently available native concepts which are available[8]. Complex concepts, dependent on several native concepts, such as SOFA scores, are additionally available. Each concept that is available in ricu can be readily used in *YAIB*. Some medications that are already implemented, such as antibiotics and vasopressors, are used in the definition of the complex Sepsis endpoint. Therefore, we decided to leave those out to have the same features for each task.

We note, additionally, that it is straightforward to implement new concepts in our pipeline; Appendix E.2 describes the addition of Potassium Chloride to the ICU harmonization package ricu. A similar process can be followed for adding new medications, which immediately improves the usability of *YAIB*. Moreover, we are actively working on integrating more features, including comorbidities and medications. We would like to note that many features are not available across all datasets; this does not mean they can not be valuable in clinical prediction tasks.

Finally, we would like to point out that *YAIB*'s end-to-end pipeline is designed as a solid starting point for 1) clinicians looking for external validation to employ ML in practice, 2) dataset creators looking for a solid platform to facilitate widespread use, and 3) the ML community to contribute novel prediction models. They can use a mature and externally developed framework, which adds to the credibility of any experiment results. Adding new feature concepts for their datasets can also increase the adoption of their datasets. They are likely domain experts for their respective datasets, meaning fewer errors are made in this process. This process will improve the usability of *YAIB* as an end-to-end benchmarking tool and improve the confidence of health experts in clinical ML.

## A.3 USING YAIB IN NOVEL SCIENTIFIC WORK

We acknowledge the importance of reproducible ML experiments. In this section, we describe how future work can transparently use *YAIB* as a platform for comparing their contributions. The authors ideally provide one or more open GitHub repositories so it is straightforward to check versioning; this includes:

---

[8]`https://github.com/eth-mds/ricu/blob/main/inst/extdata/config/concept-dict.json`

1. The concept dictionary in JSON format if they add new concepts. The main repository contains the current version of the concept dictionary of the vanilla `ricu`[9].

2. The repository that is used to generate cohorts if they introduce a new task. Ideally, this is forked from the *YAIB*-cohorts repository.

3. The `preprocessing.py` file in case this has been changed.

4. The `model.py` and `dataset.py` file that contains the definition for the model and dataset and dataloader (if adjusted).

5. `model.gin` file that specificies the used hyperparameters and hyperparameter ranges.

6. `wandb.yml` if Weights and Biases is used for running experiments with this model.

7. Provide versions of `ricu`, `YAIB-cohorts`, and `YAIB` they have used as a base.

If authors cover these aspects when presenting new work; one can easily reproduce their experiments even though they might not have used a "vanilla" implementation of YAIB. An additional benefit of providing these materials is that authors of future work can hereby participate in making YAIB more comprehensive.

## A.4 EXTENDED RELATED WORK

Comparison to existing frameworks We thank the reviewer for bringing up the preprint of TemporAI, which is still in early development at the time of writing. While we included an earlier work by the same group, Clairvoyance, in our related work, we have now updated the manuscript by adding this work in the related work section and to Table 1. We note that Pyhealth is already included in the related work section of the original manuscript. However, we elaborate on the differences between *YAIB* and both works below.

### A.4.1 CLAIRVOYANCE

Clairvoyance (Jarrett et al., 2021) is "a Unified, End-to-End AutoML Pipeline for Medical Time Series". As such, it does not focus on ICUs or benchmarking but instead standardizes model learning (imputation and training), model evaluation, and model selection, focusing on the computational aspects of developing a model. Clairvoyance comes with some code to define a task for treatment effects estimation on MIMIC III data. However, this task is hard coded and lightly documented, primarily serving as a demo of Clairvoyance. The exemplary nature of this task is further exemplified by the fact that, at no point the authors mention possible confounders/colliders of the treatment effect and whether they are conceivably adjusted for by the covariates, rendering any causal interpretation moot. It is unclear how this task can be easily adapted or extended to other databases without significant amounts of custom code.

**Advantages of *YAIB* compared to Clairvoyance:** *YAIB* puts ICU data and tasks front and center. *YAIB* supports the whole workflow, from raw data to clinical concepts to well-defined cohorts. This approach greatly facilitates the transparent and reproducible preprocessing of (often messy) ICU data, which Clairvoyance does not cover. We strongly believe that unless tasks can be adapted easily and reproducibly, it will lead to inevitable ad-hoc adaptations of the task that often end up irreproducible. *YAIB*, therefore, improves on existing modeling frameworks by putting an equal emphasis on standardized data processing for meaningful model development.

### A.4.2 TEMPORAI

TemporAI (Saveliev & van der Schaar, 2023) is a package that is currently in early development without a peer-reviewed publication associated with it. While it promises to provide: "prediction, causal inference, and time-to-event analysis, as well as common preprocessing utilities and model interpretability methods," it is unclear from current documentation how to use established datasets with this package or how to use relevant medical prediction tasks.

**The advantages of *YAIB* compared to TemporAI** are similar to those between *YAIB* and Clairvoyance: *YAIB* puts ICU data and tasks front and center for both ML scientists and clinicians.

---

[9] `https://github.com/eth-mds/ricu/blob/main/inst/extdata/config/concept-dict.json`

*YAIB* supports the whole workflow, from raw data to clinical concepts to well-defined cohorts. This approach greatly facilitates the transparent and reproducible preprocessing of (often messy) ICU data, which TemporAI, similarly to Clairvoyance, does not cover. However, we would like to note that using TemporAI (or Clairvoyance) with the *YAIB* pipeline to create a different end-to-end pipeline is possible as it allows for "swapping out" components. We provide the functionality in our `YAIB-cohorts` repository to convert any cohort to a format compatible with Clairvoyance and TemporAI.

### A.4.3 PyHealth

PyHealth (Yang et al., 2023) is "a comprehensive deep learning toolkit designed for both ML researchers and healthcare practitioners." PyHealth aims to support all EHR databases. It is thus similar in scope to our proposed framework. Unfortunately, upon closer inspection, PyHealth only supports a small subset of the information in MIMIC and eICU. While diagnoses and prescriptions are, in theory, included, they are processed as a simple bag of diagnosis codes or drug codes without information on strength/duration or semantic interpretation of what they represent (e.g., what is a vasopressor needed in calculating the SOFA score). Vital signs are not supported at all, presumably because PyHealth reads information from raw .csv files and may struggle to process large quantities of vital sign data. PyHealth further states that the datasets are independent of task definitions. This, unfortunately, appears to mean that they have to be implemented anew for each database, with custom dataset-specific code for the same task. Furthermore, all currently available ICU tasks in PyHealth use static data only and do not include any time series.

**Advantages of *YAIB* compared to PyHealth:** *YAIB* supports all databases within a common, principled interface (see the response on data harmonization above). Moreover, *YAIB* enables a single task definition that one can directly use for any included dataset. As far as they can work with time series data, *YAIB* can incorporate any model defined in PyHealth.

## B Appendix: Extended Results

This Appendix contains results that were left out of the main text.

TABLE 7: *Comparing the use of dynamic feature generation (FG) to the baseline of ICU mortality prediction, AUROC (↑). Note that an otherwise identical experiment setup was used to obtain results for the "without feature generation" results.*

| Preprocessing | AUMCdb | | HiRID | | eICU | | MIMIC-IV | |
|---|---|---|---|---|---|---|---|---|
| | w/ FG | w/o FG | w/ FG | w/o FG | w/ FG | w/o FG | w/ FG | w/o FG |
| LR | 83.7±0.6 | 82.2±0.5 | 84.0±0.3 | 81.8±0.7 | 84.8±0.2 | 81.1±0.1 | 86.1±0.1 | 80.2±0.3 |
| LGBM | **84.5±0.6** | **83.5±0.4** | **84.5±0.3** | **82.5±0.6** | **85.7±0.2** | **83.5±0.3** | **87.7±0.2** | **85.9±0.1** |

TABLE 8: *Comparing the use of dynamic feature generation (FG) to the baseline of ICU mortality prediction, AUPRC (↑). Note that an otherwise identical experiment setup was used to obtain results for the "without feature generation" results.*

| Preprocessing | AUMCdb | | HiRID | | eICU | | MIMIC-IV | |
|---|---|---|---|---|---|---|---|---|
| | w/ FG | w/o FG | w/ FG | w/o FG | w/ FG | w/o FG | w/ FG | w/o FG |
| LR | **52.9±1.2** | **50.8±1.2** | 36.9±1.1 | 33.1±0.8 | 33.0±0.7 | 28.3±0.4 | 39.7±0.6 | 32.1±0.6 |
| LGBM | **53.7±1.2** | **50.9±1.0** | **40.6±0.8** | **35.3±0.9** | **36.0±0.6** | **32.5±0.9** | **44.2±0.7** | **40.1±0.7** |

**Feature generation** We compare the use of feature generation for classical ML models. The RECIPYS package provides the functionality of assembling different preprocessing steps to be supplied by the user (i.e., a recipe). We show the results in Table 7 and 8.

**The impact of static features** We leveraged this customizable preprocessing to perform training prediction models without static features (i.e., age, sex, height, and weight). The AUROC results are

TABLE 9: *AUROC (↑) performance comparison of including static data for ICU mortality prediction.*

| | AUMCdb | | HiRID | | eICU | | MIMIC-IV | |
|---|---|---|---|---|---|---|---|---|
| **Inclusion** | w/ static | w/o static | w/ static | w/o static | w/ static | w/o static | w/ static | w/o static |
| LR | 83.7±0.6 | 82.9±0.6 | 84.0±0.3 | 82.8±0.7 | 84.8±0.2 | 84.3±0.2 | 86.1±0.1 | 84.3±0.3 |
| LGBM | **84.5±0.6** | 83.4±0.4 | **84.4±0.3** | 83.9±0.7 | **85.7±0.2** | 84.7±0.2 | 84.7±0.2 | 86.2±0.2 |
| GRU | 83.7±0.7 | 83.4±0.6 | **84.3±0.7** | 84.0±0.8 | **85.9±0.2** | 85.7±0.2 | **87.4±0.2** | 86.9±0.3 |
| LSTM | 83.7±0.7 | 82.9±0.6 | 84.0±0.7 | 83.4±0.7 | 85.5±0.2 | 85.1±0.2 | 86.7±0.4 | 86.1±0.3 |
| TCN | **84.0±0.6** | 83.5±0.6 | **84.6±0.7** | 83.9±0.8 | 85.4±0.2 | 86.4±0.3 | 87.1±0.3 | 86.4±0.3 |
| TF | **84.1±0.2** | 83.7±0.4 | **84.9±0.7** | 84.4±0.7 | **85.9±0.2** | 85.7±0.2 | 86.9±0.3 | 86.5±0.3 |

TABLE 10: *AUPRC (↑) performance comparison of including static data for ICU Mortality Prediction.*

| | AUMCdb | | HiRID | | eICU | | MIMIC-IV | |
|---|---|---|---|---|---|---|---|---|
| **Inclusion** | w/ static | w/o static | w/ static | w/o static | w/ static | w/o static | w/ static | w/o static |
| LR | 52.9±1.2 | **51.7±1.2** | 36.9±1.1 | 34.0±1.1 | 33.0±0.7 | 32.2±0.6 | 39.7±0.6 | 36.9±0.6 |
| LGBM | **53.7±1.2** | 51.1±1.1 | **40.6±0.8** | 38.9±1.5 | **36.0±0.6** | 34.1±0.7 | **44.2±0.7** | 41.0±0.7 |
| GRU | 53.1±1.5 | 52.9±1.2 | 37.6±1.2 | 37.3±1.1 | **36.1±0.9** | 35.3±0.8 | 42.4±0.6 | 41.5±0.7 |
| LSTM | **53.6±1.4** | 51.0±1.0 | 37.8±1.0 | 36.2±1.4 | 35.7±0.8 | 34.6±0.7 | 41.0±0.7 | 40.2±0.8 |
| TCN | **54.2±1.4** | 52.7±1.0 | 39.2±1.3 | 37.5±1.5 | 34.3±0.6 | 35.4±0.8 | 41.4±0.8 | 40.8±0.7 |
| TF | **54.4±1.1** | 52.7±0.9 | 39.3±1.5 | 38.4±1.5 | 34.7±0.8 | 34.5±0.7 | 42.2±0.3 | 41.3±0.8 |

found in Table 9 (the AUPRC in Table 10). From these results, we can see that including the static data seems to result in better performance across all models and datasets.

TABLE 11: *Baseline AUROC (↑), AUPRC (↑) performance of the included ML algorithms on the demo cohorts.*

| | eICU Demo | | MIMIC-III Demo | |
|---|---|---|---|---|
| **Algorithm** | AUROC | AUPRC | AUROC | AUPRC |
| **ICU Mortality** | | | | |
| LR | 67.7±0.9 | 15.4±1.3 | 52.9±3.2 | **37.6±2.3** |
| LGBM | **72.9±1.1** | **21.4±1.7** | **59.0±2.1** | 33.0±2.4 |
| GRU | 71.5±1.3 | **20.8±1.5** | 50.5±3.6 | 33.9±3.5 |
| LSTM | **71.3±1.6** | 18.9±1.9 | 55.5±2.3 | 34.2±3.3 |
| TCN | 69.2±1.7 | 18.0±1.5 | 55.1±4.0 | **38.8±3.6** |
| TF | 71.6±1.3 | 18.7±1.7 | 53.3±3.5 | **36.8±3.1** |
| **AKI** | | | | |
| LR | 61.3±0.6 | 16.8±0.4 | 52.9±2.3 | 16.8±2.0 |
| LGBM | **72.6±0.3** | **23.8±0.4** | **60.7±1.7** | **22.4±1.8** |
| GRU | 63.0±0.8 | 17.3±0.6 | 50.5±3.3 | 17.6±1.5 |
| LSTM | 61.9±0.9 | 16.2±0.6 | 53.5±2.5 | 15.4±1.3 |
| TCN | 64.5±1.0 | 17.6±0.6 | 53.7±2.8 | 19.6±2.0 |
| TF | 70.1±0.5 | 21.7±0.7 | 53.9±2.0 | 15.3±1.0 |
| **Sepsis** | | | | |
| LR | 63.8±0.9 | 3.7±0.3 | * | * |
| LGBM | 53.5±1.5 | 2.8±0.2 | * | * |
| GRU | 64.7±1.3 | 4.1±0.3 | * | * |
| LSTM | 65.3±1.3 | 4.3±0.5 | * | * |
| TCN | 66.4±1.1 | 4.2±0.3 | * | * |
| TF | **68.4±1.1** | 5.8±0.6 | * | * |

* Our sepsis definition resulted in just one sepsis case for the MIMIC-III demo dataset. As a result, we could not use the 5-fold cross-validation approach to train a model reliably.

**Demo datasets** We offer out-of-the-box (i.e., executable straight after downloading the repository) experiment definitions with five tasks defined on two demo datasets: MIMIC-III demo and eICU demo. The results can be seen in Table 11 and 12. The traditional ml models perform better,

TABLE 12: *Baseline performance on the regression tasks.* Results are reported in Mean Absolute Error (↓).

| | Kidney function | | Length of Stay | |
|---|---|---|---|---|
| | **eICU Demo** | **MIMIC-III Demo** | **eICU Demo** | **MIMIC-III Demo** |
| EN | **0.30±0.00** | **0.33±0.03** | 38.5±0.2 | 52.1±1.3 |
| LGBM | 0.87±0.01 | 0.86±0.04 | **37.7±0.2** | **50.4±1.1** |
| GRU | 0.54±0.02 | 3.23±0.19 | 39.7±0.6 | 54.7±1.0 |
| LSTM | 0.51±0.02 | 2.95±0.18 | 39.1±0.5 | 54.8±1.2 |
| TCN | 0.46±0.02 | 2.98±0.19 | 38.4±0.6 | 56.8±1.2 |
| TF | 0.60±0.03 | 3.22±0.18 | 38.9±1.2 | 57.1±1.2 |

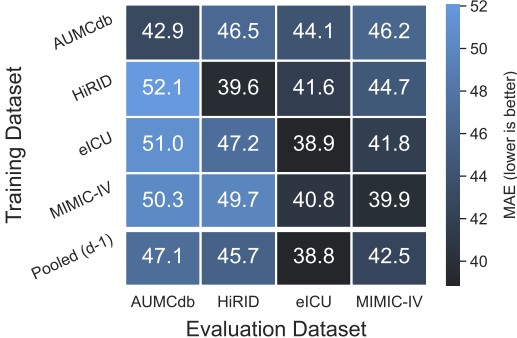

FIGURE 4: *Performance in MAE of the Transformer model on length of stay (LoS).*

most likely explained by the low number of samples. The kidney function task highlights the large difference in performance especially.

**External validation (extended)** The length of stay (LoS) results for ICU mortality prediction can be found in Figure 4.

**Fine-tuning (extended)** The AUPRC results for our experiment with transfer learning can be found below and show a similar trend to Figure 3.

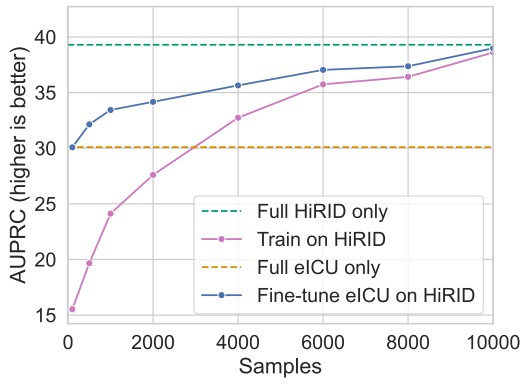

FIGURE 5: *AUPRC for fine-tuning an eICU GRU model for ICU mortality prediction on HiRID.*

## C  APPENDIX: DATASETS

This Appendix contains detailed description of the datasets and the preprocessing methodology.

### C.1  DATABASE CHARACTERISTICS

The Medical Information Mart for Intensive Care (MIMIC)-III dataset is the most commonly used dataset used for ML in ICU settings; Syed et al. (2021) found 61 eligible studies that used a form of the MIMIC dataset. It was collected in the USA at the Beth Isreal Deaconess Medical Center (Johnson et al., 2016). The newer MIMIC-IV includes several improvements, among which newer patient records and a revised structure including regular hospital information (Johnson et al., 2023).The eICU Collaborative Research Database (eICU) (Pollard et al., 2018) is an effort to collect the first sizable (200,000 admissions) multi-center dataset. It was collected using Philips ICU monitoring systems in the USA at 208 participating hospitals. The High Time Resolution ICU Dataset (HiRID) dataset was collected at Bern University Hospital, Switzerland, and has incorporated more observations than the aforementioned datasets (Hyland, 2020). The AmsterdamUMCdb (AUMCdb) is the most recently released ICU dataset (Thoral et al., 2021). Collected in the Netherlands, it has a temporal resolution of up to 1 minute and has prioritized patient de-identification. Note that there is no benchmark software for this dataset yet. Each dataset we are using has undergone de-identification procedures, and we have not tried to re-identify the people involved, as per the user agreement for each dataset. Table 13 shows some key characteristics of each dataset. A more comprehensive overview of ICU datasets can be found in the work of Sauer et al. (2022b).

TABLE 13: *Supplemental details of openly accessible ICU datasets.* Note that accessing each dataset requires completing a credentialing procedure.

| Dataset | MIMIC-III / IV | eICU | HiRID | AUMCdb |
|---|---|---|---|---|
| Stays* | 40k (0.1k)** / 73k | 201k (2k) | 34k | 23k |
| Version | v1.4 / v2.2 | v2.0 | v1.1.1 | v1.0.2 |
| Frequency (time-series) | 1 hour | 5 minutes | 2 / 5 minutes | up to 1 minute |
| Origin | USA | USA | Switzerland | Netherlands |
| Originally published | 2015 (Johnson et al., 2016) / 2020 (Johnson et al., 2023) | 2017 (Pollard et al., 2018) | 2020 (Hyland, 2020) | 2019 (Thoral et al., 2021) |
| License | A (C) / A | A (C) | A | B |
| Repository link | $\rho$ ($\rho$)/ $\rho$ | $\rho$ ($\rho$) | $\rho$ | $\alpha$ |

Note that accessing each full dataset requires completing a credentialing procedure.
*: Stays were taken and rounded from the latest available versions of the databases as of the time of writing.
**: The brackets () indicate characteristics of the demo (freely accessible) version of the dataset
A: PhysioNet Contributor Review Health Data License 1.5.0
B: Access Request Form and End User License Agreement for AmsterdamUMCdb 1.6
C: Open Data Commons Open Database License v1.0
$\rho$: Physionet
$\alpha$: Amsterdam Medical Data Science

The authors of MIMIC-III and eICU have made small selected datasets available for the purpose of experimentation. These datasets are also publicly available on Physionet. We support the publicly accessible "demo" datasets provided for eICU[10] and MIMIC-III[11]. In accordance with the demo dataset license (Open Data Commons Open Database License v1.0, see Table 13, License C), it is permitted to adapt and share the data. Still, we recommend the user to complete a human subject research training to make sure the usage of the dataset does not violate the usage proposal. They contain respectively 2,500 (eICU) and 100 stays (MIMIC-III) before exclusion. For the purposes testing and validating *YAIB*, we have created demo-cohorts, *extracted solely from these datasets*, for each of our supported tasks. Usage of the task cohorts and dataset is only permitted in accordance with the above license.

---

[10] https://physionet.org/content/eicu-crd-demo
[11] https://physionet.org/content/mimiciii-demo

TABLE 14: *Characteristics of 1) the included datasets (above) and 2) the task cohorts (below).*

| General characteristics | AUMCdb | HiRID | eICU | MIMIC-IV |
|---|---|---|---|---|
| **Version** | 1.02 | 1.1.1 | 2.0 | 2.0 |
| **Number of patients** | 19,790 | -[1] | 160,816 | 53,090 |
| **Number of ICU stays** | 22,636 | 32,338 | 182,774 | 75,652 |
| **Age at admission** (years) | 65 [55, 75][2] | 65 [55, 75] | 65 [53, 76] | 65 [53, 76] |
| **Female** | 7,699 (35) | 11,542 (36) | 83,940 (46) | 33,499 (44) |
| **Race** | | | | |
| Asian | - | - | 3,008 (3) | 2,225 (3) |
| Black | - | - | 19,867 (11) | 8,223 (12) |
| White | - | - | 140,938 (78) | 51,575 (76) |
| Other | - | - | 16,978 (9) | 5,514 (8) |
| Unknown | | | 1,983 | 8,115 |
| **Admission type** | | | | |
| Medical | 4,131 (21) | - | 134,532 (79) | 49,217 (65) |
| Surgical | 14,007 (72) | - | 31,909 (19) | 25,674 (34) |
| Other | 1,225 (6) | - | 4,702 (3) | 761 (1) |
| Unknown | 1,069 | - | 11,631 | 0 |
| **Hospital length of stay** (days) | - | - | 6 [3, 10] | 7 [4, 13] |
| **Task cohorts** | **AUMCdb** | **HiRID** | **eICU** | **MIMIC-IV** |
| **ICU mortality** | | | | |
| Number of included stays | 10,535 | 12,859 | 113,382 | 52,045 |
| Died | 1,660 (15.8) | 1,097 (8.2) | 6,253 (5.5) | 3,779 (7.3) |
| **Onset of acute kidney injury** | | | | |
| Number of included stays | 20,290 | 31,772 | 164,791 | 66,032 |
| KDIGO* $\geq$ 1 | 3,776 (18.6) | 7,383 (23.2) | 62,535 (37.9) | 27,509 (41.7) |
| **Onset of Sepsis** | | | | |
| Number of included stays | 18,184 | 29,894 | 123,864 | 67,056 |
| Sepsis-3 criteria | 764 (4.2) | 1,986 (6.6) | 5,835 (4.7) | 3,730 (5.6) |
| **Kidney function (creatinine)** | | | | |
| Number of included stays | 8,003 | 7,499 | 69,117 | 35,657 |
| Creatinine value | 0.97 [0.70, 1.61] | 0.92 [0.67, 1.50] | 1.00 [0.71, 1.68] | 1.00 [0.70, 1.60] |
| **ICU remaining length of stay** | | | | |
| Number of included stays | 22,636 | 32,338 | 182,774 | 75,652 |
| ICU length of stay (hours) | 24 [19, 77] | 24 [19, 50] | 42 [23, 76] | 48 [26, 89] |

[1] HiRID only provides stay-level identifiers.
[2] Since AUMCdb only includes age groups, we calculated the median of the group midpoints.
* KDIGO, Kidney Disease Improving Global Outcomes (KDIGO, 2012).
**Numeric** variables are summarized by *median [IQR]*.
**Categorical** variables are summarized by *incidence (%)*.

## C.2 EXCLUSION CRITERIA

We included all available ICU stays of adult patients in our analysis. For each stay, we applied the following exclusion criteria to ensure sufficient data volume and quality: remove any stays with **1)** an invalid admission or discharge time defined as a missing value or negative calculated length of stay, **2)** less than six hours spent in the ICU, **3)** less than four separate hours across the entire stay where at least one feature was measured, **4)** any time interval of $\geq$12 consecutive hours throughout the stay during which no feature was measured. Figure 6 details the number of stays overall and by dataset excluded this way.

Additional exclusion criteria were applied based on the individual tasks; the details can be found schematically in Figure 7 and 8. For ICU mortality, we excluded all patients with a length of stay of fewer than 30 hours (either due to death or discharge). A minimum length of 30 hours was chosen to exclude any patients that were about to die (the sickest patients) or be discharged (the healthiest patients) at the time of prediction at 24 hours. For creatinine (kidney function), we excluded all patients with a length of stay of fewer than 48 hours or without a creatinine measurement between 24 and 48 hours (which was the outcome of interest). For AKI and sepsis, we excluded any stays where disease onset was outside the ICU or within the first six hours of the ICU stay. To account

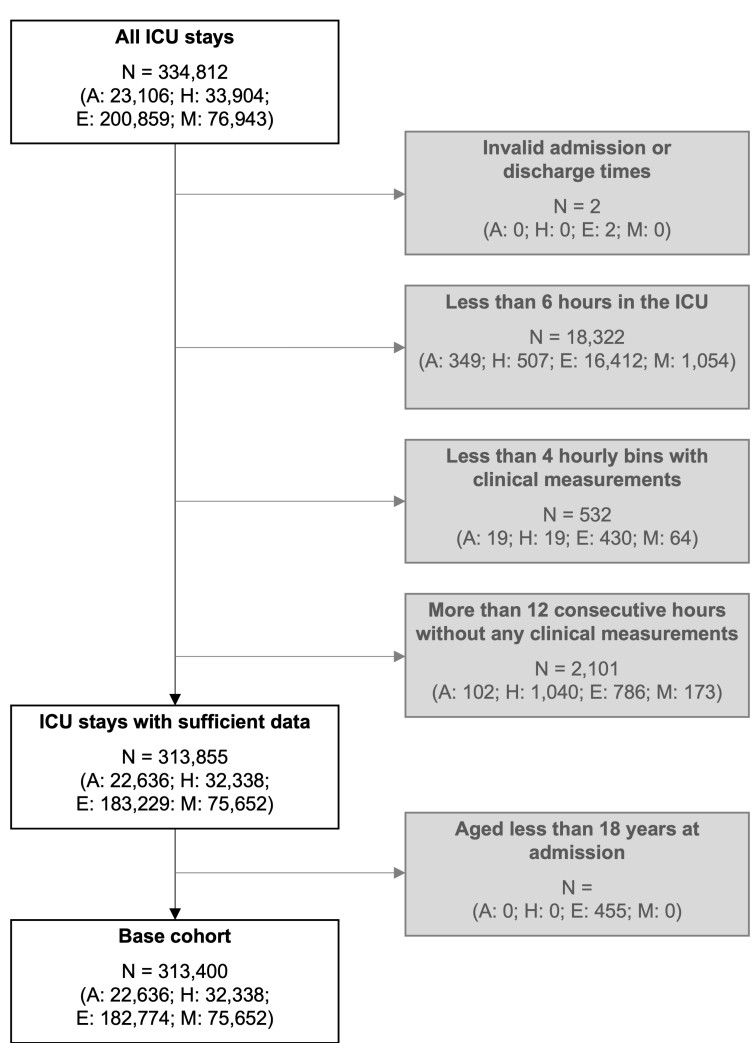

FIGURE 6: *Exclusion criteria applied to the base cohort*. **N**: Total amount of cases. **A**: AUMCdb, **H**: HiRID, **E**: eICU, **M**: MIMIC-IV

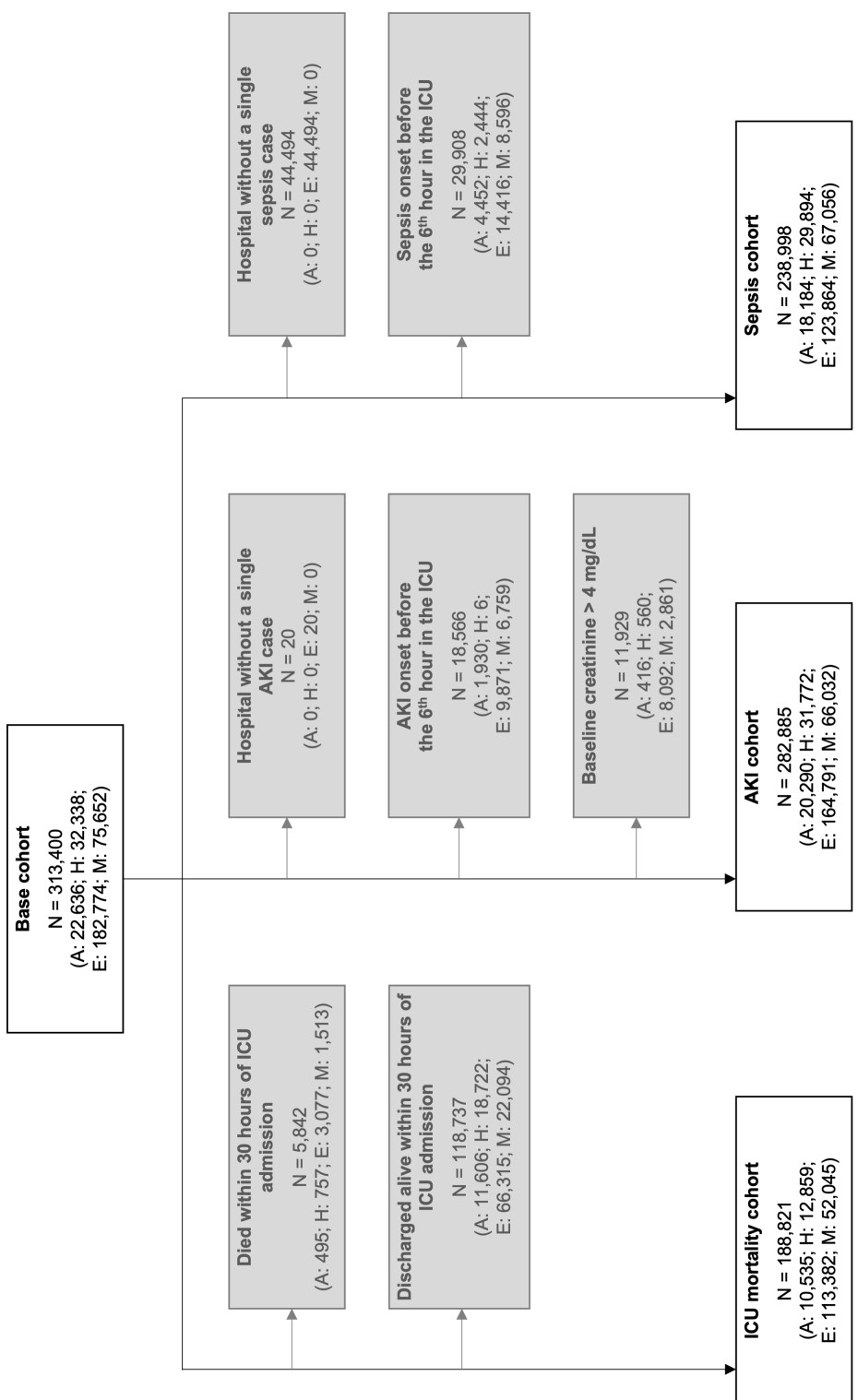

FIGURE 7: Additional exclusion criteria applied for the classification tasks.

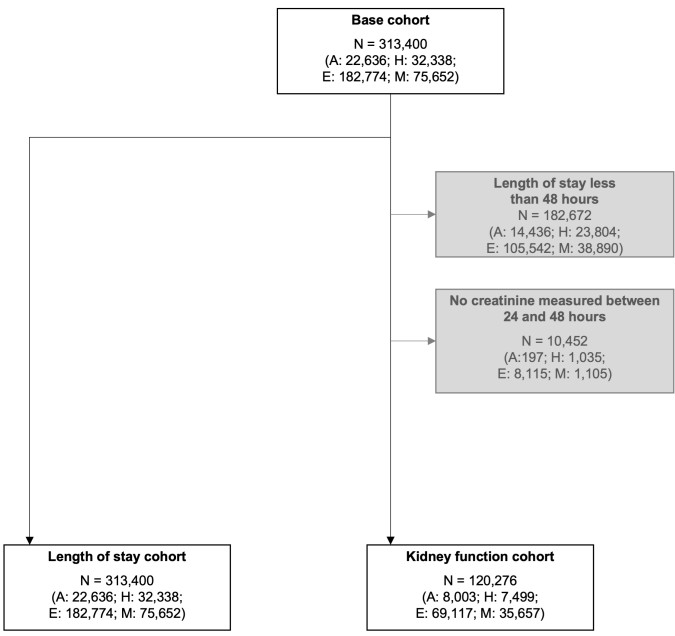

FIGURE 8: Additional exclusion criteria applied for the regression tasks.

for differences in data recording across hospitals in eICU, we further excluded hospitals that did not have a single patient with AKI or sepsis to exclude hospitals with an insufficient recording of features necessary to define the outcome. Finally, for the AKI task, we excluded stays where the baseline creatinine, defined as the last creatinine measurement prior to ICU (if exists) or the earliest measurement in the ICU, was >4 mg/dL to exclude patients with preexisting renal insufficiency. For a numerical overview, please consult Table 14.

## C.3 PREPROCESSING

A total of 52 features were used for model training (Table 15), 4 of which were static and 48 that were dynamic. These features were selected as they are available across all datasets for most patients. Dynamic features primarily include vital signs (7 variables) and laboratory tests (39 variables), with two more variables that measure input (fraction of inspired oxygen) and output (urine). All variables were extracted via the `ricu` R package (version 0.5.3). The `ricu` name for each package is shown in Table 15. The exact definition for each feature and how it was extracted from the individual databases can be found in the concept configuration file of the package's GitHub repository (commit 885bd0c). We also provided cohort definition code for this work, which can be run in both R and Python, in a github repository.

TABLE 15: *Clinical concepts used as input to the prediction models.*

| Feature | ricu | unit |
|---|---|---|
| **Static** | | |
| Age at hospital admission | age | *Years* |
| Female sex | sex | - |
| Patient height | height | *cm* |
| Patient weight | weight | *kg* |
| | | |
| **Time-varying** | | |
| Blood pressure (systolic) | sbp | *mmHg* |

TABLE 15: *Clinical concepts used as input to the prediction models (continued)*

| Feature | ricu | unit |
|---|---|---|
| Blood pressure (diastolic) | dbp | *mmHg* |
| Heart rate | hr | *beats/minute* |
| Mean arterial pressure | map | *mmHg* |
| Oxygen saturation | o2sat | *%* |
| Respiratory rate | resp | *breaths/minute* |
| Temperature | temp | *°C* |
| Albumin | alb | *g/dL* |
| Alkaline phosphatase | alp | *IU/L* |
| Alanine aminotransferase | alt | *IU/L* |
| Aspartate aminotransferase | ast | *IU/L* |
| Base excess | be | *mmol/L* |
| Bicarbonate | bicar | *mmol/L* |
| Bilirubin (total) | bili | *mg/dL* |
| Bilirubin (direct) | bili_dir | *mg/dL* |
| Band form neutrophils | bnd | *%* |
| Blood urea nitrogen | bun | *mg/dL* |
| Calcium | ca | *mg/dL* |
| Calcium ionized | cai | *mmol/L* |
| Creatinine | crea | *mg/dL* |
| Creatinine kinase | ck | *IU/L* |
| Creatinine kinase MB | ckmb | *ng/mL* |
| Chloride | cl | *mmol/L* |
| $CO^2$ partial pressure | pco2 | *mmHg* |
| C-reactive protein | crp | *mg/L* |
| Fibrinogen | fgn | *mg/dL* |
| Glucose | glu | *mg/dL* |
| Haemoglobin | hgb | *g/dL* |
| International normalised ratio (INR) | inr_pt | - |
| Lactate | lact | *mmol/L* |
| Lymphocytes | lymph | *%* |
| Mean cell haemoglobin | mch | *pg* |
| Mean corpuscular haemoglobin concentration | mchc | *%* |
| Mean corpuscular volume | mcv | *fL* |
| Methaemoglobin | methb | *%* |
| Magnesium | mg | *mg/dL* |
| Neutrophils | neut | *%* |
| $O^2$ partial pressure | po2 | *mmHg* |
| Partial thromboplastin time | ptt | *sec* |
| pH of blood | ph | - |
| Phosphate | phos | *mg/dL* |
| Platelets | plt | *1,000 / $\mu$L* |
| Potassium | k | *mmol/L* |
| Sodium | na | *mmol/L* |
| Troponin T | tnt | *ng/mL* |
| White blood cells | wbc | *1,000 / $\mu$L* |
| Fraction of inspired oxygen | fio2 | *%* |
| Urine output | urine | *mL* |

**Additional features**  Furthermore, we consulted clinical experts to identify which features might be missing from our prediction setup. Several clinical features are currently missing from this setup, which could potentially improve prediction performance: *Glasgow coma scale score, Intubation, Ventilator settings, Renal replacement therapy, and Vasopressors*. We expect to be able to integrate more concepts as we collaborate with authors of datasets to make them available.

## D APPENDIX: OUTCOME DEFINITIONS

The outcome definitions per task for each dataset are detailed in this Appendix.

### D.1 ICU MORTALITY

ICU mortality was defined as death while in the ICU. This was generally ascertained via the recorded discharge status or discharge destination. Note that our definition of ICU mortality differs from the definition of `death` in the `ricu` R package, which describes hospital mortality that is unavailable for some included datasets.

**AUMCdb** Death was inferred from the `destination` column of the `admissions` table. A destination of "Overleden" (Dutch for "passed away") was treated as a death in the ICU. Since the date of death was recorded outside of the ICU and may therefore be imprecise, the recorded ICU discharge date was used as a more precise proxy for the time of death.

**HiRID** Death was inferred from the column `discharge_status` in table `general`. The status of "dead" was treated as a death in the ICU. Time of death was inferred as the last measurement of IDs `110` (mean arterial blood pressure) or `200` (heart rate) in column `variableid` of table `observations`.

**eICU** Death was inferred from the column `unitdischargestatus` in table `patient`. The status of "Expired" was treated as a death in the ICU. The recorded ICU discharge date was used as a proxy for the time of death.

**MIMIC IV** Death was inferred from the column `hospital_expire_flag` in table `admissions`. Since MIMIC IV only records a joint ICU/hospital expiration flag, ward transfers were analyzed to ascertain the location of death. If the last ward was the ICU, the death was considered ICU mortality.

TABLE 16: Staging of AKI according to KDIGO (KDIGO, 2012)

| Stage | Serum creatinine | Urine output |
|-------|------------------|--------------|
| 1 | 1.5–1.9 times baseline

OR

$\geq$0.3 mg/dl ($\geq$26.5 $\mu$mol/l) increase within 48 hours | <0.5 ml/kg/h for 6–12 hours |
| 2 | 2.0–2.9 times baseline | <0.5 ml/kg/h for $\geq$12 hours |
| 3 | 3.0 times baseline (prior 7 days)

OR

Increase in serum creatinine to $\geq$4.0 mg/dl ($\geq$353.6 $\mu$mol/l) within 48 hours

OR

Initiation of renal replacement therapy | <0.3 ml/kg/h for $\geq$24 hours

OR

Anuria for $\geq$12 hours |

AKI, acute kidney injury; KDIGO, Kidney Disease Improving Global Outcomes.

### D.2 ACUTE KIDNEY INJURY

AKI was defined as KDIGO stage $\geq$1, either due to an increase in serum creatinine or low urine output (Table 16) (KDIGO, 2012). Baseline creatinine was defined as the lowest creatinine measurement over the last 7 days. Urine rate was calculated as the amount of urine output in ml divided by the

number of hours since the last urine output measurement (for a max gap of 24h), except for HiRID, in which urine rate was recorded directly. The earliest urine output was divided by 1. The rate per kg was calculated based on the admission weight. If weight was missing, a weight of 75 kg was assumed instead.

**AUMCdb** Creatinine was defined via the standard `ricu` concept of serum creatinine as IDs `6836`, `9941`, or `14216` in column `itemid` of table `numericitems`. Urine output was defined as IDs `8794, 8796, 8798, 8800, 8803` in column `itemid` of table `numericitems` (note that this includes more items than those included in the standard `ricu` concept of urine output).

**HiRID** Creatinine was defined via the standard `ricu` concept of serum creatinine as ID `20000600` in column `variableid` of table `observations`. Urine rate was defined as ID `10020000` in column `variableid` of table `observations`.

**eICU** Creatinine was defined via the standard `ricu` concept of serum creatinine as IDs "creatinine" in column `labname` of table `lab`. Urine output was defined via the standard `ricu` concept of urine output as IDs "Urine" and "URINE CATHETER" in column `celllabel` of table `intakeoutput`.

**MIMIC IV** Creatinine was defined via the standard `ricu` concept of serum creatinine as ID `50912` in column `itemid` of table `labevents`. Urine output was defined via the standard `ricu` concept of urine output as IDs `226557, 226558, 226559, 226560, 226561, 226563, 226564, 226565, 226566, 226567, 226584, 227510` in column `itemid` of table `outputevents`.

TABLE 17: *Comparing MIMIC-IV sepsis cohorts according to three different definitions.*

| Cohort | Seymour et al. (2016)* | Moor et al. (2021a) | Calvert et al. (2016) |
|---|---|---|---|
| **Stays** | 67,056 | 53,642 | 65,901 |
| **Prevalence** | 3,730 (5.6%) | 8,919 (16.7%) | 2,406 (3.7%) |

*Our default definition.

## D.3 SEPSIS

The onset of sepsis was defined using the Sepsis-3 criteria (Singer et al., 2016), which defines sepsis as organ dysfunction due to infection. Following guidance from the original authors of Sepsis-3 (Seymour et al., 2016), organ dysfunction was defined as an increase in SOFA score $\geq 2$ points compared to the lowest value over the last 24 hours. Suspicion of infection was defined as the simultaneous use of antibiotics and culture of body fluids. The time of sepsis onset was defined as the first time of organ dysfunction within 48 hours before and 24 hours after suspicion of infection. Time of suspicion was defined as the earlier antibiotic initiation or culture request. Antibiotics and culture were considered concomitant if the culture was requested $\leq 24$ hours after antibiotic initiation or if antibiotics were started $\leq 72$ hours after the culture was sent to the lab. Where available, antibiotic treatment was inferred from administration records; otherwise, we used prescription data. To exclude prophylactic antibiotics, we required that antibiotics were administered continuously for $\geq 3$ days (Reyna et al., 2019). Antibiotic treatment was considered continuous if an antibiotic was administered once every 24 hours for 3 days (or until death) or was prescribed for the entire time spent in the ICU. HiRID and eICU did not contain microbiological information. For these datasets, we followed Moor et al. (2021a) and defined suspicion of infection through antibiotics alone. Note, however, that the sepsis prevalence in our study was considerably lower than theirs, which was as high as 37% in HiRID. We suspect this is because they did not require treatment for $\geq 3$ days. For comparison and to contextualize the results of our experiments, we have benchmarked other sepsis definitions. See Table 17 for the number of stays and incidence for each cohort in this experiment performed on MIMIC-IV.

**AUMCdb** The SOFA score, microbiological cultures, and antibiotic treatment were defined via the standard `ricu` concepts `sofa`, `abx`, and `samp` (see the `ricu` package for more details).

**HiRID** The SOFA score and antibiotic treatment were defined via the standard `ricu` concept `sofa` and `abx` (see the `ricu` package for more details). No microbiology data were available in HiRID.

**eICU** The SOFA score and antibiotic treatment were defined via the standard `ricu` concept `sofa` and `abx` (see the `ricu` package for more details). Microbiology data in eICU was not reliable (Moor et al., 2021a) and therefore omitted.

**MIMIC-IV** The SOFA score and microbiological cultures were defined via the standard `ricu` concepts `sofa` and `samp` (see the `ricu` package for more details). Antibiotics were defined based on table `inputevents`. This differs from the standard `ricu` `abx` concept, which also considers the `prescriptions` table.

## D.4 KIDNEY FUNCTION

The median creatinine level over the course of the second day of a patient's ICU stay was defined as the target of interest. Given the available datasets, this was chosen as the most suitable proxy of kidney function.

**AUMCdb** Creatinine was defined via the standard `ricu` concept of serum creatinine as IDs `6836`, `9941`, or `14216` in column `itemid` of table `numericitems`.

**HiRID** Creatinine was defined via the standard `ricu` concept of serum creatinine as ID `20000600` in column `variableid` of table `observations`.

**eICU** Creatinine was defined via the standard `ricu` concept of serum creatinine as IDs "creatinine" in column `labname` of table `lab`.

**MIMIC IV** Creatinine was defined via the standard `ricu` concept of serum creatinine as ID `50912` in column `itemid` of table `labevents`.

## D.5 REMAINING LENGTH OF STAY

At each hour, the remaining length of stay in the ICU was calculated in hours until discharge. A maximum forecasting window of 7 days was chosen, as forecasts beyond this interval were judged extremely difficult and of lesser clinical relevance. Arguably, an even shorter window of 3 days could be chosen to support short-term capacity planning. This is left for future investigation.

**AUMCdb** Length of stay was calculated via the standard `ricu` concept using the columns `admittedat` and `dischargedat` of table `admissions`.

**HiRID** Length of stay was calculated via the standard `ricu` concept using the column `admissiontime` of table `general` as well as the last observation in table `observations`.

**eICU** Length of stay was calculated via the standard `ricu` concept using the columsn `unitadmitoffset` and `unitdischargeoffset` of table `patient`.

**MIMIC IV** Length of stay was calculated via the standard `ricu` concept using the columsn `intime` and `outtime` of table `icustays`.

## E  APPENDIX: YAIB'S USAGE AND IMPLEMENTATION

We describe the most important practical aspects of using *YAIB* in research in this Appendix. Please note that there is a *wiki*, dedicated to usage and development, available at: `https://github.com/rvandewater/YAIB/wiki`.

## E.1 USE OF EXISTING CODE REPOSITORIES

As mentioned in the main text, we used parts of the HiRID-Benchmark code and heavily modified and extended the code to support our extra features and extensibility, mentioned in Table 1. HiRID-Benchmark is available at GitHub and makes use of an MIT license, as does our code repository.

## E.2 ADDING A DATA SOURCE

Adding a new dataset type, to use within *YAIB*, can be easily done by providing it in a `.gin` task definition file, see Code Listing 1. Note, however, that any datasets formatted in the default way do

not require any changes to be used by *YAIB*. By default, we have chosen to work with the Apache parquet (Vohra, 2016) file format, which is a modern, open-source column-oriented format that does not require a lot of storage due to efficient data compression[12]. We separate the data into three separate files: `DYNAMIC`, `STATIC`, and `OUTCOME`; this is defined for dynamic variables (that change during the stay), constant parameters, and the prediction task label respectively. Our cohort definition code produces the files in exactly this format. Furthermore, we see the concept of `roles` with the definition of the `vars` dictionary. These roles are assigned as defined in RECIPYS , the preprocessing package developed alongside *YAIB*. The `GROUP` variable defines which internal dataset variable should be used to "group by" for, e.g., aggregating patient vital signs. The `SEQUENCE` variable defines the sequential dimension of the dataset (in the common case, this would be time). The other keys in this dictionary define the feature columns and outcome variables for prediction.

CODE LISTING 1: *Example preprocessing pipeline structure.*

```python
@gin.configurable("base_classification_preprocessor")
class DefaultClassificationPreprocessor(Preprocessor):
    def __init__(self, generate_features: bool = True, scaling: bool = True, use_static_features: bool = True):
        """
        Args:
            generate_features: Generate features for dynamic data.
            scaling: Scaling of dynamic and static data.
            use_static_features: Use static features.
        Returns:
            Preprocessed data.
        """

    def apply(self, data, vars):
        """
        Args:
            data: Train, validation and test data dictionary. Further divided in static, dynamic, and outcome.
            vars: Variables for static, dynamic, outcome.
        Returns:
            Preprocessed data.
        """
        ...
        return data

    def _process_static(self, data, vars):
        ...
        return data

    def _process_dynamic(self, data, vars):
        ...
        return data

    def _dynamic_feature_generation(self, data, dynamic_vars):
        ...
        return data
```

### E.3 CREATING A PREPROCESSING PIPELINE

Our preprocessing pipeline is set up to be as general as possible and allows for custom implementations, defined as subclass from the `Preprocessor` class and passed as a command-line argument. For our tasks, we have defined a default preprocessing pipeline for both classification and regression tasks. Code Listing 2 shows the class structure of the default classification preprocessor. In the private methods of this class, RECIPYS is used to apply feature generation steps (which differ for ml and dl models). The abstract `Preprocessor` has two functions that need to be implemented: `__init__()` (which initializes the preprocessor and configures the settings) and `apply(data)` (which returns the preprocessed data dictionary of features and labels for each of the train, validate, and test splits)

### E.4 EXAMPLE: TRAINING A MORTALITY PREDICTION LSTM MODEL

We demonstrate the complete process of training a Long Short-Term Memory (LSTM) model to predict sepsis on the MIMIC-III demo dataset with *YAIB*. This is shown schematically in Figure 9. In Code Listing 2, the basic task setup for mortality prediction after 24 hours is shown. We define the dataset files, by default split into 3 parquet files with the corresponding names. The listing describes three dataset components: dynamic data, outcome definitions, and static data. Below, one sees the variables and different *"roles"* assigned to concrete strings (see Appendix E.2 for detail). In this

---

[12]https://parquet.apache.org/

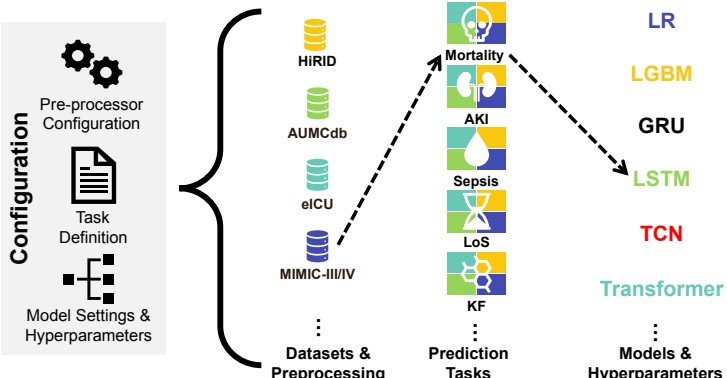

FIGURE 9: *Experiment definition schematic.* The fundamental experiment configuration of the benchmark contains three basic elements, **1)** the dataset, **2)** the prediction task, and **3)** the model and (list of) hyperparameters. Each element can be combined in different ways. Additionally, we provide an interface for extending each element (Datasets & Preprocessing, Prediction Tasks, Models & Hyperparameters) in the process. Provided is an example of an experiment configuration: predicting Sepsis on MIMIC (Thoral et al., 2021) with an LSTM (Hochreiter & Schmidhuber, 1997) model.

CODE LISTING 2: *Example configuration for a task definition configuration.* We identify 4 main categories: the prediction type (classification or regression), the deep learning loss, the dataset variables, and preprocessing settings, and the cross-validation settings. Note that the dataset is fully configurable here.

```
# COMMON IMPORTS                                                              1
include "configs/tasks/common/Imports.gin"                                   2
                                                                             3
# MODE SETTINGS                                                               4
Run.mode = "Classification"                                                  5
NUM_CLASSES = 2 # Binary classification                                      6
HORIZON = 24                                                                 7
train_common.weight = "balanced"                                            8
                                                                             9
# DEEP LEARNING                                                             10
DLPredictionWrapper.loss = @cross_entropy                                   11
# DATASET AND PREPROCESSING                                                 12
preprocess.file_names = {                                                   13
    "DYNAMIC": "dyn.parquet",                                               14
    "OUTCOME": "outc.parquet",                                              15
    "STATIC": "sta.parquet",                                                16
}                                                                          17
vars = {                                                                    18
    "GROUP": "stay_id",                                                     19
    "LABEL": "label",                                                       20
    "SEQUENCE": "time",                                                     21
    "DYNAMIC": ["alb", "alp", "alt", "ast", "be", "bicar", "bili", "bili_dir", "bnd", "bun", "ca", "cai", "ck", 22
        "ckmb", "cl",
        "crea", "crp", "dbp", "fgn", "fio2", "glu", "hgb", "hr", "inr_pt", "k", "lact", "lymph", "map", "mch", " 23
            mchc", "mcv",
        "methb", "mg", "na", "neut", "o2sat", "pco2", "ph", "phos", "plt", "po2", "ptt", "resp", "sbp", "temp", 24
            "tnt",
        "urine", "wbc"],                                                    25
    "STATIC": ["age", "sex", "height", "weight"],                          26
}                                                                          27
                                                                           28
# SELECTING PREPROCESSOR                                                    29
preprocess.preprocessor = @base_classification_preprocessor              30
preprocess.vars = %vars                                                     31
preprocess.use_static = True                                               32
                                                                           33
# SELECTING DATASET                                                        34
PredictionDataset.vars = %vars                                             35
                                                                           36
# CROSS VALIDATION                                                          37
execute_repeated_cv.cv_repetitions = 5                                      38
execute_repeated_cv.cv_folds = 5                                            39
```

listing, we also pass the vars to the preprocessing and dataset class. Finally, we see the definition of the cross-validation folds and iterations.

In Code Listing 3 we see the configuration for the LSTM. We first define the generating features from dynamic data (relevant for traditional ml). Then we bind the LSTM model with a gin flag. After this,

CODE LISTING 3: *Example configuration for hyperparameters of LSTM.* Tunable hyperparameter ranges can be floating points, integers, and categorical values.

```
import gin.torch.external_configurables
import icu_benchmarks.models.wrappers
import icu_benchmarks.models.encoders

default_preprocessor.generate_features = False

# Train params
train_common.model = @DLWrapper()

DLWrapper.encoder = @LSTMNet()
DLWrapper.optimizer_fn = @Adam
DLWrapper.train.epochs = 1000
DLWrapper.train.batch_size = 64
DLWrapper.train.patience = 10
DLWrapper.train.min_delta = 1e-4

# Optimizer params
optimizer/hyperparameter.class_to_tune = @Adam
optimizer/hyperparameter.weight_decay = 1e-6
optimizer/hyperparameter.lr = (1e-5, 3e-4)

# Encoder params
model/hyperparameter.class_to_tune = @LSTMNet
model/hyperparameter.num_classes = %NUM_CLASSES
model/hyperparameter.hidden_dim = (32, 256, "log-uniform", 2)
model/hyperparameter.layer_dim = (1, 3)

# Hyperparamter tuning
tune_hyperparameters.scopes = ["model", "optimizer"]
tune_hyperparameters.n_initial_points = 5
tune_hyperparameters.n_calls = 30
tune_hyperparameters.folds_to_tune_on = 2
```

CODE LISTING 4: *Running YAIB.* We train the LSTM model on the MIMIC-III demo dataset for the Mortality24 task.

```
#!
icu-benchmarks  train \
    -d demo_data/mortality24/mimic_demo \
    -n mimic_demo \
    -t BinaryClassification \
    -tn Mortality24 \
    --log-dir ../yaib_logs/ \
    -m LSTM \
    -gc \
    -lc \
    -s 2222 \
    -l ../yaib_logs/ \
    --tune
```

the hyperparameters are specified. The optimizer and encoder parameters are then specified. Note that we can specify ranges of hyperparameters to be tuned by the hyperparameter optimizer. Settings for this can be found in the bottom cluster of code. Code Listing 4 shows how to train our LSTM model on the `mimic_demo` dataset (included in our repository).

## E.5  OPTIONS

TABLE 18: *Options for the* `train` *command.*

| Flag | Required | Description |
|---|---|---|
| -reproducible | No | Make torch reproducible. (default: True) |
| -hp, -hyperparams | No | Hyperparameters for model. |
| -tune | No | Find best hyperparameters. (default: False) |
| -checkpoint | No | Use previous checkpoint. |

We specify the command line options that YAIB provides for benchmarking prediction tasks. One can use the `icu-benchmarks` command with either `train` or `evaluate`. Table 18 shows the

TABLE 19: *Options for the* `evaluate` *command.*

| Flag | Required | Description |
|------|----------|-------------|
| `-sn -source-name` | Yes | Name of the source dataset. |
| `-source-dir` | Yes | Directory containing gin and model weights. |

TABLE 20: *General arguments for the use of YAIB.*

| Flag | Required | Description |
|------|----------|-------------|
| `-d, -data-dir` | Yes | Path to the parquet data directory. |
| `-n, -name` | Yes | Name of the (target) dataset. |
| `-t, -task` | Yes | Name of the task gin. |
| `-m, -model` | No | Name of the model gin )Default. |
| `-tn, -task-name` | No | Name of the task, used for naming experiments. |
| `-e, -experiment` | No | Name of the experiment gin. |
| `-l, -log-dir` | No | Log directory with model weights. |
| `-s, -seed` | No | Random seed for processing, tuning, and training. |
| `-v, -verbose` | No | Whether to use verbose logging. (default: True) |
| `-cpu` | No | Set to use CPU. (default: False) |
| `-db, -debug` | No | Set to load less data. (default: False) |
| `-lc, -load_cache` | No | Set to load generated data cache. (default: False) |
| `-gc, -generate_cache` | No | Set to generate data cache. (default: False) |
| `-p, -preprocessor` | No | Load custom preprocessor from file. |
| `-pl, -plot` | No | Generate common plots. (default: False) |
| `-wd, -wandb-sweep` | No | Activates Weights and Biases hyper parameter sweep. (default: False) |
| `-imp, -pretrained-imputation` | No | Path to pre trained imputation model. |

arguments that can be used with `train`. Table 19 contains the options to use with `evaluate` (i.e., for evaluation with a pre-trained model). In Table 20, the general options for *YAIB* are shown.

# F  APPENDIX: EXTENDING YAIB

*YAIB* is built to adapt to your needs with as little effort as possible. *YAIB* allows you to change any part of your pipeline with ease: add a new dataset, define additional clinical concepts, adapt a task, implement imputation algorithms, use additional models, or evaluate custom metrics. This section provides examples on *YAIB* may be extended with respect to each of the above.

CODE LISTING 5: *ID and table configuration for the Salzburg Intensive Care Database Database in JSON.*

```
{                                                                             1
  "name": "sic",                                                             2
  "id_cfg": {                                                                3
    "patient": {                                                             4
      "id": "patientid",                                                     5
      "position": 1,                                                         6
      "start": "firstadmission",                                            7
      "end": "offsetofdeath",                                                8
      "table": "cases"                                                       9
    },                                                                      10
    "icustay": {                                                            11
      "id": "caseid",                                                       12
      "position": 2,                                                        13
      "start": "offsetafterfirstadmission",                                 14
      "end": "timeofstay",                                                  15
      "table": "cases"                                                      16
    }                                                                       17
  },                                                                        18
  "tables": {                                                               19
    "cases": {                                                              20
      "files": "cases.csv.gz",                                              21
      "defaults": {                                                         22
        "index_var": "offsetafterfirstadmission",                          23
        "time_vars": ["offsetafterfirstadmission", "offsetofdeath"]        24
      },                                                                    25
      "cols": {                                                             26
        "caseid": {                                                         27
          "name": "CaseID",                                                 28
          "spec": "col_integer"                                            29
        },                                                                  30
        "patientid": {                                                      31
          "name": "PatientID",                                             32
          "spec": "col_integer"                                            33
        },                                                                  34
        "admissionyear": {                                                  35
          "name": "AdmissionYear",                                          36
          "spec": "col_integer"                                            37
        },                                                                  38
        ...                                                                 39
      }                                                                     40
    },                                                                      41
    "d_references": {                                                       42
      ...                                                                   43
    },                                                                      44
    ...                                                                     45
  }                                                                         46
}                                                                           47
```

## F.1  ADD A NEW DATASET: SALZBURG INTENSIVE CARE DATABASE (SICDB)

SICdb is a recently published open source ICU dataset (Rodemund et al., 2023). The dataset includes 27,000 admissions to the ICU at University Hospital Salzburg (Austria) from 2013 to 2021. SICdb provides highly granular data with up to minute level resolution. To make SICdb available within *YAIB*, it must be interfaced to `ricu`. This ensures that `ricu` knows how to access and extract data from the dataset. Interfacing a new dataset requires three key steps:

1. Define the table structure and possible ID types as a JSON configuration.

2. Define how all ID systems and their origin times relate to each other.

**Define ID types and table structure**    First, `ricu` needs to know what tables and columns exist within the data source. This is specified via the JSON configuration file partially shown in Code Listing 5. Tables are defined under the `tables` element. The definition of each table contains the name of its source file, usually provided in `.csv` or `.csv.gz` format, and a list of all columns and their data types. In addition, default roles can be defined for certain columns in the table. These

usually include the time index (if present), all other time columns, and a column that is considered to contain the value of interest.

In addition to the available tables, `ricu` also expects information about the main ID types used in the dataset. Each piece of information in ICU datasets is usually linked to a certain unit of observation, most commonly the patient (`patient`), the hospital admission (`hospadm`), or the specific ICU stay (`icustay`). By knowing what IDs a piece of information is measured for, `ricu` is able to temporally relate all information within the dataset. For example, `labevents` in MIMIC IV are recorded for hospital admissions, whereas `chartevents` are recorded for ICU stays. Defining how these two ID systems relate to each other allows them to be mapped to a common time scale (e.g., time since ICU admission). At the same time, knowing that an ICU stay is detailed in `icustays` in MIMIC and in `cases` in SICdb allows to define the same semantic reference point in both databases.

The two ID types available in SICdb are the `patient` and the `icustay`. There is no separate demarcation of the `hospadm`. The observation time for a `patient` ranges from their first observed admission to their death (if it occurred). `icustays` range from the current admission to the ICU until the end of the stay.

**Calculate origin times for each ID type**   After `ricu` has been told which ID systems exist in the dataset, it also needs to know when they start and end. Much of this process is automated. For SICdb, all origin times are already provided in a format suitable for use with `ricu`, and thus the default behavior is appropriate for them. However, minor adjustments are often necessary. For example, discharge times in SICdb are not provided as absolute times but in seconds since the start of admission. Such adjustments can be made on a case-by-case basis by subtyping the respective functions (in this case `id_win_helper`) and overwriting the default behavior (Code Listing 6). Since SICdb provides time in seconds since admission, `ricu` must further be told to work with relative times in seconds. This can be conveniently achieved through existing helper functions (Code Listing 6).

Following the steps above makes SICdb available and fully usable within `ricu`. While some further helper functions may be necessary to enable optional functionality such as automatic determination of measurement units (SICdb stores units in a separate reference table that needs to be merged at runtime), these are not essential to the main functionality of `ricu`. Note that the above only interfaces SICdb. It does *not* automatically map all existing clinical concepts for SICdb. Defining a clinical concept for SICdb still requires manual mapping of the concept to SICdb data items, for example via the appropriate measurement IDs (see also the next session on adding a clinical concept). We do not expect that this process can ever be fully avoided (unless it was already performed prior, for example by mapping to and providing the data in OMOP format). However, we found that the framework and helper functions that `ricu` provides greatly simplify this process. Additional information on `ricu` and its design principles can be found in Bennett et al. (2023).

### F.2   DEFINE A CLINICAL CONCEPT: POTASSIUM CHLORIDE

*YAIB* comes with a wide range of pre-defined clinical concepts. However, it is likely that new tasks and applications require additional variables. An area of particular interest in this respect are medications. Using the exemplar of Potassium Chloride (a fluid frequently administered in the ICU), we demonstrate how new drugs can be added to *YAIB* with minimal effort. For many concepts, all that is required is a JSON dict that describes the correct measurement IDs within each dataset (Code Listing 7). The JSON snippet can then be appended to the existing `ricu` concept file (for additions that should become part of the main package) or added to the search path of `load_dictionary` via the `cfg_dirs` parameter. The definition of complicated transformations such as calculation of hourly rates is also supported by existing helper functions. If custom calculations are necessary, they can be provided as user-defined functions via the `callback` element.

### F.3   ADAPT A TASK: KDIGO STAGE

Cohorts created in *YAIB* are built to be easily adaptable. For example, predicting ordinal KDIGO stage rather than binary presence/absence of AKI is as simple as changing the outcome variable from `aki` to `kdigo`. More substantial changes to the task are also supported. If the general setup (i.e., covariates, exclusion criteria, etc.) remains the same but a novel outcome should be predicted,

CODE LISTING 6: *Helper functions that calculate the origin and temporal relation between events in the Salzburg Intensive Care Database.*

```
id_win_helper.sic_env <- function(x) {
  # return a mapping between two ID systems (e.g., ICU stay ID and patient ID),
  # including relative start and end times
  cfg <- sort(as_id_cfg(x), decreasing = TRUE)

  ids <- field(cfg, "id")
  sta <- field(cfg, "start")
  end <- field(cfg, "end")

  tbl <- as_src_tbl(x, unique(field(cfg, "table")))

  mis <- setdiff(sta, colnames(tbl))

  res <- load_src(tbl, cols = c(ids, intersect(sta, colnames(tbl)), end))

  assert_that(length(mis) == 1L)
  res[, firstadmission := 0L]

  res <- res[, c(sta, end) := lapply(.SD, s_as_mins), .SDcols = c(sta, end)]
  res[, timeofstay := offsetafterfirstadmission + timeofstay] # convert to absolute discharge time

  res <- setcolorder(res, c(ids, sta, end))
  res <- rename_cols(res, c(ids, paste0(ids, "_start"),
                       paste0(ids, "_end")), by_ref = TRUE)

  as_id_tbl(res, ids[2L], by_ref = TRUE)
}

load_difftime.sic_tbl <- function(x, rows, cols = colnames(x),
                           id_hint = id_vars(x),
                           time_vars = ricu::time_vars(x), ...) {
  # Load time differences in SICdb by treating each time variable as
  # the relative time since admission in seconds
  load_as_relative_time(x, {{ rows }}, cols, id_hint, time_vars, s_as_mins)
}
```

the outcome should be created as a clinical concept in `ricu` which can then simply be used as the outcome variable. If covariates, exclusion criteria, or prediction times change, existing helper functions can be utilized. See the existing task definitions for further examples.

## F.4 ADDING A NEW PREDICTION MODEL: RANDOM FOREST, RNN, AND TEMPORAL FUSION TRANSFORMER

We allow prediction models to be easily added and integrated into a Pytorch-lighting (PL) (Falcon & team, 2023) module. This incorporates advanced logging and debugging capabilities, as well as built-in parallelism. Our interface derives from the PL `BaseModule`[13].

For standard Scikit-Learn type ML models (e.g., Light Gradient Boosting Machine (LGBM) (Ke et al., 2017)), one can implement the `MLWrapper`, incorporating the model steps in the process. Note that this class is also derived from the PLBaseModule; this leads to minimal code overhead. See Code Listing 8 for details for the implementation of a random forest model. The only needed code here is the hyperparameter configuration and the initialization of the superclass.

The definition of DL models can be done by creating a subclass from the `DLPredictionWrapper`; this inherits the standard methods needed for training DL learning models. Again, our implementation using PL significantly reduces the code overhead and complexity. See Code Listing 9 for the example of a simple RNN model. We can then create a gin configuration file for this model such as that in Code Listing 11 to specify default parameters and hyperparameter ranges for hp-tuning.

More advanced, or state-of-the-art, models are also easily implemented. One of *YAIB*'s users has implemented a Temporal Fusion Transformer architecture (Lim et al., 2021). This model provides good performance on multi-horizon time series forecasting, as well as interpretable insights. Specifically, Lim et al. (2021) describe the TFT as follows: *"(1) examining the importance of each input variable in prediction, (2) visualizing persistent temporal patterns, and (3) identifying any regimes or events that lead to significant changes in temporal dynamics"*. See Code Listing 10 for

---

[13]https://lightning.ai/docs/pytorch/stable/common/lightning_module.html

CODE LISTING 7: *JSON definition for rate of potassium chloride across included datasets.*

```
"kcl_dur": {
  "description": "potassium chloride duration",
  "category": "medications",
  "aggregate": "max",
  "sources": {
    "aumc": [
      {
        "ids": 9001,
        "table": "drugitems",
        "sub_var": "itemid",
        "stop_var": "stop",
        "grp_var": "orderid",
        "callback": "aumc_dur"
      }
    ],
    "eicu": [
      {
        "regex": "^potassium chloride",
        "table": "infusiondrug",
        "sub_var": "drugname",
        "callback": "eicu_duration(gap_length = hours(5L))",
        "class": "rgx_itm"
      }
    ],
    "hirid": [
      {
        "ids": 1000396,
        "table": "pharma",
        "sub_var": "pharmaid",
        "grp_var": "infusionid",
        "callback": "hirid_duration"
      }
    ],
    "miiv": [
      {
        "ids": [225166, 227522],
        "table": "inputevents",
        "sub_var": "itemid",
        "stop_var": "endtime",
        "grp_var": "linkorderid",
        "callback": "mimic_dur_inmv"
      }
    ]
  }
}
```

CODE LISTING 8: *Example ML model definition.* We create a Random Forest classifier that implements the default *YAIB* prediction model interface

```
@gin.configurable
class RFClassifier(MLWrapper):
    _supported_run_modes = [RunMode.classification]

    def __init__(self, *args, **kwargs):
        self.model = self.set_model_args(ensemble.RandomForestClassifier, *args, **kwargs)
        super().__init__(*args, **kwargs)
```

implementation details. This implementation is based upon the NVIDIA PyTorch implementation[14]. To train and optimize this model from a choice of hyperparameters, we need to specify a GIN file to bind the parameters, see Code Listing 11. Note that we can use modifiers for the optimizer (e.g, Adam optimizer) and ranges that we can specify in rounded brackets "()". Square brackets, "[]", result in a random choice where the variable is uniformly sampled.

### F.5 ADDING A NEW IMPUTATION METHOD: CSDI

We have added a range of imputation methods to *YAIB*, including interfaces to existing imputation libraries (Du, 2023; Jarrett et al., 2022). Here, we describe the addition of a recently introduced method that uses conditional score-based diffusion models conditioned on observed data: the Conditional Score-based Diffusion models for Imputation (CSDI)(Tashiro et al., 2021). To make the process of implementing these models easier, we have created the ImputationWrapper class that extends

---

[14]https://github.com/NVIDIA/DeepLearningExamples/tree/master/PyTorch/Forecasting/TFT

CODE LISTING 9: *Example DL model definition.* We use the inbuilt Torch RNN layers to built a Recurrent Neural Network.

```
@gin.configurable
class RNNet(DLPredictionWrapper):
    """Torch standard RNN model"""

    _supported_run_modes = [RunMode.classification, RunMode.regression]

    def __init__(self, input_size, hidden_dim, layer_dim, num_classes, *args, **kwargs):
        super().__init__(
            input_size=input_size, hidden_dim=hidden_dim, layer_dim=layer_dim, num_classes=num_classes, *args, **
                kwargs
        )
        self.hidden_dim = hidden_dim
        self.layer_dim = layer_dim
        self.rnn = nn.RNN(input_size[2], hidden_dim, layer_dim, batch_first=True)
        self.logit = nn.Linear(hidden_dim, num_classes)

    def init_hidden(self, x):
        h0 = x.new_zeros(self.layer_dim, x.size(0), self.hidden_dim)
        return h0

    def forward(self, x):
        h0 = self.init_hidden(x)
        out, hn = self.rnn(x, h0)
        pred = self.logit(out)
        return pred
```

the pre-existing `DLWrapper` (itself a subclass of the `LightningModule` of Pytorch-lightning) with extra functionality.

The CSDI model is a diffusion model that follows the general architecture of conditional diffusion models(Ho et al., 2020); It introduces noise into a subset of time series data used as conditional observations to later denoise the data and predict accurate values for the imputation targets. CSDI is based on a U-Net architecture(Ronneberger et al., 2015) including residual connections.

Tashiro et al. (2021) included two additional features into their model, which are inspired by DiffWave (Kong et al., 2021): an attention mechanism and the ability to input side information. The attention mechanism uses transformer layers, as shown in Figure 10. An input with K features, L length, and C channels is reshaped first to apply temporal attention and later reshaped again to apply feature attention. The second additional feature allows side information to be used as input to the model by a categorical feature embedding (Tashiro et al., 2021).

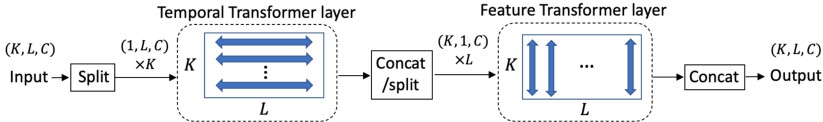

FIGURE 10: The attention mechanism of CSDI adapted from Tashiro et al. (2021).

See Code Listing 12 for the most important implementation code: the model initialization. We note that of this code, very little has been adapted from the original code repository[15] included in the original publication (Tashiro et al., 2021).

### F.6 ADDING AN EVALUATION METRIC: JENSEN SHANNON DIVERGENCE AND BINARY FAIRNESS

We support adding multiple types of evaluation metrics for benchmarking DL or ML models. We additionally support three common metric libraries: TorchMetrics (Nicki Skafte Detlefsen et al., 2022), Ignite (Fomin et al., 2020), and Scikit-Learn (Pedregosa et al., 2011). Adding a metric is a straightforward procedure. We added the Jensen Shannon Divergence (JSD) with the help of the SciPy library (Virtanen et al., 2020). See Code Listing 13 for details.

One can then add the metric to be evaluated for a particular, see Code Listing 15.

---

[15]https://github.com/ermongroup/CSDI/tree/main

CODE LISTING 10: *Temporal Fusion Transformer model definition.* Note that this implementation is similar to existing code and can use existing methods and the original dataloader of *YAIB* which avoids code duplication.

```python
class TFT(DLPredictionWrapper):
    """
    Implementation of Temporal Fusion Transformer, https://arxiv.org/abs/1912.09363.
    """

    _supported_run_modes = [RunMode.classification, RunMode.regression]
    def __init__(
        self,
        num_classes, # Classes for multiclass classification
        encoder_length, # Determines interval to use for prediction
        hidden, # Amount of hidden layers
        dropout, # Dropout layers
        n_heads, # Attention heads
        dropout_att,
        example_length, # Determines interval to predict
        quantiles=[0.1, 0.5, 0.9], # quantiles to produce
        static_categorical_inp_size=[2], # Number of categories
        temporal_known_categorical_inp_size=[],
        temporal_observed_categorical_inp_size=[48], # Number of categorical observed variables
        static_continuous_inp_size=3, # Number of static continuous variables
        temporal_known_continuous_inp_size=0,
        temporal_observed_continuous_inp_size=48,
        temporal_target_size=1, # Number of target variables
        **kwargs,
    ):

        #derived variables
        num_static_vars = len(static_categorical_inp_size) + static_continuous_inp_size
        num_future_vars = len(temporal_known_categorical_inp_size) + temporal_known_continuous_inp_size
        num_historic_vars = sum([num_future_vars, temporal_observed_continuous_inp_size, temporal_target_size,
            len(temporal_observed_categorical_inp_size),])

        super().__init__(num_classes=num_classes, encoder_length=encoder_length, hidden=hidden,
            n_heads=n_heads, dropout_att=dropout_att, example_length=example_length, quantiles=quantiles,
                num_static_vars=num_static_vars, num_future_vars=num_future_vars, num_historic_vars=
                num_historic_vars, *args,static_categorical_inp_size=1, temporal_known_categorical_inp_size
                =0, temporal_observed_categorical_inp_size=48, static_continuous_inp_size=3,
                temporal_known_continuous_inp_size=0, temporal_observed_continuous_inp_size=48,
                temporal_target_size=1, **kwargs)

        self.encoder_length = encoder_length # Determines from how distant past we want to use data from

        self.embedding = LazyEmbedding(static_categorical_inp_size, temporal_known_categorical_inp_size,
            temporal_observed_categorical_inp_size, static_continuous_inp_size,
            temporal_known_continuous_inp_size, temporal_observed_continuous_inp_size, temporal_target_size,
            hidden) # embeddings for all variables

        self.static_encoder = StaticCovariateEncoder(num_static_vars, hidden, dropout) # encoding for static
            variables
        self.TFTback = TFTBack(encoder_length, num_historic_vars, hidden, dropout, num_future_vars, n_heads,
            dropout_att, example_length,quantiles)
        self.logit = nn.Linear(len(quantiles), num_classes) # Linear layer to output to the number of classes
            and allow modification by predictionwrapper.
```

In order to asses fairness within ML prediction, a common metric to check is group fairness. This is computed through the ratio between positivity rates and true positives rates for different groups. Two types of these metrics are demographic parity (Calders et al., 2009) and equal opportunity ratio (Hardt et al., 2016). The TorchMetrics (Nicki Skafte Detlefsen et al., 2022) library includes the group fairness module interface which we can adapt for use in *YAIB*. We use a wrapper, that extends the TorchMetrics implementation and extracts a "group tensor" that indicates to which group the sample belongs. See Code Listing 14 to achieve the desired result. After this, we pass the data and feature names in the training step, as the function requires information about the assigned groups of the dataset; here, we track demographic parity by default, i.e., we use "sex" as the group name. In our pipeline, this has been one-hot encoded to 0 or 1. Then, adding this to the constants file, as seen in Code Listing 15, automatically calculates this metric during the training process.

CODE LISTING 11: *Temporal Fusion Transformer parameter configuration.*

```
# Hyperparameters for TFT model.

# Common settings for DL models
include "configs/prediction_models/common/DLCommon.gin"

# Optimizer params
train_common.model = @TFT

optimizer/hyperparameter.class_to_tune = @Adam
optimizer/hyperparameter.weight_decay = 1e-6
optimizer/hyperparameter.lr = (1e-5, 3e-4)

# Encoder params
model/hyperparameter.class_to_tune = @TFT
model/hyperparameter.encoder_length = 24
model/hyperparameter.hidden = 256
model/hyperparameter.num_classes = %NUM_CLASSES
model/hyperparameter.dropout = (0.0, 0.4)
model/hyperparameter.dropout_att = (0.0, 0.4)
model/hyperparameter.n_heads =4
model/hyperparameter.example_length=25
```

CODE LISTING 12: *Implementing the CSDI architecture in YAIB*. Note that our implementation is very similar to the original github repository, which demonstrates the flexibility of implementing new models in YAIB.

```
{
    def __init__(
        self, input_size, time_step_embedding_size, feature_embedding_size, unconditional, target_strategy,
            num_diffusion_steps, diffusion_step_embedding_dim, n_attention_heads, num_residual_layers,
            noise_schedule, beta_start, beta_end, n_samples, conv_channels, *args, **kwargs,
    ):
        super().__init__(...)
        self.target_dim = input_size[2]
        self.n_samples = n_samples

        self.emb_time_dim = time_step_embedding_size
        self.emb_feature_dim = feature_embedding_size
        self.is_unconditional = unconditional
        self.target_strategy = target_strategy

        self.emb_total_dim = self.emb_time_dim + self.emb_feature_dim
        if not self.is_unconditional:
            self.emb_total_dim += 1 # for conditional mask
        self.embed_layer = nn.Embedding(num_embeddings=self.target_dim, embedding_dim=self.emb_feature_dim)

        input_dim = 1 if self.is_unconditional else 2
        self.diffmodel = diff_CSDI(
            conv_channels,
            num_diffusion_steps,
            diffusion_step_embedding_dim,
            self.emb_total_dim,
            n_attention_heads,
            num_residual_layers,
            input_dim,
        )

        # parameters for diffusion models
        self.num_steps = num_diffusion_steps
        if noise_schedule == "quad":
            self.beta = np.linspace(beta_start**0.5, beta_end**0.5, self.num_steps) ** 2
        elif noise_schedule == "linear":
            self.beta = np.linspace(beta_start, beta_end, self.num_steps)

        self.alpha_hat = 1 - self.beta
        self.alpha = np.cumprod(self.alpha_hat)
        self.alpha_torch = torch.tensor(self.alpha).float().unsqueeze(1).unsqueeze(1)

}
```

# G  APPENDIX: EXPERIMENTAL SETUP AND REPRODUCIBILITY

To reproduce the results obtained in this paper, we have detailed our methodology in this Appendix.

## G.1  INFRASTRUCTURE AND HARDWARE

We used a high-performance computing cluster to perform our experimentsNo data was transferred to any external parties in this process. This computing cluster is run with renewable energy and can be

CODE LISTING 13: *Implementing JSD using SciPy in YAIB*. In this case we used the Ignite interface, but users can also choose to extend from the TorchMetric or SK-Learn interface.

```
class JSD(EpochMetric):
    def __init__(self, output_transform: Callable = lambda x: x, check_compute_fn: bool = False) -> None:
        super(JSD, self).__init__(lambda x, y: JSD_fn(x, y), output_transform=output_transform, check_compute_fn
            =check_compute_fn)

    def JSD_fn(y_preds: torch.Tensor, y_targets: torch.Tensor):
        return jensenshannon(abs(y_preds).flatten(), abs(y_targets).flatten()) ** 2
```

CODE LISTING 14: *Adding a wrapper for Group fairness metric.*

```
class BinaryFairnessWrapper(BinaryFairness):
    """
    This class is a wrapper for the BinaryFairness metric from TorchMetrics.
    """
    group_name = None
    def __init__(self, group_name = "sex", *args, **kwargs) -> None:
        self.group_name = group_name
        super().__init__(*args, **kwargs)
    def update(self, preds, target, data, feature_names) -> None:
        """ Standard metric update function"""
        groups = data[:, :, feature_names.index(self.group_name)]
        group_per_id = groups[:, 0]
        return super().update(preds=preds.cpu(),
                    target=target.cpu(),
                    groups=group_per_id.long().cpu())
```

considered climate-neutral. The cluster is running the SLURM (Yoo et al., 2003) management tool on Ubuntu 20.04.6 LTS. with a number of Nvidia A100, A40, RTX 2080TI, and RTX Titan GPUs. The traditional machine learning algorithms were trained with Intel Xeon Platinum 8160 and AMD EPYC 7643 CPU resources.

TABLE 21: *Average estimated duration of training tasks*

|  | Once per stay classification | Hourly classification | Hourly regression |
|---|---|---|---|
| Machine Learning Models | 3 minutes | 10 minutes | 10 minutes |
| Deep Learning Models | 30 minutes | 60 minutes | 90 minutes |

### G.2 LIBRARIES

A full list of libraries is available in the *YAIB* repository; please use the Conda environment manager to install these. We have aimed to use the most recent library versions (while maintaining compatibility) to improve efficiency and reduce errors. At time of writing, the most recent version of Python that supports the used libraries was used: `Python 3.10`. The most important libraries are: `Gin-config 0.5.0`, `Pytorch 2.0` (in combination with Cuda 11.8.0), `Pytorch-lightning 2.0`, `Scikit-learn 1.2.2`, `Lightgbm 3.3.5`, `Pandas 2.0.0`.

### G.3 COMPLEXITY

As mentioned in Yèche et al. (2022), the transformer memory complexity concerning the sequence length is quadratic. With our hardware, training deep learning models overall takes less than 3 hours. Training ml models takes less than 1 hour. See Table 21 for average training durations. Note that there was large variability between the cohorts due to the size difference of datasets (i.e., eICU Collaborative Research Database contains almost eight times the amount of stays as AmsterdamUMCdb). The algorithmic complexity is specified in the implementation details of the Scikit-learn (Pedregosa et al., 2011) (logistic regression (LR) and elastic net (EN)), LightGBM (Ke et al., 2017) (Light Gradient Boosting Machine (LGBM)), and PyTorch library (Falcon & team, 2023) (Long Short-Term Memory (LSTM), Recurrent Neural Network (RNN), Gated Recurrent Unit

CODE LISTING 15: *Metrics recorded for binary classification.* Adding the metrics to this dictionary results in automatic logging. See the whole file for more details.

```
class DLMetrics:
    BINARY_CLASSIFICATION = {
        "AUC": AUROC(task="binary"),
        "PR" : AveragePrecision(task="binary"),
        "F1": F1Score(task="binary", num_classes=2),
        "Calibration_Error": CalibrationError(task="binary",n_bins=10)
        "Calibration_Curve": CalibrationCurve,
        "PR_Curve": PrecisionRecallCurve,
        "RO_Curve": RocCurve,
        "JSD": JSD,
        "Binary_Fairness": BinaryFairnessWrapper(num_groups=2, task='demographic_parity', group_name="sex")
        }
```

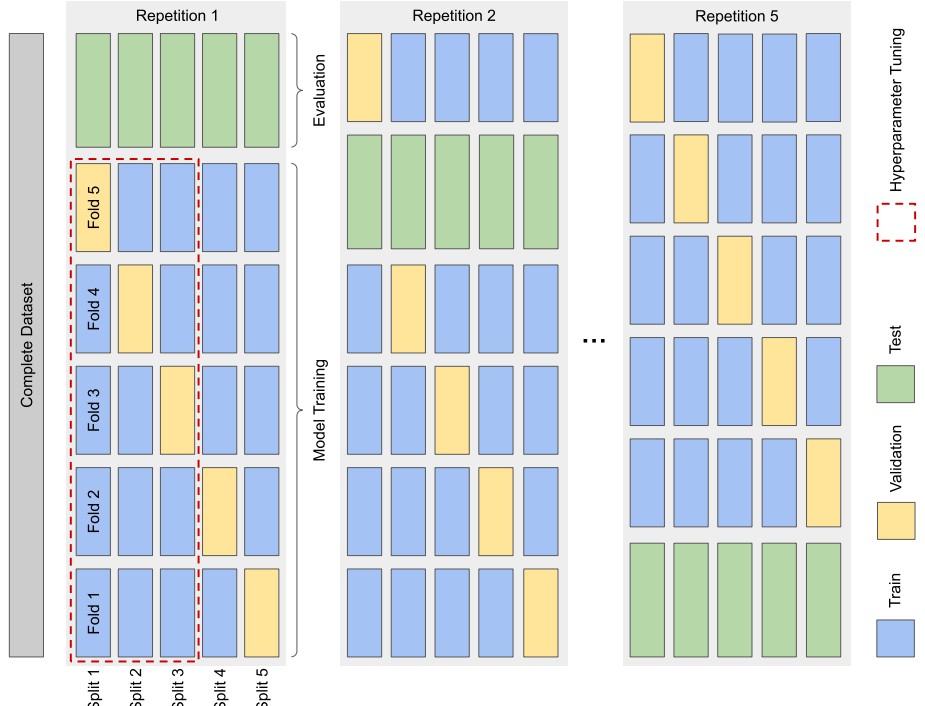

FIGURE 11: *Schematic overview of 5 times repeated 5-fold cross-validation.* This example uses only the first three splits of the first repetition for hyperparameter tuning. The repetitions and amount of folds and the folds to tune on can be easily adjusted (see Code Listing 3).

(GRU)). Furthermore, Bai et al. (2018) (Temporal Convolutional Network (TCN)) and (Yèche et al., 2022) (transformer) are implementations are also detailed in their respective works. Figure 11 shows our repeated cross-validation training method. Note that in order to obtain our final results, we do 5 repetitions of 5 fold cross-validation with an excluded test set.

## G.4 REPRODUCIBLITY

For a detailed description to reproduce our experiments, we refer to the Paper reproducibility file (included in the repository). We have followed the standards specified by official ML reproducibility guidelines by Papers with Code[16].

---

[16]https://github.com/paperswithcode/releasing-research-code

### G.5 EXTERNAL VALIDATION

All external validation validation models have been trained with an 80/20 train/val split to use as much of the dataset as possible. The test splits are the same as used for the same dataset experiments (diagonal in Figure 2) For the external validation pooled results (d-1), we used subsets of 10,000 stays for each dataset to simulate a setting where the datasets have a similar sample size. We wanted to ensure the size of the datasets, which differs significantly between datasets, had no undue influence on the training.

### G.6 FINE-TUNING

For our fine-tuning experiment, shown in Figure 3 we used an ADAM optimizer with a starting learning rate of 0.00001 and an exponential learning rate scheduler to reduce learning rates gradually. The rest of the hyperparameters are exactly as in the original source models.

# H  HYPERPARAMETERS

Here, we detail the tuning setup and hyperparameters used in our experiments.

## H.1  TUNING APPROACH

In Table 22 and 23, we specify the hyperparameters used for hyperparameter-tuning for the baseline experiments for deep learning and machine learning models, respectively. We incorporated different sampling methods for hyperparameter selection. Hyperparameters were chosen to be mostly identical to Yèche et al. (2022), to improve comparability and for reproducibility reasons. However, we have chosen to allow for continuous ranges of hyperparameters in some cases, to improve the performance and functionality of *YAIB*. Log-uniform means that the parameters are sampled according to the

TABLE 22: *Model hyperparameters, default values for all models (above), and the distributions for the DL models (below), considered during Bayesian hyperparameter optimization.*

| Category | Parameter | Value |
|---|---|---|
| Hyperparameter search | Initial points | 5 |
| | Calls | 30 |
| | Folds to tune on | 2 |
| General Parameters | Epochs | 1000 |
| | Min delta | 1E-4 |
| | Patience | 10 |
| | Batch size | 64 |
| | Weight | Balanced |
| Loss | Regression task | Mean Squared Error |
| | Classification task | Cross-entropy |
| Optimizer Parameters | Optimizer | Adam |
| | Weight decay | 1e-6 |
| | Learning rate | Uniform([1E-5, 3E-4]) |

| Model | Parameter | Value |
|---|---|---|
| LSTM | Hidden dimension | Log-uniform([32, 128]) |
| | Hidden dimension | RandomInt(1, 3) |
| | Dropout probability | Uniform([0.0, 0.4]) |
| GRU | Hidden dimension | Log-uniform([32, 128]) |
| | Number of layers | RandomInt(1, 3) |
| TCN | Hidden dimension | Log-uniform([32, 128]) |
| | Number of layers | Log-uniform([32, 256]) |
| | Kernel size | Log-uniform([2, 32]) |
| | Horizon | 24 |
| Transformer | Hidden dimension | Log-Uniform([32, 128]) |
| | Number of layers | RandomInt(1, 10) |
| | Number of heads | Log-uniform(1, 8) |
| | Depth | Uniform([1,3]) |
| | Kernel size | Log-uniform([2, 32]) |
| | Dropout probability | Uniform([0.0, 0.4]) |
| | Dropout attention | Uniform([0, 0.4]) |
| | L1 regularization | 0.0 |
| | Hidden multiplication | 2 |

reciprocal distribution:

$$f(x; a, b) = \frac{1}{x[\log_e(b) - \log_e(a)]} \quad \text{for } a \leq x \leq b \text{ and } a > 0. \tag{1}$$

Uniform means that the parameters are sampled according to the uniform distribution:

$$f(x) = \begin{cases} \frac{1}{b-a} & \text{for } a \leq x \leq b, \\ 0 & \text{for } x < a \text{ or } x > b \end{cases} \tag{2}$$

TABLE 23: *Model hyperparameters, default values for all models (above), and distributions for the ML models (below), considered during Bayesian hyperparameter optimization.*

| Category | Parameter | Value |
|---|---|---|
| Hyperparameter search | Initial points | 10 |
| | Calls | 50 |
| | Folds to tune on | 3 |
| General Parameters | Patience | 10 |
| | Jobs | 8 |
| Loss | Regression task | Logloss |
| | Classification task | Cross-entropy |

| Model | Parameter | Value |
|---|---|---|
| LR | C | Log-uniform([1E-3, 1E1]) |
| | Penalty | Choice(l1, l2, elasticnet) |
| | L1 Ratio | Uniform([0.0, 1.0]) |
| | Solver | saga |
| | Max iterations | 100000 |
| EN | Alpha | Log-uniform([1E-2, 1E1]) |
| | Tol | Log-uniform([1E-5, 1E-1]) |
| | Hidden dimension | RandomInt(1, 3) |
| | Dropout probability | Uniform([0.0, 0.4]) |
| | L1 Ratio | Uniform([0.0, 1.0]) |
| | Solver | saga |
| | Max iterations | 10000 |
| LGBM | Column sample | Uniform([0.33, 1.0]) |
| | Sub sample | Uniform([0.33, 1.0]) |
| | Leaves | Log-uniform([8, 128]) |
| | Max depth | RandomInt(3, 7) |
| | Hidden dimension | RandomInt(1, 3) |
| | Estimators | 10000 |
| | Min child samples | 1000 |
| | Subsample frequency | 1 |

## H.2 DEEP LEARNING MODELS

We detail the hyperparameters that have been chosen using our Bayesian hyperparameter optimization approach.

**Gated Recurrent Unit (GRU)**    The range of hyperparameters considered for the GRU model are found in Table 24.

TABLE 24: *Chosen hyperparameters for Gated Recurrent Unit (GRU).*

| Dataset | Learning Rate | Layer Dimension | Hidden Dimension |
|---|---|---|---|
| **Mortality** | | | |
| AUMC | 3.00E-04 | 3 | 48 |
| HiRID | 2.37E-04 | 2 | 52 |
| eICU | 3.00E-04 | 1 | 135 |
| MIMIC-IV | 1.43E-04 | 2 | 77 |
| **AKI** | | | |
| AUMC | 2.81E-04 | 3 | 256 |
| HiRID | 2.06E-04 | 3 | 115 |
| eICU | 1.43E-04 | 3 | 240 |
| MIMIC-IV | 2.82E-04 | 3 | 139 |
| **Sepsis** | | | |
| AUMC | 2.28E-04 | 2 | 77 |
| HiRID | 3.00E-04 | 3 | 59 |
| eICU | 8.51E-05 | 2 | 77 |
| MIMIC-IV | 2.39E-04 | 3 | 52 |
| **KF** | | | |
| AUMC | 3.00E-04 | 3 | 93 |
| HiRID | 1.11E-04 | 3 | 196 |
| eICU | 6.35E-05 | 3 | 196 |
| MIMIC-IV | 1.11E-04 | 1 | 148 |
| **LoS** | | | |
| AUMC | 5.45E-05 | 3 | 158 |
| HiRID | 1.00E-05 | 1 | 117 |
| eICU | 1.03E-05 | 3 | 254 |
| MIMIC-IV | 1.57E-05 | 2 | 237 |

**Long Short-Term Memory (LSTM)** The range of hyperparameters considered for the LSTM model are found in Table 25.

TABLE 25: *Chosen hyperparameters for Long Short-Term Memory (LSTM).*

| Dataset | Learning Rate | Layer Dimension | Hidden Dimension |
|---|---|---|---|
| **Mortality** | | | |
| AUMC | 1.87E-04 | 1 | 145 |
| HiRID | 1.54E-04 | 2 | 256 |
| eICU | 3.00E-04 | 3 | 149 |
| MIMIC-IV | 3.00E-04 | 2 | 185 |
| **AKI** | | | |
| AUMC | 2.62E-04 | 3 | 57 |
| HiRID | 3.00E-04 | 3 | 54 |
| eICU | 3.00E-04 | 3 | 70 |
| MIMIC-IV | 3.00E-04 | 3 | 256 |
| **Sepsis** | | | |
| AUMC | 2.10E-04 | 1 | 153 |
| HiRID | 2.48E-04 | 1 | 139 |
| eICU | 1.12E-04 | 2 | 40 |
| MIMIC-IV | 2.46E-04 | 1 | 161 |
| **KF** | | | |
| AUMC | 2.79E-04 | 2 | 81 |
| HiRID | 3.00E-04 | 1 | 256 |
| eICU | 1.75E-04 | 3 | 33 |
| MIMIC-IV | 2.49E-04 | 1 | 256 |
| **LoS** | | | |
| AUMC | 3.24E-05 | 1 | 62 |
| HiRID | 6.65E-05 | 3 | 255 |
| eICU | 2.86E-05 | 3 | 215 |
| MIMIC-IV | 1.80E-05 | 3 | 253 |

**Temporal Convolutional Network (TCN)** The range of hyperparameters considered for the TCN model are found in Table 26.

TABLE 26: *Chosen hyperparameters for Temporal Convolutional Network (TCN).*

| Dataset | Learning Rate | Dropout | Kernel | Number of Channels |
|---|---|---|---|---|
| **Mortality** | | | | |
| AUMC | 5.84E-05 | 2.71E-01 | 6 | 92 |
| HiRID | 2.14E-04 | 1.12E-01 | 6 | 80 |
| eICU | 2.35E-05 | 1.15E-02 | 23 | 100 |
| MIMIC-IV | 1.81E-05 | 3.50E-01 | 3 | 130 |
| **AKI** | | | | |
| AUMC | 3.00E-04 | 4.00E-01 | 3 | 144 |
| HiRID | 2.56E-04 | 2.56E-01 | 12 | 168 |
| eICU | 3.00E-04 | 1.23E-01 | 3 | 81 |
| MIMIC-IV | 2.98E-04 | 7.61E-02 | 3 | 249 |
| **Sepsis** | | | | |
| AUMC | 3.00E-04 | 0.00E+00 | 2 | 32 |
| HiRID | 3.00E-04 | 0.00E+00 | 2 | 256 |
| eICU | 1.93E-04 | 3.98E-01 | 2 | 61 |
| MIMIC-IV | 2.23E-04 | 1.06E-01 | 4 | 78 |
| **KF** | | | | |
| AUMC | 2.56E-04 | 2.78E-01 | 6 | 169 |
| HiRID | 1.75E-04 | 2.33E-01 | 3 | 34 |
| eICU | 2.15E-04 | 1.92E-01 | 5 | 138 |
| MIMIC-IV | 1.88E-05 | 2.15E-01 | 2 | 33 |
| **LoS** | | | | |
| AUMC | 3.00E-04 | 1.92E-01 | 2 | 32 |
| HiRID | 2.74E-04 | 1.71E-01 | 29 | 43 |
| eICU | 3.00E-04 | 2.91E-01 | 10 | 32 |
| MIMIC-IV | 1.57E-04 | 1.67E-01 | 12 | 44 |

**Transformer** The range of hyperparameters considered for the transformer model are found in Table 27.

TABLE 27: *Chosen hyperparameters for transformer.*

| Dataset | Learning Rate | Dropout | Heads | Hidden Dimension | Depth |
|---|---|---|---|---|---|
| **Mortality** | | | | | |
| AUMC | 1.29E-04 | 1.32E-01 | 2 | 95 | 2 |
| HiRID | 1.58E-04 | 0.00E+00 | 1 | 247 | 1 |
| eICU | 1.00E-05 | 4.00E-01 | 3 | 256 | 1 |
| MIMIC-IV | 6.18E-05 | 1.65E-01 | 1 | 48 | 3 |
| **AKI** | | | | | |
| AUMC | 1.18E-04 | 9.75E-03 | 8 | 52 | 3 |
| HiRID | 3.00E-04 | 1.50E-01 | 1 | 154 | 2 |
| eICU | 1.28E-04 | 1.33E-01 | 2 | 96 | 2 |
| MIMIC-IV | 1.22E-04 | 4.13E-02 | 2 | 72 | 3 |
| **Sepsis** | | | | | |
| AUMC | 2.61E-04 | 4.39E-02 | 1 | 32 | 3 |
| HiRID | 3.00E-04 | 0.00E+00 | 1 | 32 | 1 |
| eICU | 2.76E-05 | 1.05E-02 | 2 | 211 | 2 |
| MIMIC-IV | 3.53E-05 | 3.67E-01 | 1 | 98 | 3 |
| **KF** | | | | | |
| AUMC | 1.95E-04 | 4.56E-02 | 1 | 51 | 2 |
| HiRID | 2.48E-04 | 9.34E-02 | 7 | 160 | 3 |
| eICU | 2.62E-04 | 2.82E-02 | 1 | 52 | 1 |
| MIMIC-IV | 1.53E-04 | 8.62E-02 | 5 | 160 | 2 |
| **LoS** | | | | | |
| AUMC | 1.13E-04 | 4.11E-02 | 3 | 76 | 2 |
| HiRID | 1.88E-05 | 1.88E-05 | 4 | 102 | 1 |
| eICU | 3.96E-05 | 5.67E-02 | 3 | 172 | 2 |
| MIMIC-IV | 1.13E-04 | 4.11E-02 | 3 | 76 | 2 |

## H.3 MACHINE LEARNING MODELS

**Logistic regression (LR)**   The range of hyperparameters considered for the LR model are found in Table 28.

TABLE 28: *Chosen hyperparameters for logistic regression (LR).*

| Dataset | C | Penalty | L1 Ratio |
|---|---|---|---|
| **Mortality** | | | |
| AUMC | 3.63E-02 | elasticnet | 1.00E+00 |
| HiRID | 3.45E-02 | l2 | 6.63E-02 |
| eICU | 2.78E-02 | elasticnet | 1.00E+00 |
| MIMIC-IV | 2.05E-01 | elasticnet | 1.00E+00 |
| **AKI** | | | |
| AUMC | 1.77E-02 | l1 | 1.00E+00 |
| HiRID | 1.00E+01 | l1 | 4.16E-01 |
| eICU | 2.52E-02 | l1 | 5.99E-01 |
| MIMIC-IV | 1.28E-01 | l1 | 2.98E-01 |
| **Sepsis** | | | |
| AUMC | 4.87E-02 | l1 | 2.21E-01 |
| HiRID | 2.59E-03 | l2 | 3.74E-01 |
| eICU | 1.98E-03 | elasticnet | 6.20E-01 |
| MIMIC-IV | 2.20E-03 | l1 | 1.81E-02 |

**Elastic net (EN)**   The range of hyperparameters considered for the EN model are found in Table 29.

TABLE 29: *Chosen hyperparameters for elastic net (EN).*

| Dataset | Alpha | Tol | L1 Ratio |
|---|---|---|---|
| **KF** | | | |
| AUMC | 1.04E-02 | 3.60E-02 | 1.22E-03 |
| HiRID | 1.04E-02 | 3.60E-02 | 1.22E-03 |
| eICU | 1.05E-02 | 9.57E-04 | 8.09E-03 |
| MIMIC-IV | 1.00E-02 | 2.45E-03 | 0.00E+00 |
| **LoS** | | | |
| AUMC | 1.00E-02 | 1.67E-05 | 7.66E-02 |
| HiRID | 1.00E-02 | 1.02E-05 | 2.40E-02 |
| eICU | 1.00E-02 | 4.82E-02 | 0.00E+00 |
| MIMIC-IV | 1.00E-02 | 7.42E-02 | 0.00E+00 |

**Light Gradient Boosting Machine (LGBM)** The range of hyperparameters considered for the LGBM model are found in Table 30.

TABLE 30: *Chosen hyperparameters for Light Gradient Boosting Machine (LGBM).*

| Dataset | Depth | Column Sample | Leaves | Subsample |
|---|---|---|---|---|
| **Mortality** | | | | |
| AUMC | 6 | 9.89E-01 | 117 | 9.90E-01 |
| HiRID | 5 | 1.00E+00 | 8 | 1.00E+00 |
| eICU | 7 | 5.41E-01 | 128 | 1.00E+00 |
| MIMIC-IV | 7 | 1.00E+00 | 28 | 1.00E+00 |
| **AKI** | | | | |
| AUMC | 7 | 1.00E+00 | 110 | 8.69E-01 |
| HiRID | 7 | 1.00E+00 | 79 | 8.69E-01 |
| eICU | 7 | 9.97E-01 | 117 | 8.35E-01 |
| MIMIC-IV | 7 | 1.00E+00 | 128 | 1.00E+00 |
| **Sepsis** | | | | |
| AUMC | 7 | 1.00E+00 | 128 | 7.83E-01 |
| HiRID | 4 | 6.18E-01 | 17 | 8.22E-01 |
| eICU | 7 | 8.97E-01 | 52 | 6.04E-01 |
| MIMIC-IV | 7 | 1.00E+00 | 128 | 5.84E-01 |
| **KF** | | | | |
| AUMC | 7 | 7.01E-01 | 15 | 9.99E-01 |
| HiRID | 3 | 9.58E-01 | 16 | 9.95E-01 |
| eICU | 7 | 1.00E+00 | 47 | 1.00E+00 |
| MIMIC-IV | 7 | 9.30E-01 | 33 | 9.96E-01 |
| **LoS** | | | | |
| AUMC | 7 | 3.30E-01 | 83 | 1.00E+00 |
| HiRID | 6 | 3.30E-01 | 49 | 5.01E-01 |
| eICU | 7 | 3.30E-01 | 51 | 3.30E-01 |
| MIMIC-IV | 7 | 3.30E-01 | 128 | 3.30E-01 |

