# OpenReview forum: "Yet Another ICU Benchmark: A Flexible Multi-Center Framework for Clinical ML"
_ICLR.cc/2024/Conference — ICLR 2024 poster_

### Official Review · Reviewer_5BSM · 2023-10-24

**Soundness:** 2 fair
**Presentation:** 2 fair
**Contribution:** 3 good
**Rating:** 6
**Confidence:** 3

**Summary:**

The paper presents a multi-dataset framework that consists of preprocessing and processing techniques to aid reproducible research with consistent data and experiment methods. The authors demonstrate that the experiment choices can have a substantial impact on model prediction performance.

**Strengths:**

I commend the authors for their hard work to create an open source code repository, which is not a trivial task. Based on the description in the paper, this repository could have substantial impact for streamlining research with ICU datasets.

The paper is very easy to follow and organized well. Thorough descriptions of datasets and code are included.

Paper includes example scripts in the appendix to demonstrate code simplicity and example use cases.

Table 1 presents an excellent overview and comparison of the proposed methods and prior work.

The paper demonstrates use of the proposed framework with four prominent ICU datasets in ML literature. Additionally the datasets cover a range of populations including those in US and Europe, which can promote research that is applicable across different populations.

**Weaknesses:**

1. The baseline table is useful for future work to compare against. However, this table does not seem relevant to motivating the proposed framework and could be included in the appendix. To better motivate the value of the proposed unifying framework, it would be useful to show more experiments regarding significant differences in model performance when different experiment methods are used (outcome labels, cohort selection, etc.). For example, you could consider an ablation study to show that model performance significantly improves or decreases based on various outcome definitions for datasets with consistent preprocessing, model choice, hyperparameters. The experiment mentioned in the paper regarding exclusion criteria would also be a good example to elaborate on and include table results comparing impact of different exclusion criteria.

2. Be specific about performance gains and losses (example: improved by XX%) and their statistical significance. A significant difference in model performance due to unaligned data preprocessing and processing pipeline would help motivate the proposed unifying framework. Stating that “differences were bigger for some datasets than others” is not a robust explanation.

3. Novel ML research often proposes a new model that is evaluated across multiple datasets. Enabling support for new models to be easily integrated into this framework is important. The appendix provides example code for training a specific model, such as those in scikit-learn or implemented in pytorch. Including more thorough explanation and/or code examples for how to train a novel/custom model, such as pytorch-based model implemented in a separate python file, would be useful.

4. The author mentions flexibility of the repo in allowing users to configure data preprocessing in a streamlined fashion across multiple datasets. However, does this somewhat deflate the original purpose to enable reproducible research through consistent dataset preprocessing and processing. Researchers may still conduct research with various preprocessing steps that differ across literature. The value of this framework hinges on if research exactly states the detailed use of YAIB in their paper/methods AND if future work follows these exact same steps. It would be useful if the author's address this point and potentially expand their unifying framework to include recommended approaches for how to reference/cite use of YAIB in future work that could promote reproducible research.

The low score is mostly because of points 1-3. If these points are addressed I would be happy to raise the score.

**Questions:**

1. It is possible that thresholds used to define outcomes from time series data may update over time based on new knowledge, thus impacting outcome labels. Does the repo have support to enable users to define their own outcome definitions across datasets? For example, a patient AKI outcome may occur when patient creatinine falls below a specific threshold. Or are outcome labels strictly based on definitions included in the datasets?

2. Are models other than those in scikit-learn hardcoded into the repo or are there wrappers for these as well? Is there support for users to include and evaluate new/customized model architectures in the pipeline?

3. Is there a reason why this framework is limited to ICU/healthcare settings and not time series in general. I could see a pipeline like this being applicable to time series from many domains. It may be worth elaborating on why the framework is ICU-specific. It may also be useful to compare YAIB to similar non-healthcare related frameworks.

---

> ### Author Response · Authors · 2023-11-16
> **Response to reviewer 5BSM [1/4]**
>
> We thank the reviewer for their feedback to improve the paper and the provided opportunity to raise the score.
>
>
>
> **Baseline table**
>
> We acknowledge the possibility of more experiments and thank the reviewer for providing examples. Unfortunately, we have had to be selective in the experiments we included. While writing the paper, we performed several experiments, that partially ended up in the appendix due to the lack of space. The appendix contains results for 1) the use of dynamic feature generation, 2) excluding static features, 3) out-of-the-box results with demo cohorts provided with our repository, 4) Length of Stay external validation, and 5) AUPRC results for the fine-tuning experiment in the main text.
>
>
>
> Additionally, we recognize that many experiments can be performed in addition to the baseline. However, we believe one of our main contributions is demonstrating a baseline of different tasks across datasets and models to compare their work. We, therefore, respectfully disagree with the reviewer that a table with baseline results across models, tasks, and datasets does not motivate our framework. Future work can directly utilize these results to get an understanding for the performance of their model, dataset, preprocessing pipeline, or clinical task. Furthermore, including it in the main text is important for interpreting our work as a reference paper. If we were to include it in the appendix, we risk misinterpretation of our work, whereas we want to communicate the goal of YAIB as a baseline as clearly as possible.
>
>
>
> **Performance gains and losses**
>
> We thank the reviewer for pointing out the comment about performance gains and losses but are not exactly sure what the reviewer means. In the mentioned section, we have removed the sentence mentioned and added percentage-wise difference ranges when comparing approaches against each other. We also provide standard deviations for each result.  We hope this addresses the reviewer's comment.
>
>
>
> **Integration of Novel ML**
>
> The reviewer would like additional explanation on the ease of integrating novel ML. As the main paper claims, adding new models to YAIB is straightforward. We provide examples in Appendix E.2 on extending YAIB concerning datasets, concepts, tasks, models, and metrics to demonstrate this. Among these examples is a Temporal Fusion Transformer ([https://arxiv.org/abs/1912.09363](https://arxiv.org/abs/1912.09363)), which may be considered state-of-the-art and was recently added by a benchmark user.
>
>
>
> We additionally refer to the work added to the supplementary materials (Closing Gaps: An Imputation Analysis of ICU Vital Signs) which has been accepted to the First Workshop on Generative Modelling for Health at NeurIPS 2023 and uses YAIB as a base.
>
>
>
> To accommodate the reviewers' comments, we have added three files; these are *rnn.gin, rnn.py, and instructions.md* in the folder *Adding a model* of the supplemental material. to the supplemental material to demonstrate the ease of adding an RNN model. Additionally, we have added a wiki to show how to add new models; this usually comes down to adding just two files (model specification and hyperparameter bindings). Finally, Appendix F.3 shows how users can transparently share new setups with YAIB as core.

---

> ### Author Response · Authors · 2023-11-16
> **Response to reviewer 5BSM (continued) [2/4]**
>
> **Flexibility or tradeoff of extensibility and reproducibility**
>
> The reviewer argues that providing extensibility could sacrifice the reproducibility of YAIB. We provide an out-of-the-box end-to-end pipeline that can be used for benchmarking. However, many users would like to tweak aspects of this pipeline to create novel benchmarks. We allow for this use case as well a “pure” benchmarking against a new method without changing any of the code.  We acknowledge that we can never create a benchmark that fits the requirements of all users going forward.
>
>
>
> We designed YAIB to be as extensible as possible while retaining full reproducibility. This means easy support of new databases, clinical concepts, tasks, experiment configurations, preprocessing pipelines, imputation methods, models, and evaluation metrics. If changes are necessary, they need to be reproducible and easily shareable across research teams.
>
> If the user only requires a few default ICU tasks from a single i.i.d. dataset to test their new method, any existing ICU benchmarks could be sufficient. Users do not need to apply for access to multiple datasets and do not have to deal with the intricacies of the clinical task definition. As long as the integration of a new model is seamless, such simple frameworks are fit-for-purpose and abstract much of the complexity, allowing the user to only worry about one thing: their model. If multiple papers used the exact same benchmark, results are also directly comparable between papers (an “apples-to-apples” comparison).
>
>
>
> However, we found that this setup tends to be too restrictive and thus unrealistic. Users often want to highlight a particular aspect of their model, prompting them to adapt to the default task. At other times, they want to show clinical impact and need to adapt the default task to make it more realistic. Given the lack of successful translation of prediction models into clinical practice, reviewers are also increasingly requesting external validation – sometimes with multiple endpoints – which is difficult to shoehorn into most existing solutions.
>
> YAIB embraces the need to tweak experimental setups. Results will no longer be directly comparable between papers, but we argue that actual apples-to-apples comparisons were inherently rare. Instead of forcing users into a rigid framework, it allows for adaptations but requires them to be done transparently. Absolute performance should be compared only within the same paper or among papers with the same task setup (see our examples in Tables 5 and 6).
>
>
>
> To facilitate the transparency of adaptations, we rely on a sophisticated framework to define clinical concepts across multiple datasets (ricu). We have adapted and extended ricu to provide a standard workflow for YAIB to integrate new databases and define new clinical concepts. To date, it has been successfully used to bring 4/5 +1 ICU datasets into a common format (including our addition of the Salzburg Intensive Care Database, which is currently in quality control). This approach is flexible enough that we have yet to encounter significant restrictions in mapping admissions, demographics, vital signs, laboratory values, medication (including rates and durations), clinical scores, and outcomes at different time scales across datasets. The main restriction of ricu is that it is currently implemented in the R language only, but we provide guidance on how to access it via rpy2, and we are in the process of porting it to Python; this will make our pipeline even more accessible, especially to clinical researchers. Our cohort definition functionality provides helper functions to apply inclusion/exclusion criteria on top of ricu and report step-by-step attrition numbers. The cohorts can be used in a modular fashion with custom preprocessing steps, imputations, prediction models, and evaluation metrics, all using the exact same code across multiple datasets.
>
>
>
> Even so, there will likely be situations where the user may be better off with a custom solution. We expect this to occur once their use case diverges significantly from standard supervised learning. For example, federated learning or reinforcement learning setups may require significantly different training and evaluation loops. These are not currently supported, but we consider this as future work. In any case, the user can still use our data processing, cohort generation, and possibly other parts of YAIB (e.g., by exchanging the default training module with a custom module). Authors using YAIB should, therefore, provide their code and a detailed list of the changes they have made to the repository; modern version control allows us to verify this against the original YAIB repository easily. The newly added Appendix F.3 addresses the exact steps to provide transparency for novel work.

---

> ### Author Response · Authors · 2023-11-16
> **Response to reviewer 5BSM (continued) [3/4]**
>
> **Flexibility or tradeoff of extensibility and reproducibility (continued)**
>
> The YAIB pipeline has helped us to produce reproducible results quickly and provides the required extensibility for our purposes. We are in touch with some researchers who have used YAIB to date and provided feedback, although mainly in an informal way. We refer to the work “Closing Gaps: An Imputation Analysis of ICU Vital Signs” accepted to the 1st Workshop on Deep Generative Models for Health at NeurIPS 2023 in the supplemental material. The authors of this work used YAIB as a bedrock for implementing imputation methods and are in the process of extending this to more methods and downstream tasks. For concrete examples of how to extend YAIB, for other authors, we refer to Appendix D and the wiki documentation we added to the supplemental materials.
>
> **Using YAIB in future work**
>
>
>
> Authors using YAIB should provide their code and a detailed list of the changes they have made to the repository if they use it in their work; modern version control allows us to verify this against the original YAIB repository easily. To address the reviewer's concerns, we have added appendix F.3, which describes exactly which files have to be provided if authors make changes to the existing pipeline. We aim to provide a guide to produce transparent research with YAIB, without limiting authors the existing pipeline.
>
>
>
> Again, we provide detailed documentation in the wiki, which has been added to the supplementary materials (Example-usage.md). Moreover, in the experiments folder of the main repository, YML config files can be used to reproduce the experiments in the paper using the convenient Weights and Biases experiment tracking software; instructions according to the papers with code template can be found in the file PAPER.md in the root of the main repository. We hope this addresses the reviewer's issue regarding extending YAIB.
>
>
>
> ## Questions
>
> **Support for defining new outcomes**
>
> As we use the ricu framework, one can define new tasks on the basis of the existing data. This means that we can redefine thresholds and generate new cohorts that reflect a different clinical prediction task. We have added a YAIB wiki to show the process of adding new tasks with the updated supplemental material. Additionally, we refer to the ricu user guide ([https://eth-mds.github.io/ricu/](https://eth-mds.github.io/ricu/)) and Appendix E.3. We note that most outcomes are not a label in the data; they are a complex combination of events. For example, for sepsis use of it is the use of antibiotics while there is a suspicion of infection (for details see YAIB-cohorts/R/sepsis.R in the supplementary materials).
>
> **Adding new models**
>
> We provide extensive support for adding new models as this is one of the main use cases of YAIB. Both ML and DL model have one common wrapper from which they inherent. We have a separate wrapper for deep learning models as they have the mechanics of a neural network (requiring e.g., a forward method). Adding a model and a new dataloader can be done in a straightforward manner by subclassing them.  We refer to the newly added wiki in the supplementary materials (Adding-a-new-model.md) and Appendix E.4.

---

> ### Author Response · Authors · 2023-11-16
> **Response to reviewer 5BSM (continued) [4/4]**
>
> **Generalizability to other domains**
>
> Healthcare is a complicated application area of ML. We see this in the lack of adoption of ML in this domain [1,2]. The complete end-to-end pipeline that YAIB provides has been designed for temporal ICU data. The data harmonization and cohort creation are currently specific to ICU data. We acknowledge that extending the framework to general EHR data and other time-series data can be straightforward in some cases. However, we believe that is infeasible to achieve a completely domain-agnostic pipeline; we have instead focused on the end-to-end pipeline that is able to unite both clinicians and ML researchers. One of our main motivations is to solve the lack of reproducibility in clinical ML [1,2], which impedes clinical studies with ML models.  Moreover, given the specific use cases for YAIB, we believe that it is not useful to compare to more works than the 13 works in Table 1.
>
>
>
> *[1] Lucas M. Fleuren, Patrick Thoral, Duncan Shillan, Ari Ercole, Paul W. G. Elbers, Mark Hoogendoorn, Ben Gibbison, Thomas L. T. Klausch, Tingjie Guo, Luca F. Roggeveen, Eleonora L. Swart, Armand R. J. Girbes, and Right Data Right Now Collaborators. Machine learning in intensive care medicine: Ready for take-off? Intensive Care Medicine, 46(7):1486–1488, July 2020b. ISSN 1432-1238.doi: 10.1007/s00134-020-06045-y.*
>
> *[2] Bar Eini-Porat, Ofra Amir, Danny Eytan, and Uri Shalit. Tell me something interesting: Clinical utility of machine learning prediction models in the ICU. Journal of Biomedical Informatics, 132: 104107, August 2022. ISSN 15320464. doi: 10.1016/j.jbi.2022.104107.*
>
> *[3] Tabinda Sarwar, Sattar Seifollahi, Jeffrey Chan, Xiuzhen Zhang, Vural Aksakalli, Irene Hudson, Karin Verspoor, and Lawrence Cavedon. The Secondary Use of Electronic Health Records for Data Mining: Data Characteristics and Challenges. ACM Computing Surveys, 55(2):1–40, March 2023. ISSN 0360-0300, 1557-7341. doi: 10.1145/3490234.*
>
> *[4] Christopher J. Kelly, Alan Karthikesalingam, Mustafa Suleyman, Greg Corrado, and Dominic King. Key challenges for delivering clinical impact with artificial intelligence. BMC Medicine, 17(1): 195, December 2019. ISSN 1741-7015. doi: 10.1186/s12916-019-1426-2.*
>
>
>
>
> **Proposed actions:**
>
> - Added a wiki that provides a low barrier of entry to using and extending YAIB.
>
> - Added sentence to the design philosophy that indicates that users of YAIB should openly provide their methodology and code and compare it with the original benchmark.
>
> - Added standalone Python files in the folder docs/adding_model to demonstrate adding a model to YAIB. Adding just two short files allows us to use a new model in most cases. For other models, we have included instructions in the wiki (Adding-a-new-model.md).
>
> - Added Appendix F.3 which serves as a guideline for users to use YAIB in their work to benchmark new models (or other parts of the pipeline).
>
> - Adjusted the description of the experiments to remove vague claims and use percentage-wise differences where appropriate.
>
>
> We hope this rebuttal addresses the reviewer's points. We invite them to respond with additional points that could be used to improve our work further.

---

> > ### Comment · Reviewer_5BSM · 2023-11-20
> >
> > Thank you for the detailed response. Given that the authors have addressed the comments and questions, I have raised my score. I appreciate the authors' addition of percentage-wise difference ranges when comparing approaches against each other. The proposed wiki that outlines YAIB is an excellent extension to facilitate use of the repo.
> >
> > To address the authors questions regarding clarifying performance gains and losses:
> > To state that the computed metrics are “higher” or “lower” is does not provide sufficient context for whether the results are meaningful. Instead state the exact percentage differences (ex:, there was an X% difference in AUROC for two preprocessing methods). Stating whether the results are statistically significant based on hypothesis testing or standard deviations is also general practice. The word 'significant' is used several times in the paper; I would reserve use of this word for statistical significance and provide evidence/explanation to verify this claim.

---

> ### Author Response · Authors · 2023-11-21
> **Second response to Reviewer 5BSM**
>
> Thank you for the feedback and the increased score. We agree with the reviewer on both points and will incorporate this in our manuscript, potentially bundled with further improvements.
>
> For the next manuscript version, we will provide more percentage-wise differences where we think they can be helpful to provide relative performance. We do not see the use of statistical tests in most cases as we report the standard deviations.
>
> We agree with the second point and will replace the word significant when discussing results to avoid confusion with statistical significance.

---

### Official Review · Reviewer_Q5t5 · 2023-10-28

**Soundness:** 4 excellent
**Presentation:** 4 excellent
**Contribution:** 4 excellent
**Rating:** 8
**Confidence:** 4

**Summary:**

The authors present an innovative framework that facilitates reproducible and comparable machine learning experiments using publicly available ICU datasets. This framework offers: i) thorough and standardized definitions of cohorts and feature extractions, ii) Transparent feature engineering and preprocessing, iii) commonly compared deep learning baselines (including RNNs, TCN, Transformer) with flexibility in the network architecture and hyper-parameters optimization, and iv) performance metrics. This work addresses the growing demand for a standardized framework for comparing and reproducing widely used machine learning and deep learning benchmarks. Moreover, this framework can be further utilized to provide a standardized way of evaluating newly developed DL methods. This work will be extremely helpful in opening the door for wider adoption of recent deep learning methods in real clinical practices.

**Strengths:**

1.	The paper is well-written.
2.	The authors provide a very solid framework, covering from defining clinically relevant cohorts in ICU admissions to training and evaluating ML/DL methods. Moreover, this framework has the flexibility to incorporate user-specific DL networks which will be extremely useful in providing comparable and reproducible evaluations.
3.	The well-defined cohorts and harmonized datasets hold significant potential for various applications, including domain adaptation, time-series generation, and more.
4.	The experiments that emphasize the impact of incorporating different features and employing different data assembly processes (e.g., cohort/label definition) are thoroughly investigated.

**Weaknesses:**

1.	Although the authors incorporate a wide range of measurements from the ICU datasets, many therapeutic interventions and comorbidities are missing, which can be crucial for predicting clinical outcomes of interest. Moreover, the time-varying features are mostly continuous while there exist many binary and categorical features in these ICU datasets.

2. Minor comments: typo in p4 “repositoryto” $\rightarrow$ “repository to”

**Questions:**

1.	There exists a harmonized ICU dataset called BlendedICU [A] that incorporates AmsterdamUMCdb, eICU, HiRID, and MIMIC-IV. While the reviewer acknowledges that [A] has not been available online when the proposed work was submitted, what is the distinction and contribution of the proposed work from [A]?
2.	Regarding Weakness #1: As far as the reviewer is aware, variations in units and features can occur both within and across datasets due to differences in observation circumstances (e.g., using different medical devices). How did the authors handle such issues? Additionally, what is the reason for the extracted datasets having missing therapeutic interventions and comorbidities (while some of them are mentioned in Appendix C), which are often extremely important for predicting clinical outcomes?

[A] M. Oliver et al., "Introducing the BlendedICU dataset, the first harmonized, international intensive care dataset," Journal of Biomedical Informatics, 2023.

---

> ### Author Response · Authors · 2023-11-16
> **Response to reviewer Q5t5**
>
> We thank the reviewer for his time in reviewing our manuscript. We address the points made by the reviewer below.
>
> **Additional features**
>
> We chose the 52 most common clinical features shared by all datasets. They were readily available in all benchmarked datasets, demonstrating YAIB's adaptability. Our work focuses on the interoperability of datasets and the opportunity for experiments with a.o. transfer learning and domain adaption. We strongly believe there is the most value in providing a modular setup where the user can add or remove features to suit their needs better and, most importantly, do so reproducibly.
>
>
>
> Nevertheless, several medications for eICU and MIMIC-IV are readily available; the ricu package ([https://github.com/eth-mds/ricu/blob/main/inst/extdata/config/concept-dict.json](https://github.com/eth-mds/ricu/blob/main/inst/extdata/config/concept-dict.json)) maintains a complete list of the currently available native concepts which are available. Complex concepts, dependent on several native concepts, such as SOFA scores, are additionally available. Each concept that is available in ricu can be readily used in YAIB. Some medications that are already implemented, such as antibiotics and vasopressors, are used in the definition of the complex Sepsis endpoint. Therefore, we decided to leave those out to have the same features for each task. This also includes categorical medications. Comorbidities are missing in HiRID and AUMCdb; we decided not to include these for a fair comparison across datasets. However, they can certainly be extracted from eICU and MIMIC; this is also part of our current efforts.
>
> We note, additionally, that it is straightforward to implement new concepts in our pipeline; Appendix E.2 describes the addition of Potassium Chloride to the ICU harmonization package ricu. A similar process can be followed for adding new medications, which immediately improves the usability of YAIB. Moreover, we are actively working on integrating more features, including therapeutic interventions, comorbidities, and medications. We want to note that many features are not available in each dataset; this does not mean they can not be valuable in clinical prediction tasks.
>
>
>
> Finally, we would like to point out that YAIB's end-to-end pipeline is designed as a solid starting point for 1) clinicians looking for external validation to employ ML in practice, 2) dataset creators looking for a solid platform to facilitate widespread use, and 3) the ML community to contribute novel prediction models. They can use a mature and externally developed framework, which adds to the credibility of any experiment results. Adding new feature concepts for their datasets can also increase the adoption of their datasets. They are likely domain experts for their respective datasets, meaning fewer errors are made in this process. This process will improve the usability of YAIB as an end-to-end benchmarking tool and improve the confidence of health experts in clinical ML. We have added our argumentation to Appendix F.2 as well.
>
> ## Questions
>
>
> **BlendedICU**
>
> Thank you for bringing our attention to this work. BlendedICU seems like a good effort as far as a static data combination pipeline is concerned. However, the authors of BlendedICU seem to have completely bypassed the ricu package that does harmonization for our work. This is a major problem when it comes to extensibility. Whereas we are close to adding more datasets to our benchmark, they have a static pipeline that does not seem to provide extensibility. They do not provide cohort generation functionality to define adaptable tasks. Moreover, there is no benchmarking framework which is a crucial part of our end-to-end pipeline to ensure comparable results. We would gladly adapt YAIB to be able to work with this work as well and will attempt to contact the authors to integrate their work into our pipeline to increase the usability of both works; this is why OMOP compatibility is on our list of future work. The work has been added to the related work section in the manuscript to show the distinction between it and YAIB.
>
> **Unit changes**
>
> Unit changes are handled by mapping to a common unit within ricu. This makes our harmonization work in a reproducible manner and allows other datasets to be added and harmonized to the common format. Moreover, different machines can indeed lead to different kinds of erroneous data. The datasets were therefore extensively checked for unrealistic values. Further information is available in Appendix B and C as well as the ricu documentation ([https://eth-mds.github.io/ricu/](https://eth-mds.github.io/ricu/)). We hope to have answered the question about comorbidities in the above section of this response (Additional features).

---

> > ### Comment · Reviewer_Q5t5 · 2023-11-19
> > **Thank you for the rebuttal**
> >
> > Thank you for the detailed answer.
> >
> > I do not have further questions and I think the authors have managed to answer my comments and questions.
> > However, I will stick to my original score of 8.

---

> ### Author Response · Authors · 2023-11-16
> **Response to reviewer Q5t5 (continued)**
>
> **Proposed Actions:**
>
> - Added differences between BlendedICU and YAIB to the related work section and Table 1.
>
> - We have corrected the mentioned typo and checked the manuscript for any typos or inconsistent sentences.
>
> - Added Appendix F.2 “The choice of features”
>
>
> We hope this rebuttal addresses the reviewer's points. We invite them to respond with additional points that could be used to improve our work further.

---

### Official Review · Reviewer_o57d · 2023-10-30

**Soundness:** 3 good
**Presentation:** 4 excellent
**Contribution:** 2 fair
**Rating:** 5
**Confidence:** 3

**Summary:**

In this study, the authors present an ICU-benchmark for clinical machine learning. They employ this framework to benchmark four ICU datasets. The toolkit offered encompasses data preprocessing, model training, and evaluation.

**Strengths:**

The design of the benchmark framework is clear. The paper offers an exhaustive detailing of preprocessing, modeling, and experimental results. The code's structure is clear, and the authors also provide comprehensive guidelines.

**Weaknesses:**

My primary concerns are:

While the effort in constructing a comprehensive benchmark is great, I am uncertain about its alignment with the primary focus of ICLR. The paper appears to lean heavily towards a benchmarking contribution rather than a technical innovation. It might be more fitting for this work to be submitted to NeurIPS's benchmark track or journals such as Scientific Data.

My other comment is about the comparative scope of the study. The authors posit that their framework offers flexibility and the potential for application to other datasets, suggesting its closeness to a clinical ML toolkit. I recommend contrasting this work with existing clinical ML toolkit contributions, such as references [1,2] for a more comprehensive perspective.

[1] Saveliev, E. S., & van der Schaar, M. (2023). TemporAI: Facilitating Machine Learning Innovation in Time Domain Tasks for Medicine. arXiv preprint arXiv:2301.12260.

[2] Yang, C., Wu, Z., Jiang, P., Lin, Z., Gao, J., Danek, B. P., & Sun, J. (2023, August). PyHealth: A Deep Learning Toolkit for Healthcare Applications. In Proceedings of the 29th ACM SIGKDD Conference on Knowledge Discovery and Data Mining (pp. 5788-5789).

**Questions:**

Please see the weaknesses above.

---

> ### Author Response · Authors · 2023-11-16
> **Response to reviewer o57d**
>
> We sincerely appreciate the time and effort invested by the reviewer and have used their feedback to adapt our paper.
>
> **Fit to ICLR**
>
> The reviewer argues that the paper might not be suitable for the venue. We respectfully but strongly disagree with this statement.
>
> Firstly, we would like to refer to the description of the Call for Papers that ICLR provides to authors. The cursive text indicates that the text is taken directly from the ICLR call for papers ([https://iclr.cc/Conferences/2024/CallForPapers](https://iclr.cc/Conferences/2024/CallForPapers)) [Accessed 2023-11-13]:
>
> *We consider a broad range of subject areas including feature learning, metric learning, compositional modeling, structured prediction, reinforcement learning, uncertainty quantification and issues regarding large-scale learning and non-convex optimization, as well as applications in vision, audio, speech, language, music, robotics, games, healthcare, biology, sustainability, economics, ethical considerations in ML, and others.*
>
> Our work fits with the applications in healthcare listed above.
>
> Moreover, our work fits the following specific topics in the listed call for papers:
>
> *- unsupervised, self-supervised, semi-supervised, and supervised representation learning*
>
> *- transfer learning, meta learning, and lifelong learning*
>
> *- datasets and benchmarks*
>
> *- infrastructure, software libraries, hardware, etc.*
>
>
> We provide reasoning on why our work fits these topics. We 1) provide an extensive and extendable framework with various experiments, models, endpoints, and datasets for representation learning for a highly relevant application area. We 2) provide transfer learning and external validation experiments that can be easily reproduced and enhanced. We 3) provide both an adaptable pipeline for generating datasets with various clinical endpoints as well as a comprehensive benchmark that explores the current state of ML for clinical prediction. We 4) provide an end-to-end pipeline and software library with extensive documentation that actively encourages the medical and ML community to contribute.
>
> **Technical contribution**
>
> Additionally, we would like to attend the reviewer to the primary focus of this paper, introducing a "Flexible Multi-Center Framework for Clinical ML." Our technical contribution is in addressing the severe reproducibility challenges in ICU task prediction and, by extension, the reproducibility of ML in general. This is needed because there is currently no extensible dataset-independent end-to-end pipeline for ICU task prediction (see the Related Work section for details). The result is that many works reinvent the wheel by providing opaque, highly divergent methodologies that supposedly demonstrate their models' superiority [1,2]. Code is often unmaintained, limiting authors of new models from being able to compare their newly developed models to just a few existing solutions. This severely limits the usability of any technical innovations developed for the highly relevant task of ICU task prediction.
>
> Very few ICU prediction models make it to practice [3,4]. We have collaborated with clinicians in developing YAIB and writing the manuscript to address these problems. We see our work as a connecting factor between the "real world" of clinical practice and machine learning.
>
> Table 2 in the manuscript demonstrates this through a non-exhaustive list of relevant (ML in health) literature that could benefit from our work. Moreover, our work facilitates future technical contributions in the areas of, for example, transfer learning, domain generalization, domain shift, explainability, and fairness. We specifically bridge the gap between the theory presented at ML venues and the practice of clinical ML.
>
> *[1] Tabinda Sarwar, Sattar Seifollahi, Jeffrey Chan, Xiuzhen Zhang, Vural Aksakalli, Irene Hudson, Karin Verspoor, and Lawrence Cavedon. The Secondary Use of Electronic Health Records for Data Mining: Data Characteristics and Challenges. ACM Computing Surveys, 55(2):1–40, March 2023. doi: 10.1145/3490234.*
>
> *[2] Christopher J. Kelly, Alan Karthikesalingam, Mustafa Suleyman, Greg Corrado, and Dominic King. Key challenges for delivering clinical impact with artificial intelligence. BMC Medicine, 17(1): 195, December 2019. doi: 10.1186/s12916-019-1426-2.*
>
> *[3] Lucas M. Fleuren, Patrick Thoral, Duncan Shillan, Ari Ercole, Paul W. G. Elbers, Mark Hoogendoorn, Ben Gibbison, Thomas L. T. Klausch, Tingjie Guo, Luca F. Roggeveen, Eleonora L. Swart, Armand R. J. Girbes, and Right Data Right Now Collaborators. Machine learning in intensive care medicine: Ready for take-off? Intensive Care Medicine, 46(7):1486–1488, July 2020b. doi: 10.1007/s00134-020-06045-y.*
>
> *[4] Bar Eini-Porat, Ofra Amir, Danny Eytan, and Uri Shalit. Tell me something interesting: Clinical utility of machine learning prediction models in the ICU. Journal of Biomedical Informatics, 132: 104107, August 2022. doi: 10.1016/j.jbi.2022.104107.*

---

> ### Author Response · Authors · 2023-11-16
> **Response to reviewer o57d (continued)**
>
> **Comparison to existing frameworks**
>
> We thank the reviewer for bringing up the preprint of TemporAI, which is still in early development at the time of writing. While we included an earlier work by the same group, Clairvoyance, in our related work, we have now updated the manuscript by adding this work in the related work section and to Table 1. We note that Pyhealth is already included in the related work section of the original manuscript. However, we elaborate on the differences between YAIB and both works below.
>
>
>
> **TemporAl**
>
> TemporAI is a package that is currently in early development without a peer-reviewed publication associated with it. While it promises to provide: "prediction, causal inference, and time-to-event analysis, as well as common preprocessing utilities and model interpretability methods," it is unclear from current documentation how to use established datasets with this package or how to use relevant medical prediction tasks.
>
>
>
> The advantages of YAIB compared to TemporAI are similar to those between YAIB and Clairvoyance: YAIB puts ICU data and tasks front and center for ML scientists and clinicians. YAIB supports the whole workflow, from raw data to clinical concepts to well-defined cohorts. This approach greatly facilitates the transparent and reproducible preprocessing of (often messy) ICU data, which TemporAI, similarly to Clairvoyance, does not cover. We strongly believe that unless tasks can be adapted quickly and reproducibly, it will lead to inevitable ad-hoc adaptations of the task that often end up irreproducible. YAIB, therefore, improves on existing modeling frameworks by putting an equal emphasis on standardized data processing for meaningful model development by providing itself as a benchmark to function as a comparison for other methods. However, we would like to note that using TemporAI (or Clairvoyance) with the YAIB pipeline to create a different end-to-end pipeline is possible as it allows for "swapping out" components. We provide the functionality in our YAIB-cohorts repository to convert any cohort to a format compatible with Clairvoyance and TemporAI.
>
>
>
> **PyHealth**
>
> PyHealth is "a comprehensive deep learning toolkit designed for both ML researchers and healthcare practitioners." PyHealth aims to support all EHR databases. It is thus similar in scope to our proposed framework. Unfortunately, upon closer inspection, PyHealth only supports a small subset of the information in MIMIC and eICU. While diagnoses and prescriptions are, in theory, included, they are processed as a simple bag of diagnosis codes or drug codes without information on strength/duration or semantic interpretation of what they represent (e.g., what is a vasopressor needed in calculating the SOFA score). Vital signs are not supported at all, presumably because PyHealth reads information from raw .csv files and may struggle to process large quantities of vital sign data. PyHealth further states that the datasets are independent of task definitions. This, unfortunately, appears to mean that they have to be implemented anew for each database, with custom dataset-specific code for the same task. Furthermore, all currently available ICU tasks in PyHealth use static data only and do not include any time series.
>
>
>
> Advantages of YAIB compared to PyHealth: YAIB supports all databases within a common, principled interface (see the response on data harmonization above). Moreover, YAIB enables a single task definition that one can directly use for any included dataset. As far as they can work with time series data, YAIB can incorporate any model defined in PyHealth.
>
>
>
> **Proposed actions**
>
> - We have added TemporAI to the related work section and Table 1 and highlighted the limitations compared to YAIB.
>
> - We have added appendix F.4 “Extended related work” to address the reviewer's concerns thoroughly. It includes an extensive discussion on Clairvoyance, TemporAI, and Pyhealth.
>
>
> We hope this rebuttal addresses the reviewer's points. We invite them to respond with additional points that could be used to improve our work further.

---

> ### Comment · Reviewer_o57d · 2023-11-20
>
> Thanks to the authors for their comprehensive replies. I have changed my scores. I agree that the benchmark improves the reproducibility and the possibility of clinical application - I appreciate the efforts. I am still uncertain about the alignment between these contributions and the scope of ICLR.
>
> I didn't find technical flaws in this paper. If the SPC decides that this paper falls within the scope of ICLR, I tend to accept this paper.

---

### Official Review · Reviewer_oWMD · 2023-10-31

**Soundness:** 3 good
**Presentation:** 3 good
**Contribution:** 3 good
**Rating:** 6
**Confidence:** 3

**Summary:**

This paper introduces Yet Another ICU Benchmark (YAIB), a modular framework which allows researchers to define and conduct clinical machine learning (ML) experiments on multiple open-source medical datasets (MIMIC III/IV, eICU, HiRID, AUMCdb). The YAIB framework includes modules for 1) clinical concept definition on harmonized datasets 2) task specification and cohort selection 3) data preprocessing and feature extraction 4) ML model training and evaluation (both traditional ML and deep learning models). Through experiments on five pre-defined clinical tasks, this work demonstrated comparable results of benchmarking baseline models on different datasets, analyzed the impact of varying task/cohort definition and showed the utility of YAIB in transfer learning via a pre-train and fine-tune paradigm.

**Strengths:**

**Originality**: In previous literature, the ICU data benchmarks mostly focused on a single dataset or only supported subsets of multiple datasets, and often required modifications to core codebase for adding new clinical tasks. The originality of this paper lies in that 1) it utilizes the full datasets of five open-source ICU datasets, and for the first time introduced the AmsterdamUMCdb (AUMCdb) dataset into a benchmark 2) it is built on a flexible framework which decouples each individual module and allows users to easily add new tasks or adjust current modules.

**Quality**: This paper specifically accounted for the common issues that researchers encounter when using ICU benchmarks, such as the lack of comparisons across multiple datasets, different task/cohort definitions. Via extensive experiments, this paper demonstrated the utilities of the proposed framework and analyzed the potential impact of small perturbations to the task/cohort definitions by ablation studies. The experiment results supported their claims that 1) comparable results can be achieved when different datasets are harmonized and ML pipelines are unified 2) small changes in definitions can lead to different results.

**Clarity**: This paper is clear and well-written. The tables and figures are informative and easy to understand.

**Significance**: Data processing and clinical task definition have always been a burden for researchers in machine learning for healthcare. Once made public, this work has the potential to help reduce the overhead in benchmarking and developing new ICU prediction methods. This work will also allow researchers to validate their findings by evaluating and comparing the results on multiple real-world ICU datasets. Thanks to the modular implementation, researchers working on other tasks involving ICU data, e.g. reinforcement learning or representation learning, may also benefit from some modules in the proposed framework. Thus, this work has both high technical significance and clinical relevance.

**Weaknesses:**

1. The definition of several terms used in the paper is not clear, including "harmonization", "clinical concepts". Though it can be inferred from later text what they may refer to, I think it would still benefit the readers if you can clearly define them the first time you use them in the paper.

2. Based on the experiments, it seems that the time series features are extracted from numeric data rather than waveform data (please correct me if I am wrong). Currently, most ICU benchmarks used numeric data for prediction tasks and only very few used waveform data, but waveform data are very informative and may greatly help improve the predictive performance. Thus I think the contribution would be more solid if waveform data can be utilized for a new ICU benchmark.

3. Basic statistics of the datasets, e.g. number of patients, ICU stays, and the prevalence of classes or the range of regression targets, are missing in the main paper (found them in appendix). Inclusion of such information will help readers get a general idea of the datasets and understand some of the experiment results, e.g. the discrepancy in the performances shown in Figure 2 may be due to the difference in prevalence in mortality across the datasets. Also, for the pooled dataset, is there any possibility that one dataset is dominating the training set for any task?

**Questions:**

1. In Section 4, you investigated the effects of small variations in task definitions. For the three variations you tried, the first and second are variations to the cohort and the feature selection but not to the clinical task definition, only the third is varying the definition of sepsis - which I understand is the only one directly relevant to clinical task definition. In this case, you may need to change how you describe the variations or redefine what you mean by "task" to avoid the confusion.

2.  In Section 4 Preprocessing, for the aggregated features, what are the time windows for aggregation and the frequencies that you update those features? Is there any specific consideration in the choice for them?

3. In Table 4, it would be helpful to also include the range or the units of KF/LoS, or additionally show the MAEs as % relative to the valid ranges of KF/LoS (absolute MAEs do not provide much information).

4. A minor typo, a space is missing between "repository" and "to" near the bottom on Page 4.

---

> ### Author Response · Authors · 2023-11-16
> **Response to reviewer oWMD**
>
> We thank the reviewer for their insightful comments that we have used to improve our manuscript.
>
>
>
> **Unclear concepts and terms**
>
> We thank the reviewer for the suggestions to clarify certain terms in our manuscript. The manuscript has been updated to reflect these changes. If any other terms are unclear, please feel free to address them in a response.
>
>
>
> **Waveform data**
>
> The reviewer has brought up that our work could be improved if we included support for waveform data. We understand waveform data to be time series data with a certain sampling frequency (e.g., 10 Hz). Waveform data are already included in our work as time-series features. These have been discretized to the same period for each dataset. As an example of highly frequent measurements, we include discretized heart rate (hr), respiratory rate (resp), oxygen saturation (o2sat), mean arterial pressure (map), systolic blood pressure (sbp), and diastolic blood pressure (dbp). Table 13 in the appendix shows that, e.g., MIMIC has a lower resolution for some features. It is relatively straightforward to create a preprocessing pipeline that can include waveform data, and it can be considered as future work once more authors publish waveform data. Moreover, we refer to the general response (Choice of Features) and the newly added Appendix F.2 for considerations on the chosen features.
>
>
>
> **Basic statistics for the datasets**
>
> The reviewer argues that the dataset statistics should be in the main text. We acknowledge the importance of providing basic statistics for each dataset. We have adjusted the description of the mentioned experiment by providing clarification and reference to the mentioned table. However, due to the page limit, we only see a way of providing this information in the main text if we cut important manuscript parts.
>
> The reviewer points out the possibility of one dataset dominating another in pooled training for external validation. We have taken measures to prevent this. For the external validation experiments, we take a stratified sample of the same amount of patients for each dataset (10,000 for the mortality task). A short elaboration on this measure has been added to the experiments section. Further details can be found in Appendix G.5.
>
>
>
> **Answers to the questions**
>
> 1. We have changed the subsection header to: "Using YAIB as an experimental framework" to avoid the confusion of using "task" when referring to a prediction task. We hope this clears up any confusion.
>
> 2. If we understand the reviewer's question correctly, they seem to refer to the aggregation of time series for non-deep learning methods. We have chosen to do this once per stay as we seem to get reasonable results for this aggregation. However, one can change this to multiple times per stay by customizing the preprocessing pipeline (appendix D3). For the general preprocessing to attain the "harmonized cohorts," we refer to Appendix B. If the answer refers to the aggregation per hour, we provide details in Appendix B as well; the main reason is that MIMIC provides measurements once per hour. We hope this answers the question, but invite the reviewer to ask follow-up questions.
>
> 3. We acknowledge that absolute MAE might not mean without context, although one can still compare the performance of each method relative to each other. It is still informative to have the absolute MAE, as it is a universally recognized metric. Table 14 contains the average and IQR for each task. To increase the interpretability of the results and accommodate the reviewers' feedback, we have included the units of both Kidney function and Length of Stay in Table 4 and provide a footnote to refer to Table 14 (where one can find the average and IQR for each dataset).
>
> 4. Thank you for pointing out the typo. It has been corrected in the updated manuscript.
>
>
> **Proposed actions:**
>
> - We have added extra elaboration to the manuscript for some terms that can be interpreted ambiguously.
>
> - To avoid confusion, we renamed section 4.3 to "Using YAIB as experimental ML framework."
>
> - An explanation for the divergent score of external validation for AUMCdb has been added, as well as a reference to the elaborate dataset and cohort table (Table 14).
>
> - We added Appendix F.1 and F.2 to provide the choice of features.
>
> - In Table 4, we added the units and refer to Table 14 for details on the average and IQR.
>
> - We have corrected the mentioned typo and checked the manuscript for any typos or inconsistent sentences.
>
>
> We hope this rebuttal addresses the reviewer's points. We invite them to respond with additional points that could be used to improve our work further.

---

> > ### Comment · Reviewer_oWMD · 2023-11-20
> >
> > Thank you for the detailed answers.
> >
> > The authors have answered most of my comments and questions. One question I would like to clarify is: for **Waveform data**, what I mean is high-resolution waveform data, e.g. the physiologic signals collected at 125 Hz in MIMIC-III Waveform Database (https://physionet.org/content/mimic3wdb/1.0/). Based on the time-series frequencies in Table 13, it seems that you are using the time-series numerics data in MIMIC-III Clinical Database rather than the continuous waveform (please correct me if I am wrong).
> >
> > I do not have further questions.

---

> ### Author Response · Authors · 2023-11-20
> **Second Response to Reviewer oWMD**
>
> Dear reviewer,
>
>
>
> Thank you for taking the time to clarify your response to our response. Let us elaborate on why we do not include waveform data. We indeed took the primary datasets for MIMIC-IV, MIMIC-III, eICU, HiRID, and AUMCdb. The reasons we did not include the waveform databases is because:
>
> 1. Waveform is, to the best of our knowledge, only available for MIMIC-IV and MIMIC-III.
>
> 2. Waveform data is only available for some of the features.
>
> 3.  Waveform data requires separate (signal-related) feature extraction that differs from the primary database.
>
> 4.  Waveform is only available for a subset of the patients of the patients of MIMIC-III and a fraction of the MIMIC-IV patients.
>
>
>
>
> **Reason 1**
>
> We could not do the same preprocessing for the other datasets (HiRID, AUMCdb), and including them in a cross-dataset comparison could be an unfair method.
>
>
>
> **Reason 2**
>
> We would need to drastically increase the time intervals for each benchmark for a minority of the features. This is easily done within YAIB but would immensely increase model training time. Moreover, it would mean that a majority of the features will remain the same in resolution so the theoretic improvement in performance is limited; for MIMIC-III, mainly heart rate (hr), respiratory rate (rr), blood oxygen saturation (o2sat), and arterial blood pressure (abp) are provided [1] which are all already in YAIB; for ECG see below.
>
>
>
> **Reason 3**
>
> For features like ECG, we need a separate feature preprocessing pipeline. We are aware of existing feature extraction/preprocessing pipelines that convert them to dataloader-ready data, so integration into YAIB is theoretically possible. Our focus, however, was on providing a multi-center framework, which resulted in concentrating our efforts on a unified feature set for each database.
>
>
>
> **Reason 4**
>
> MIMIC waveform datasets do not include all the patients in the primary dataset Afshar et al. [1] extracted about 11,000 ICU stays from 9,000 patients that do not necessarily overlap with the “regular” MIMIC-III database.
>
> For MIMIC-IV, there is an even smaller amount of waveform data:
>
>
>
> *This initial release contains 200 records from 198 patients.* (from: [https://physionet.org/content/mimic4wdb/0.1.0/](https://physionet.org/content/mimic4wdb/0.1.0/) [Accessed 2023-11-19])
>
>
>
> It is hard to justify training models on such a small subset.
>
>
>
> [1] Afshar AS, Li Y, Chen Z, Chen Y, Lee JH, Irani D, Crank A, Singh D, Kanter M, Faraday N, Kharrazi H. An exploratory data quality analysis of time series physiologic signals using a large-scale intensive care unit database. JAMIA Open. 2021 Aug 2;4(3):ooab057. doi: 10.1093/jamiaopen/ooab057. PMID: 34350392; PMCID: PMC8327372.
>
>
>
> **Future work**
>
> We nevertheless thank the reviewer for the suggestion. A standardized feature extraction pipeline of waveform data could be valuable for future work as more data becomes available. We would happily collaborate to make this a reality; YAIB’s nature of adaptability provides a sound basis for the remaining pipeline.

---

> > ### Comment · Reviewer_oWMD · 2023-11-20
> >
> > Thank you for the comprehensive replies. I do not have further questions and will stick to my original score of 6.

---

### Official Review · Reviewer_VWQN · 2023-11-01

**Soundness:** 3 good
**Presentation:** 3 good
**Contribution:** 2 fair
**Rating:** 6
**Confidence:** 3

**Summary:**

This paper introduces an ICU benchmark that incorporates several ICU datasets, including MIIMC, eICU, HiRID, and AUMCdb. It covers multiple tasks such as mortality risk, kidney function, sepsis, acute kidney injury, and length of stay predictions. Furthermore, the paper offers an end-to-end pipeline for data preprocessing, model construction, training, and evaluation.

**Strengths:**

- The paper addresses issues of comparability and reproducibility, both of which are crucial in the field of machine learning for healthcare.
- The benchmark unifies four commonly used datasets and allows transfer learning.
- The experiments on adapting task definitions clearly demonstrate the impact of cohort definitions, preprocessing strategies, and training protocols.

**Weaknesses:**

- I acknolwedge the motivation of this paper and appreciate considerable effort invested in establishing such a benchmark dataset. However, while the paper's central claim is the adaptability of the YAIB benchmark to other datasets, tasks, and models, this assertion necessitates a comprehensive experience with the benchmark, which is challenging to evaluate during the brief review period.

- Datasets like MIMIC and eICU are highly complex and heterogeneous. The proposed YAIB benchmark only supports 52 clinical features (mainly vital signs and lab tests). A vast amount of other available features are ignored (like diagnosis, prescriptions, clinical notes, x-rays). This limited feature support might hinder future model advancements (which can be seen in Tables 3 & 4 where the performance between classic ML and DL models are very similar).

**Questions:**

Refer to weaknesses section.

---

> ### Author Response · Authors · 2023-11-16
> **Response to reviewer VWQN**
>
> We sincerely appreciate the time and effort invested by the reviewer in evaluating our manuscript. We would like to address the points mentioned by reviewer VQQN.
>
>
>
> **Adaptability of YAIB**
>
> The reviewer argues it is hard to assess the adaptability to other datasets, tasks, and models. Firstly, we would like to stress that YAIB is not a benchmark dataset but rather a framework that standardizes the process of benchmarking new methods on ICU/EHR data.
>
> Furthermore, the YAIB pipeline has helped us to produce reproducible results quickly and provides the required extensibility for our purposes. We are in touch with some researchers who have used YAIB to date and have helped improve the framework. We additionally refer to the work “Closing Gaps: An Imputation Analysis of ICU Vital Signs” accepted to the 1st Workshop on Deep Generative Models for Health at NeurIPS 2023 in the supplemental material. The authors of this work used YAIB as a bedrock for implementing imputation methods and are in the process of extending this to more methods and downstream tasks. For concrete examples of how to extend YAIB, for other authors, we refer to Appendix D and the wiki documentation we added to the supplemental materials.
>
>
>
> Therefore, we would argue that our work provides solid evidence of the adaptability the reviewer has pointed out. We further elaborate on the design of YAIB in our common response and Appendix F.1. We hope this addresses possible uncertainty regarding adaptability.
>
> **Heterogeneity of datasets and support of features**
>
> The reviewer argues that the choice of features might limit usability. We chose the 52 most common clinical features shared by all datasets. They were readily available in all benchmarked datasets, demonstrating YAIB's adaptability. Our work focuses on the interoperability of datasets and the opportunity for experiments with a.o. transfer learning and domain adaption. We strongly believe there is the most value in providing a modular setup where the user can add or remove features to suit their needs better and, most importantly, do so reproducibly.
>
> Nevertheless, several medications for eICU and MIMIC-IV are readily available; the ricu package (https://github.com/eth-mds/ricu/blob/main/inst/extdata/config/concept-dict.json) maintains a complete list of the currently available native concepts which are available. Complex concepts, dependent on several native concepts, such as SOFA scores, are additionally available. Each concept that is available in ricu can be readily used in YAIB. Some medications that are already implemented, such as antibiotics and vasopressors, are used in the definition of the complex Sepsis endpoint. Therefore, we decided to leave those out to have the same features for each task.
>
> We note, additionally, that it is straightforward to implement new concepts in our pipeline; Appendix E.2 describes the addition of Potassium Chloride to the ICU harmonization package ricu. A similar process can be followed for adding new medications, which immediately improves the usability of YAIB. Moreover, we are actively working on integrating more features, including comorbidities and medications. We want to note that many features are not available in each dataset; this does not mean they can not be valuable in clinical prediction tasks. The reviewer additionally argues we might support clinical notes and X-rays as well. Unfortunately, these are not available across all datasets, which was the focus of this work, and require very different (non-timeseries oriented) models. Additionally, we see this as a feature that is rarely used by existing ICU prediction models.
>
> Given the reasoning above, we disagree with the reviewer’s suggestion that future model advancements are hindered by the chosen features. However, we see this as a natural extension of YAIB when moving towards general EHR data. To accommodate the reviewer, we have added this point to the discussion.
>
> Additionally, we would like to point out that YAIB's end-to-end pipeline is designed as a solid starting point for 1) clinicians looking for external validation to employ ML in practice, 2) dataset creators looking for a solid platform to facilitate widespread use, and 3) the ML community to contribute novel prediction models. They can use a mature and externally developed framework, which adds to the credibility of any experiment results. Adding new feature concepts for their datasets can also increase the adoption of their datasets. They are likely domain experts for their respective datasets, meaning fewer errors are made in this process. This process will improve the usability of YAIB as an end-to-end benchmarking tool and improve the confidence of health experts in clinical ML.

---

> ### Author Response · Authors · 2023-11-16
> **Response to reviewer VWQN (continued)**
>
> **Similar performance ML and DL**
>
> Furthermore, we disagree with the reviewer that the similar performance of our experiments can solely be explained by little features. Our preprocessing pipeline produces processed datasets suitable to both ML and DL; the effect is that both types of models can perform similarly. The use of “regular” ML is still widespread in the medical field as it does not require large processing power and provides explainability. Additionally, we refer to the HiRID benchmark [1], which has similar differences between ML and DL while including most of the features specific to HiRID. They provide feature generation that allows the LGBM model even to outperform deep learning models.
>
> We address the choice of features in the above response, in the general response (Choice of Features), and in Appendix F.2.
>
> *[1] Hugo Yèche, Rita Kuznetsova, Marc Zimmermann, Matthias Hüser, Xinrui Lyu, Martin Faltys, and Gunnar Rätsch. HiRID-ICU-Benchmark – A Comprehensive Machine Learning Benchmark on High-resolution ICU Data. arXiv:2111.08536 [cs], January 2022*
>
> **Proposed actions:**
>
> - Added a wiki to the supplementary materials to show users how to adapt datasets, tasks, preprocessing, and models
>
> - Added an anonymized version of the work “Closing Gaps: An Imputation Analysis of ICU Vital Signs” accepted to the 1st Workshop on Deep Generative Models for Health at NeurIPS 2023 to the supplemental material. The work uses YAIB as an extendable base to show the performance of several imputation methods on ICU vital signs.
>
> - Added an explanation in the experiments section of the manuscript to indicate that one is not limited to the used features but 1) can use more features for some datasets and 2) use our extensive guide in Appendix E or our wiki.
>
> - Added Appendix F.1 and F.2 to provide information on extensibility and reproducibility and the choice of features.
>
> - Added clinical notes and medical images as future extensions in the discussion.
>
>
> We hope this rebuttal addresses the reviewer's points. Additionally, we invite them to respond with additional points that could be used to improve our work.

---

> ### Author Response · Authors · 2023-11-21
> **Response to reviewer VWQN [2]**
>
> Dear reviewer VWQN,
>
> We greatly appreciate your first review, which helped us strengthen our submission considerably. We would strongly appreciate it if you could take the time to respond to the reply we have given you within the discussion period. This allows us to provide additional explanations and improvements where needed and further strengthen our submission. Thank you for your time and efforts.

---

> > ### Comment · Reviewer_VWQN · 2023-11-23
> >
> > Thanks to the authors for their reply. I acknowledge the motivation of this paper and appreciate the considerable effort invested in establishing such a benchmark dataset. I will raise my ratings.

---

### Author Response · Authors · 2023-11-16
**General Rebuttal Response**

We thank each reviewer for their extensive feedback and have worked hard to incorporate it into the manuscript, supplemental materials, and software repository. We provide an overview of the changes below. Additionally, we provide a common response to authors. We have provided this for external readers and the area chair as we have individualized responses to each reviewer.

We have made the following content additions to the manuscript:
- Added TemporAI and BlendedICU to the related work (Table 1) and motivated why our work fills an important gap that none of the related works address.
- Added an explanation in the experiments section of the manuscript to indicate that one is not limited to the used features but 1) can use more features for some datasets and 2) use our extensive guide in Appendix E or our wiki.
- An explanation for the divergent score of external validation for AUMCdb has been added, as well as a reference to the elaborate dataset and cohort table (Table 14).
- Added sentence to the design philosophy that indicates that users of YAIB should openly provide their methodology and code and compare it with the original.

Some clarifications to the manuscript have also been made:
- Changed the description of some experiment results to indicate relative performance.
- To avoid confusion, we renamed section 4.3 to “Using YAIB as an experimental ML framework”.
- We have added extra elaboration to the manuscript for some terms that can be interpreted ambiguously.
- Added clinical notes and medical images as future extensions in the discussion.
In Table 4, we added the units and refer to Table 14 for details on the average and IQR.
- We have corrected a mentioned typo and checked the manuscript for other typos or inconsistent sentences.

Additionally, we have added Appendix F: YAIB’s contribution in context that contains
1. An extended discussion on potential tradeoffs of extensibility and reproducibility similar to that found below.
2. A commentary on the choice of the features used in YAIB similar to that found below.
3. A guide to providing transparent ML development for future authors to extend or change YAIB.
4. “Extended related work” to thoroughly address the differences between YAIB and existing work. It includes an extensive discussion on Clairvoyance, TemporAI, and Pyhealth.

We have made the following changes to the repository and supplements:
- Added detailed wiki as a folder with md files that describe how to use and extend YAIB (will be available in the GitHub repository to all users). Start at YAIB-wiki-home.md to read the wiki. This addresses the reviewer's comments regarding the use of YAIB.
- Added standalone Python files in the folder “Adding a model” to demonstrate adding a model to YAIB by providing the external files. Adding just two short files allows us to use a new model for all datasets and tasks in most cases. We have included an instruction in the wiki for adding more models.
- Added an anonymized version of the work “Closing Gaps: An Imputation Analysis of ICU Vital Signs” accepted to the 1st Workshop on Deep Generative Models for Health at NeurIPS 2023 to the supplemental material. The work uses YAIB as an extendable base to show the performance of several imputation methods on ICU vital signs.

We provide two more sections of “general response,” which have been mentioned by two or more reviewers as an overview to the area chair.

---

> ### Author Response · Authors · 2023-11-16
> **Extensibility and Reproducibility (VWQN, 5BSM)**
>
> We designed YAIB to be as extensible as possible while retaining full reproducibility. This means easy support of new databases, clinical concepts, tasks, experiment configurations, preprocessing pipelines, imputation methods, models, and evaluation metrics. If changes are necessary, they need to be reproducible and easily shareable across research teams.
> If the user only requires a few default ICU tasks from a single i.i.d. dataset to test their new method, any existing ICU benchmarks could be sufficient. Users do not need to apply for access to multiple datasets and do not have to deal with the intricacies of the clinical task definition. As long as the integration of a new model is seamless, such simple frameworks are fit-for-purpose and abstract much of the complexity, allowing the user to only worry about one thing: their model. If multiple papers used the exact same benchmark, results are also directly comparable between papers (an “apple-to-apple” comparison).
>
> However, we found that this setup tends to be too restrictive and thus unrealistic. Users often want to highlight a particular aspect of their model, prompting them to adapt to the default task. At other times, they want to show clinical impact and need to adapt the default task to make it more realistic. Given the lack of successful translation of prediction models into clinical practice, reviewers are also increasingly requesting external validation – sometimes with multiple endpoints – which is difficult to shoehorn into most existing solutions.
> YAIB embraces the need to tweak experimental setups. Results will no longer be directly comparable between papers, but we argue that actual apples-to-apples comparisons were inherently rare. Instead of forcing users into a rigid framework, it allows for adaptations but requires them to be done transparently. Absolute performance should be compared only within the same paper or among papers with the same task setup (see our examples in Tables 5 and 6).
>
> To facilitate the transparency of adaptations, we rely on a sophisticated framework to define clinical concepts across multiple datasets (ricu). We have adapted and extended ricu to provide a standard workflow for YAIB to integrate new databases and define new clinical concepts. To date, it has been successfully used to bring 4/5 +1 ICU datasets into a common format (including our addition of the Salzburg Intensive Care Database, which is currently in quality control). This approach is flexible enough that we have yet to encounter significant restrictions in mapping admissions, demographics, vital signs, laboratory values, medication (including rates and durations), clinical scores, and outcomes at different time scales across datasets. The main restriction of ricu is that it is currently implemented in the R language only, but we provide guidance on how to access it via rpy2, and we are in the process of porting it to Python; this will make our pipeline even more accessible, especially to clinical researchers. Our cohort definition functionality provides helper functions to apply inclusion/exclusion criteria on top of ricu and report step-by-step attrition numbers. The cohorts can be used in a modular fashion with custom preprocessing steps, imputations, prediction models, and evaluation metrics, all using the exact same code across multiple datasets.
>
> Even so, there will likely be situations where the user may be better off with a custom solution. We expect this to occur once their use case diverges significantly from standard supervised learning. For example, federated learning or reinforcement learning setups may require significantly different training and evaluation loops. These are not currently supported, but we consider this as future work. In any case, the user can still use our data processing, cohort generation, and possibly other parts of YAIB (e.g., by exchanging the default training module with a custom module). Authors using YAIB should, therefore, provide their code and a detailed list of the changes they have made to the repository; modern version control allows us to verify this against the original YAIB repository easily. The newly added Appendix F.3 addresses the exact steps to provide transparency for novel work.
>
> The YAIB pipeline has helped us to produce reproducible results quickly and provides the required extensibility for our purposes. We are in touch with some researchers who have used YAIB to date and provided feedback, although mainly in an informal way. We refer to the work “Closing Gaps: An Imputation Analysis of ICU Vital Signs” in the supplemental material. The authors of this work used YAIB as a bedrock for implementing imputation methods and are in the process of extending this to more methods and downstream tasks. For concrete examples of how to extend YAIB, for other authors, we refer to Appendix D and the wiki documentation we added to the supplemental materials.

---

> ### Author Response · Authors · 2023-11-16
> **Choice of Features (VWQN, oWMD, Q5t5)**
>
> We chose the 52 most common clinical features shared by all datasets. They were readily available in all benchmarked datasets, demonstrating YAIB's adaptability. Our work focuses on the interoperability of datasets and the opportunity for experiments with a.o. transfer learning and domain adaption. We strongly believe there is the most value in providing a modular setup where the user can add or remove features to suit their needs better and, most importantly, do so reproducibly.
>
> Nevertheless, several medications for eICU and MIMIC-IV are readily available; the ricu package (https://github.com/eth-mds/ricu/blob/main/inst/extdata/config/concept-dict.json) maintains a complete list of the currently available native concepts which are available. Complex concepts, dependent on several native concepts, such as SOFA scores, are additionally available. Each concept that is available in ricu can be readily used in YAIB. Some medications that are already implemented, such as antibiotics and vasopressors, are used in the definition of the complex Sepsis endpoint. Therefore, we decided to leave those out to have the same features for each task.
>
> We note, additionally, that it is straightforward to implement new concepts in our pipeline; Appendix E.2 describes the addition of Potassium Chloride to the ICU harmonization package ricu. A similar process can be followed for adding new medications, which immediately improves the usability of YAIB. Moreover, we are actively working on integrating more features, including comorbidities and medications. We want to note that many features are not available in each dataset; this does not mean they can not be valuable in clinical prediction tasks.
>
> Finally, we would like to point out that YAIB's end-to-end pipeline is designed as a solid starting point for 1) clinicians looking for external validation to employ ML in practice, 2) dataset creators looking for a solid platform to facilitate widespread use, and 3) the ML community to contribute novel prediction models. They can use a mature and externally developed framework, which adds to the credibility of any experiment results. Adding new feature concepts for their datasets can also increase the adoption of their datasets. They are likely domain experts for their respective datasets, meaning fewer errors are made in this process. This process will improve the usability of YAIB as an end-to-end benchmarking tool and improve the confidence of health experts in clinical ML.

---

### Author Response · Authors · 2023-11-19

Dear Reviewers,

We have carefully addressed all your comments in the past week by updating the manuscript and the supplementary materials. The improved material was released as soon as possible so you would have time to respond to the updated work and our replies to your reviews. We would greatly appreciate your feedback so your reviews may reflect the updated material.

Again, we thank you for the feedback you have given us that enabled us to improve and clarify our work.

Best,

The authors

---

### Meta-Review · Area_Chair_dhz1 · 2023-12-13

**Metareview:**

This well-written paper has been assessed by five knowledgeable reviewers. Four of them recommended its acceptance (three at a marginal level, one full accept) and one voted for marginal rejection. The proposed benchmark does have limitations, correctly pointed out by the reviewers. Perhaps the most limiting in practice is lack of direct support for high density waveform data that had been shown in prior work to bring clinically useful information not reflected in downsampled "numeric" rendition of bedside monitoring data. However, the work is relevant to ICLR and sufficiently mature to be included in this year's proceedings.

**Justification For Why Not Higher Score:**

This paper, not without limitations, is just above the acceptance threshold.

**Justification For Why Not Lower Score:**

The reviewers have provided generally positive assessments.

---

### Decision · Program_Chairs · 2024-01-16

Accept (poster)